# Bongard-RWR+: Real-World Representations of Fine-Grained Concepts in Bongard Problems

**Szymon Pawlonka**[1]  **Mikołaj Małkiński**[1]  **Jacek Mańdziuk**[1,2]
[1]Warsaw University of Technology, Warsaw, Poland
[2]AGH University of Krakow, Krakow, Poland
`{szymon.pawlonka.stud, mikolaj.malkinski.dokt,`
`jacek.mandziuk}@pw.edu.pl`

## Abstract

Bongard Problems (BPs) provide a challenging testbed for abstract visual reasoning (AVR), requiring models to identify visual concepts from just a few examples and describe them in natural language. Early BP benchmarks featured synthetic black-and-white drawings, which might not fully capture the complexity of real-world scenes. Subsequent BP datasets employed real-world images, albeit the represented concepts are identifiable from high-level image features, reducing the task complexity. Differently, the recently released Bongard-RWR dataset aimed at representing abstract concepts formulated in the original BPs using fine-grained real-world images. Its manual construction, however, limited the dataset size to just 60 instances, constraining evaluation robustness. In this work, we introduce Bongard-RWR+, a BP dataset composed of $5\,400$ instances that represent original BP abstract concepts using real-world-like images generated via a vision language model (VLM) pipeline. Building on Bongard-RWR, we employ Pixtral-12B to describe manually curated images and generate new descriptions aligned with the underlying concepts, use Flux.1-dev to synthesize images from these descriptions, and manually verify that the generated images faithfully reflect the intended concepts. We evaluate state-of-the-art VLMs across diverse BP formulations, including binary and multiclass classification, as well as textual answer generation. Our findings reveal that while VLMs can recognize coarse-grained visual concepts, they consistently struggle with discerning fine-grained concepts, highlighting limitations in their reasoning capabilities.

## 1 Introduction

Abstract visual reasoning (AVR) domain (Stabinger et al., 2021; van der Maas et al., 2021) refers to visual tasks, solving which requires identifying and reasoning about abstract patterns expressed through image-based analogies. Classical AVR tasks and associated benchmark problems include Raven's Progressive Matrices (RPMs) (Barrett et al., 2018; Zhang et al., 2019), visual analogy problems (Hill et al., 2019; Webb et al., 2020), and more (Małkiński & Mańdziuk, 2023). These tasks mainly focus on testing the ability of systematic reasoning and generalisation to new feature distributions. Nevertheless, they typically require supervised training on large-scale datasets. This stands in contrast to human intelligence assessments, which focus on rapid adaptation to novel, potentially never-encountered problems based on prior knowledge.

A distinct alternative to conventional AVR benchmarks is offered by Bongard Problems (BPs), originally proposed in 1970 (Bongard, 1970). Each BP consists of two sides, each containing six images, separated according to an abstract rule. The solver's task is to infer this underlying concept and articulate it in natural language (see Fig. 1). An alternative BP formulation relies on classifying a novel test image (or a pair of test images) to appropriate side(s). Despite its deceptively simple format, the BP setting poses several unique challenges. First, it naturally constitutes a few-shot learning problem (Fei-Fei et al., 2006; Wang et al., 2020b) – identifying the separating concept requires generalization from just six examples per side. Second, the interpretation of images is inherently contextual (Linhares, 2000). For instance, in Fig. 1a, a visual feature such as curvature may appear salient on the left side, but only through comparison with the opposite side the true rule, a difference

in arrow directions, becomes evident. Third, solving BPs requires integrating visual perception with text generation, demanding bi-modal reasoning capabilities.

Early BPs were created manually, resulting in only a few hundred instances contributed by individuals (Foundalis, 2006b). To scale beyond this limitation, Bongard-LOGO (Nie et al., 2020) introduced a procedural generation method based on the action-oriented LOGO language, enabling creation of a large set of problems. While this approach provided sufficient data for training deep models, the resulting BPs were restricted to synthetic black-and-white drawings. Subsequent efforts have shifted BPs into the real-world domain. Bongard HOI (Jiang et al., 2022) utilized natural images of human-object interactions. Bongard-OpenWorld (Wu et al., 2024) employed an open-vocabulary of free-form concepts to capture the complexity and ambiguity of real-world scenes. Most recently, Bongard-RWR (Małkiński et al., 2025) linked synthetic and real-world domains by using real images to represent abstract concepts found in synthetic BPs, facilitating direct comparison of model performance across both domains. Bongard-RWR, which focuses on abstract concepts, poses a greater challenge to contemporary models than Bongard HOI and Bongard-OpenWorld, which involve real-world concepts, despite all three using real-world images (see Fig. 2 for an illustration). However, since Bongard-RWR was constructed manually, its small scale limits the robustness and breadth of possible evaluations, highlighting the need for a more scalable approach to dataset construction.

To overcome the scalability limitations of Bongard-RWR, we introduce Bongard-RWR+, a new benchmark featuring real-world representations of selected synthetic BPs (see an overview in Table 1). Unlike Bongard-RWR, which was constructed manually, our approach leverages recent advances in vision language models (VLMs), including Image-to-Text (I2T) and Text-to-Image (T2I) models, to automate dataset creation. For each BP in the original Bongard-RWR, we (1) use Pixtral-12B (MistralAI, 2024) to describe each image, (2) generate new image descriptions aligned with the matrix concept, (3) employ Flux.1-dev (Labs, 2024) to synthesize images from these descriptions, and (4) manually verify that generated images reflect the intended concept (see Fig. 3). This pipeline enables construction of a dataset comprising $5\,400$ BPs with the original abstract concepts preserved.

We conduct a comprehensive evaluation on Bongard-RWR+ to systematically assess the reasoning capabilities of state-of-the-art VLMs across a range of problem setups. Specifically, we formulate: (1) binary classification tasks, where the model assigns either a single test image or a pair of images to the correct side(s) of the matrix; (2) a multiclass classification task, where the model selects the concept that best matches the BP matrix from a set of candidates; and (3) a free-form text generation task, in which the model articulates the underlying concept in natural language. Beyond these primary tasks, we perform several ablations to investigate factors influencing model performance. These include examining the effect of model size, comparing color vs. greyscale inputs, contrasting model performance on BPs constructed with real vs. generated images, and varying the number of images per matrix side. Our findings show that, while current VLMs exhibit some capacity to identify high-level, coarse-grained concepts, they consistently struggle with discerning fine-grained concepts, highlighting gaps in their visual reasoning capabilities.

**Contributions.** In summary, this paper makes the following contributions:

1) We develop a semi-automated pipeline to generate real-world-like images of abstract concepts.

2) We introduce Bongard-RWR+, a new BP benchmark of $5\,400$ matrices generated with this pipeline.

3) We conduct extensive evaluations on the proposed dataset, demonstrating that current VLMs struggle to identify fine-grained visual concepts, revealing important limitations in their multi-image AVR abilities.

## 2 RELATED WORK

**Overview of AVR challenges.** Analogy-making is a cornerstone of human cognition, closely tied to fluid intelligence—the ability to generalize learned skills to novel situations (Lake et al., 2017). In the visual domain, such tasks present demanding challenges for representation learning (Chalmers et al., 1992) and highlight the need to integrate perception and cognition (Hofstadter, 1995), motivating diverse AVR benchmarks. Among these, RPMs (Raven, 1936; Raven & Court, 1998) are an often studied problem type (Hoshen & Werman, 2017; Małkiński & Mańdziuk, 2024b; 2025; Mańdziuk & Żychowski, 2019), typically involving a small set of logic-based rules. Datasets like SVRT (Fleuret

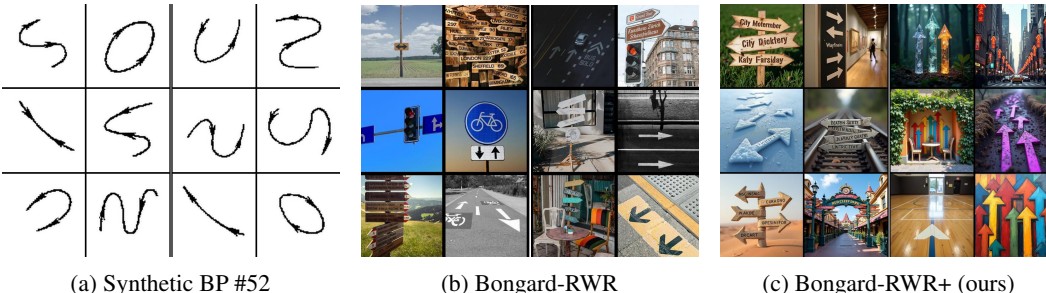

(a) Synthetic BP #52     (b) Bongard-RWR     (c) Bongard-RWR+ (ours)

Figure 1: **Bongard Problems.** All matrices present the same abstract concept: Left side: Arrows pointing in different directions. Right side: Arrows pointing in the same direction. (a) A manually-designed synthetic BP (Bongard, 1970; Foundalis, 2006b). (b) A manually-designed real-world representation from Bongard-RWR (Małkiński et al., 2025). (c) An automatically generated real-world representation from Bongard-RWR+.

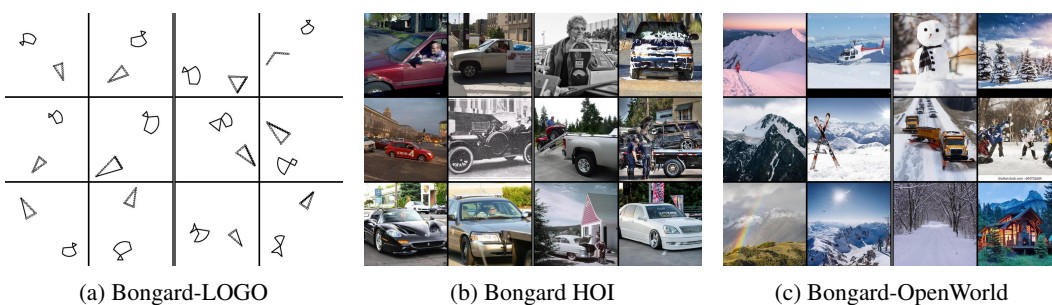

(a) Bongard-LOGO     (b) Bongard HOI     (c) Bongard-OpenWorld

Figure 2: **Related BP datasets.** (a) A synthetic BP from Bongard-LOGO (Nie et al., 2020); Left: Shapes are the same. Right: Shapes are different. (b) A real-world BP from Bongard HOI (Jiang et al., 2022); Left: A person driving a car. Right: Not a person driving a car. (c) A real-world BP from Bongard-OpenWorld (Wu et al., 2024); Left: The top of a snow-covered mountain. Right: Not the top of a snow-covered mountain. Unlike Bongard-LOGO, which involves synthetic images unfamiliar to VLMs, or Bongard HOI and Bongard-OpenWorld, which focus on coarse-grained concepts, Bongard-RWR+ is designed around abstract concepts expressed through realistic images that require fine-grained visual reasoning.

et al., 2011) and those based on BPs (Jiang et al., 2022; Małkiński et al., 2025; Nie et al., 2020; Wu et al., 2024) introduce more abstract and diverse rules expressed through visual analogies in a few-shot learning framework. Another line of work has proposed generative challenges, such as ARC (Chollet, 2019) and PQA (Qi et al., 2021). Several benchmarks (Bitton et al., 2023; Ichien et al., 2021; Teney et al., 2020) construct AVR tasks using real-world images, enabling evaluation of models pre-trained on large visual datasets. Our proposed dataset builds on this landscape by combining few-shot learning with real-world-like images and supporting both classification and free-form text generation tasks, offering a diverse and challenging testbed for evaluating AVR models.

**BP solution methods.** In response to the wide range of AVR challenges, various approaches have been developed (Hernández-Orallo et al., 2016; Małkiński & Mańdziuk, 2024a; Mitchell, 2021). Specifically, for BPs, methods include program synthesis and inductive logic programming (Saito & Nakano, 1996; Sonwane et al., 2021), cognitive architectures (Foundalis, 2006a), Bayesian inference (Depeweg et al., 2018; 2024), convolutional networks (Kharagorgiev, 2018), and meta-learning approaches such as prototypical networks (Snell et al., 2017), SNAIL (Mishra et al., 2018), or Meta-Baseline (Chen et al., 2021). Recently, LLM-based solutions combine image captioning models with LLMs for text descriptions (Wu et al., 2024) or leverage VLMs that handle both modalities (Małkiński et al., 2025; Wüst et al., 2025). To establish strong baselines, we evaluate state-of-the-art VLMs. Additionally, prior work has framed BPs in various settings, such as binary classification where test images are assigned to matrix sides (Nie et al., 2020; Kharagorgiev, 2018), or free-form text generation where the model articulates the underlying concept (Małkiński et al., 2025; Wu et al.,

Table 1: **BP datasets.** Bongard-RWR+ offers a broad set of matrices that uniquely combine generated real-world-like images with abstract concepts.

| Dataset | Images | Concepts | # Matrices | # Concepts |
|---|---|---|---|---|
| Synthetic BPs (Bongard, 1970) | Synthetic | Abstract | 394 | 388 |
| Bongard-LOGO (Nie et al., 2020) | Synthetic | Abstract | 12 000 | 627 |
| Bongard HOI (Jiang et al., 2022) | Real-world | Real-world | 53 000 | 242 |
| Bongard-OpenWorld (Wu et al., 2024) | Real-world | Real-world | 1 010 | 1 010 |
| Bongard-RWR (Małkiński et al., 2025) | Real-world | Abstract | 60 | 55 |
| **Bongard-RWR+ (ours)** | Generated | Abstract | 5 400 | 49 |

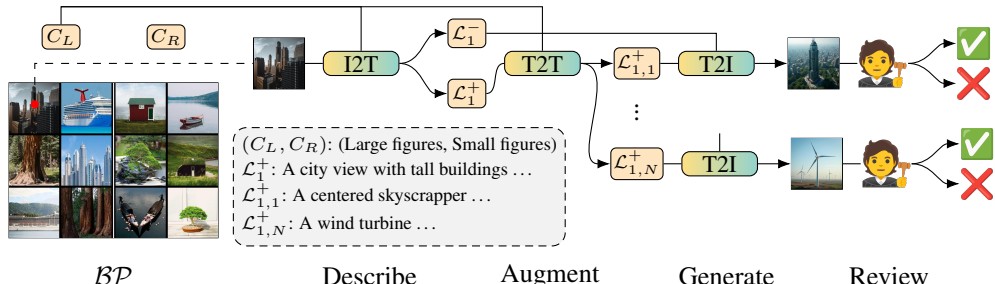

Figure 3: **Generative pipeline.** Starting from a Bongard problem $\mathcal{BP}$ with concept $(C_L, C_R)$, the pipeline: (1) describes each image using an I2T model to produce paired positive/negative captions $\mathcal{L}_i^+$ and $\mathcal{L}_i^-$; (2) augments each positive caption with a T2T model into $N$ diverse descriptions $\{\mathcal{L}_{i,j}^+\}_{j=1}^N$ that preserve the underlying concept; (3) generates candidate images for each new description using a T2I model; and (4) involves a human judge to review and filter the generated images. For readability, the figure illustrates the processing flow for the first image from the left matrix side.

2024). To systematically evaluate model capabilities, we cover diverse setups, including image (or description) to side classification, a multiclass concept selection task, and free-form text generation.

**Data generation with T2I models.** Generative T2I models have gained significant attention for their broad application. Prior work has applied T2I models to generate synthetic data for supervised learning (Fan et al., 2024; Ravuri & Vinyals, 2019; Sarıyıldız et al., 2023), contrastive representation learning (Jahanian et al., 2022; Tian et al., 2024; 2023), and data augmentation (Trabucco et al., 2024). In contrast, we employ a pre-trained T2I model to automatically generate real-world-like images depicting predefined abstract concepts, enabling scalable construction of Bongard-RWR+ instances.

## 3 METHODS

To advance research on abstract reasoning, we introduce Bongard-RWR+, a dataset comprising 5 400 BPs in its main setting, with additional instances in ablation variants. The dataset supports a range of task formulations, including binary and multiclass classification, and free-form text generation. Each instance is defined as $\mathcal{BP} = (L, R, T, C)$, where $L = \{L_1, \ldots, L_P\}$ and $R = \{R_1, \ldots, R_P\}$ denote the left and right panel sets (each with $P = 6$ images), $T = \{T_L, T_R\}$ contains two test images (one per side), and $C = (C_L, C_R)$ specifies the underlying concept in natural language, with $C_L/C_R$ describing the concept shared by all images on the left / right side, resp.

### 3.1 BONGARD-RWR+

To generate Bongard-RWR+, we considered each $\mathcal{BP}$ from Bongard-RWR. For each image $L_i \in L$ and $R_i \in R$, we generated paired positive and negative textual descriptions based on the side's concept: $\text{Describe}(L_i, C_L) = (\mathcal{L}_i^+, \mathcal{L}_i^-)$ and $\text{Describe}(R_i, C_R) = (\mathcal{R}_i^+, \mathcal{R}_i^-)$, using Pixtral-12B (MistralAI, 2024). This I2T model was prompted to produce positive descriptions that faithfully captured the image's content and negative descriptions designed to steer the T2I model away from depicting

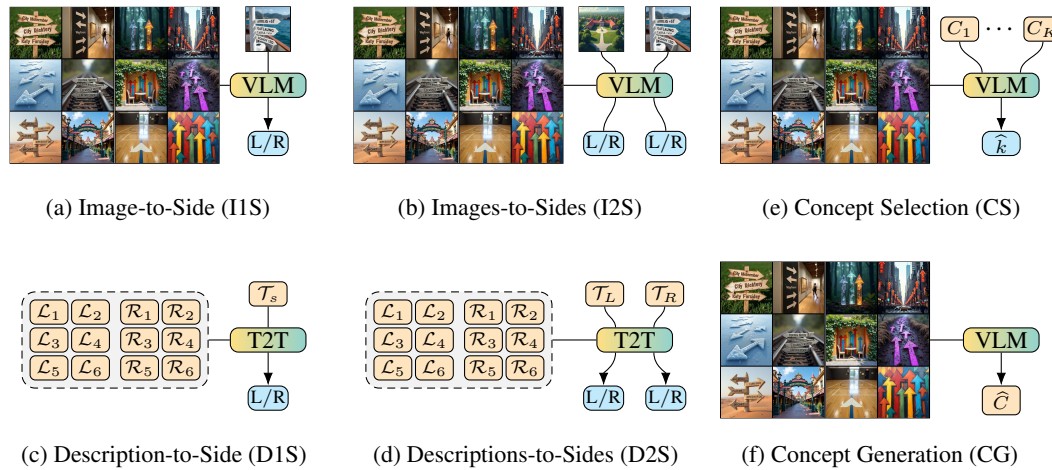

Figure 4: **BP formulations.** We define six tasks of variable complexity: (a; I1S) assign a single test image to the left or right side; (b; I2S) assign a pair of test images to the respective sides; (c; D1S / d; D2S) use descriptions from an I2T model and classify with a T2T model; (e; CS) select the correct concept index $\widehat{k}$ such that $C_{\widehat{k}} = C^*$; (f; CG) generate a natural language description of concept $\widehat{C}$.

the opposite concept. Each positive description was augmented with a Text-to-Text (T2T) model into $N = 15$ alternative descriptions reflecting the side's concept: $\text{Augment}(\mathcal{L}_i^+, C_L) = \{\mathcal{L}_{i,j}^+\}_{j=1}^N$ and $\text{Augment}(\mathcal{R}_i^+, C_R) = \{\mathcal{R}_{i,j}^+\}_{j=1}^N$. Each generated alternative description, paired with its corresponding negative prompt, was passed to the Flux.1-dev model (Labs, 2024) to render $512 \times 512$ candidate images: $\text{Render}(\mathcal{L}_{i,j}^+, \mathcal{L}_i^-) = L_{i,j}$ and $\text{Render}(\mathcal{R}_{i,j}^+, \mathcal{R}_i^-) = R_{i,j}$. Appendix F describes the employed model prompts. All generated images were manually reviewed to ensure they accurately reflected the intended concept ($C_L$ or $C_R$) without introducing elements from the opposite side's concept. Images failing this criterion were discarded. Appendix E.2 presents further details.

From the candidate images $\{L_{i,j}^+\}$ and $\{R_{i,j}^+\}$, we composed $M = 10$ left and right sides, denoted $\{L'\}_{m=1}^M$ and $\{R'\}_{n=1}^M$, by iteratively selecting subsets that maximized intra-set visual diversity. Each subset $L'_m \subset \{L_{i,j}^+\}$ and $R'_n \subset \{R_{i,j}^+\}$ contained 7 images (6 context images and 1 test image), chosen iteratively to minimize total pairwise cosine similarity of ViT-L/14 embeddings (Dosovitskiy et al., 2021). To cover more images and increase dataset diversity, in each iteration we selected a random image from each subset and removed it from the overall image pool in the next iterations. Finally, each left side $L'_m$ was paired with each right side $R'_n$ to construct $M^2 = 100$ new BP instances aligned with the original concept. Algorithm details are explained in Appendix B. Using this pipeline, we constructed $5\,400$ BPs corresponding to $54$ Bongard-RWR matrices (100 new problems per source BP). The 6 remaining Bongard-RWR matrices were discarded due to difficulties in generating a sufficient number of images faithfully depicting the underlying concepts.

We additionally introduce several dataset variants to support ablation studies. The first variant, Bongard-RWR+/GS, consists of grayscale matrices. Since Bongard-RWR+ concepts are primarily structural, as they originate from black-and-white synthetic BPs, this variant isolates the role of color, which is expected to be non-essential for concept detection. The second variant, Bongard-RWR+/LP ($P = 2, \dots, 6$), e.g., denoted Bongard-RWR+/L2 for $P = 2$, modifies the subset construction method by omitting the image removal step. In effect, it contains fewer unique images ($1\,503$ in Bongard-RWR+/L6 vs. $4\,157$ in Bongard-RWR+), but exhibits lower average intra-side embedding similarity. Lower similarity implies greater visual diversity within a matrix, meaning the same concept is illustrated through more varied content—for example, the concept "Vertical" may be instantiated as a tree, a building, or a standing figure—potentially making the concept easier to identify. In contrast, high similarity may reflect repeated visual aspects (e.g., several images of trees), which may hint at unrelated concepts like "Nature" or "Green", making the intended concept harder to capture. We consider Bongard-RWR+/LP variants with varying numbers of images per side ($P$), enabling analysis of how the number of demonstrations influences model performance. All variants are derived from the same image pool as the main dataset. Additional details are provided in Appendix B.

## 3.2 PROBLEM FORMULATIONS

We cast BP into several task formulations of increasing complexity, as illustrated in Fig. 4. We begin with binary classification settings. In the Image-to-Side (I1S) task, the model classifies a single test image ($T_L$ or $T_R$) as belonging to either the left or right side. The Images-to-Sides (I2S) variant extends this by requiring the model to assign a pair of test images ($T_L$ and $T_R$), each from a different class, to their respective sides. To assess the effect of an intermediate image captioning step, we introduce Description-to-Side (D1S) and Descriptions-to-Sides (D2S). These tasks first convert images into natural language descriptions using an I2T model: $\text{Describe}(L_i) = \mathcal{L}_i$, $\text{Describe}(R_i) = \mathcal{R}_i$, $\text{Describe}(T_s) = \mathcal{T}_s$, $\forall i \in \{1, \ldots, P\}, \forall s \in \{L, R\}$. The resulting descriptions are passed to a T2T model for a binary classification, either for a single test description (D1S) or a pair (D2S). In practice, I2T and T2T models may be implemented by a single VLM that processes both visual and textual inputs, though T2T can also be realized by text-only models (e.g., LLMs).

Beyond binary classification settings, we introduce the Concept Selection (CS) task, a multiclass classification setup where the model selects the correct concept $C^*$ describing the matrix from a candidate set of concepts $\{C_k\}_{k=1}^K$ ($C^* \in \{C_k\}$), with $K$ controlling task difficulty. To construct distractors ($\{C_k \mid C_k \neq C^*\}$), we sample concepts from other matrices, ensuring that each candidate represents a distinct concept to avoid ambiguity. Finally, we consider the most challenging formulation, Concept Generation (CG), where the model is required to generate a free-form textual description of the concept underlying the BP matrix.

## 4 EXPERIMENTS

We evaluate 4 state-of-the-art open-access VLMs on Bongard-RWR+: InternVL2.5 78B (IVL2.5) (Chen et al., 2024a;b), Qwen2-VL 72B-Instruct (Q2VL) (Bai et al., 2023; Wang et al., 2024), LLaVA-Next 110B (LLaVA) (Jiang et al., 2023; Liu et al., 2024a;b), and MiniCPM-o 2.6 8B (MCPM) (Yao et al., 2025; Yu et al., 2025). Main experiments use the largest available models, with smaller variants covered in ablations. All models are evaluated at a fixed decoding temperature of 0.5, using structured

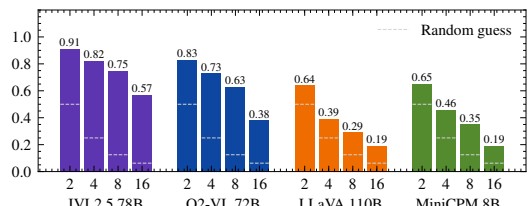

Figure 5: **Concept Selection.** Accuracy in the CS task on Bongard-RWR+ for $K \in \{2, 4, 8, 16\}$.

JSON output format enforced via the Outlines decoding backend (Willard & Louf, 2023). Inference was performed on an internal computing cluster equipped with NVIDIA DGX A100 and H100 nodes. The largest models required up to 24 hours of inference per task using 4 H100 GPUs.

We developed a Similarity Classifier (SC) to serve as a non-parametric baseline. For image-based setups (I1S and I2S), it computes ViT-L/14 embeddings (Dosovitskiy et al., 2021) for all matrix images and the test image. For text-based tasks (D1S and D2S), embeddings of image descriptions are obtained with fine-tuned MiniLM (HuggingFace, 2021; Wang et al., 2020a). Next, for each matrix side, the embedding that is most distant from the test embedding (based on Euclidean distance) is identified. The test input is then assigned to the side with the lower maximum distance (i.e., higher similarity). Appendix C.8 explains the reasons of SC strong performance.

**Concept selection.** We begin with the CS task, where the model selects the correct concept from a candidate set of size $K \in \{2, 4, 8, 16\}$. Increasing $K$ reduces accuracy, confirming the growing difficulty of the task (Fig. 5). InternVL2.5 performs best across all variants, correctly identifying the concept in 91% of cases for $K = 2$. However, its accuracy drops to 57% for $K = 16$, indicating that even the strongest model struggles when faced with multiple distractors. Meanwhile, both MiniCPM-o 2.6 and LLaVA-Next achieve only 19% for $K = 16$, underscoring their limited capacity even in this discriminative setting, and highlighting the utility of the CS task as a benchmark for AVR. Notably, MiniCPM-o 2.6 matches LLaVA-Next's performance despite a much smaller parameter count (8B vs. 110B), suggesting that size alone is insufficient to excel in the introduced challenge.

To better characterize model behavior, we categorized Bongard-RWR+ matrix concepts into 9 semantic groups reflecting the key factor behind each concept: Size, Position, Count, Branching, Similarity, Contour, Shape, Rotation, and Angle. This grouping follows recent efforts advocating

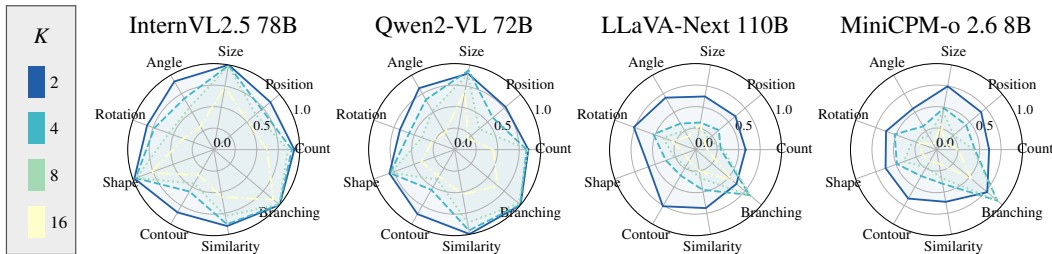

Figure 6: **Concept Selection.** Accuracy in the CS task on Bongard-RWR+ for $K \in \{2, 4, 8, 16\}$.

for semantically grounded evaluations to isolate model strengths and weaknesses (Mitchell, 2021; Odouard & Mitchell, 2022). The results are shown in Fig. 6. For $K = 2$, all models exhibit broadly similar profiles across concept groups. However, for more challenging setups ($K > 2$), differences emerge. For instance, InternVL2.5 retains high performance on Shape, Size, and Branching for $K = 16$, with accuracy near 75%. These categories involve high-level visual attributes that VLMs appear to capture reliably. In contrast, InternVL2.5 accuracy drops below 50% for Contour, Rotation, and Angle. These concepts often rely on subtle visual cues (Contour) or precise spatial relationships (Rotation and Angle), which current models struggle to capture reliably. We expand on this in Appendix C.10. Overall, performance patterns remain consistent across values of $K$, supporting the utility of such analysis as a diagnostic tool for characterizing model capabilities.

**Image(s)-to-side(s).** Next, we consider the I1S and I2S tasks, where models must assign one or two test images, resp., to the correct matrix side. Notably, in I2S (and D2S), the model performs two independent binary classifications; while the prompt specifies that the test inputs belong to different classes (matrix sides), this constraint is not enforced in the output format, causing some models to incorrectly assign both inputs to the same side. The results are shown in Table 2. For both tasks, VLMs performed at or near chance, with several models falling below random accuracy in I2S. Strikingly, the simple SC baseline outperformed all VLMs, suggesting that the tested models fail to robustly identify the underlying concept that separates semantically the matrix sides. Compared to the CS task, where models can rely on high-level heuristics to eliminate implausi-

Table 2: **\*-to-Side(s) classification.** Results with I1S, I2S, D1S, and D2S on Bongard-RWR+. Captions in D1S and D2S were produced with InternVL2.5 78B.

|       | I1S  | I2S  | D1S  | D2S  |
|-------|------|------|------|------|
| IVL2.5 | 0.50 | 0.39 | 0.57 | 0.49 |
| Q2VL  | 0.49 | 0.44 | **0.58** | 0.42 |
| LLaVA | 0.50 | 0.50 | 0.54 | 0.43 |
| MCPM  | 0.48 | 0.45 | 0.51 | 0.41 |
| DS-R1 | N/A  | N/A  | 0.57 | **0.56** |
| SC    | **0.52** | **0.54** | 0.49 | 0.50 |

ble options, I1S and I2S require a deeper understanding of the depicted concept to correctly classify new inputs. These results highlight a fundamental limitation in current VLMs' AVR capabilities.

**Description(s)-to-side(s).** Given the difficulty of the introduced dataset, we hypothesized that decomposing the problem via an intermediate captioning step may improve model performance. To test this, we generated image descriptions using each of the 4 VLMs and used these captions as input to the same set of models for solving the side prediction task. Since the prediction task is purely text-based, we also evaluated the DeepSeek-R1 70B (DS-R1) reasoning model (Guo et al., 2025), which does not support visual input ("N/A" for I1S and I2S in Table 2) but offers strong language understanding. As shown in Fig. 7, final accuracy in the D1S task varies depending on both the captioning (rows) and prediction (columns) models. Descriptions produced by InternVL2.5 yield the highest overall accuracy, with Qwen2-VL also producing relatively strong captions. Interestingly, captions from MiniCPM-o 2.6 lead to better results than those from LLaVA-Next, despite MiniCPM-o 2.6 being much

|          | IVL2.5 0.55 | Q2VL **0.56** | LLaVA 0.52 | MCPM 0.50 | DS-R1 0.55 |
|----------|------|------|------|------|------|
| **IVL2.5** 0.55 | 0.57 | 0.58 | 0.54 | 0.51 | 0.57 |
| **Q2VL** 0.54 | 0.56 | 0.57 | 0.53 | 0.50 | 0.55 |
| **LLaVA** 0.52 | 0.54 | 0.54 | 0.50 | 0.50 | 0.52 |
| **MCPM** 0.53 | 0.55 | 0.55 | 0.51 | 0.50 | 0.54 |

Figure 7: **Description-to-Side.** Accuracy in the D1S task on Bongard-RWR+ for a pair-wise combination of I2T description models (row-wise) and T2T prediction models (column-wise).

smaller, highlighting its promising image-to-text capabilities. Among prediction models (columns), Qwen2-VL achieves the best average performance, followed by InternVL2.5, indicating relatively strong text-based reasoning abilities of both models. DeepSeek-R1, while known for strong perfor-

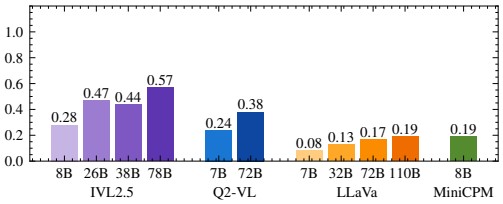

(a) Varying model size on Bongard-RWR+

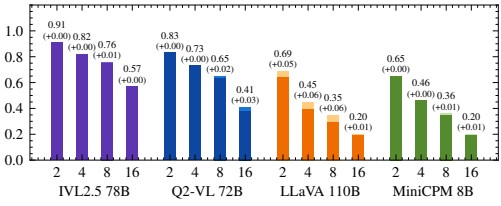

(b) Bongard-RWR+/GS

Figure 8: **Concept Selection sensitivity analysis.** (a) Accuracy of models of varying sizes for $K = 16$. (b) The impact of using grayscale images for $K \in \{2, 4, 8, 16\}$. Differences w.r.t. color images are shown using lighter colors and are annotated in parentheses above the corresponding bars.

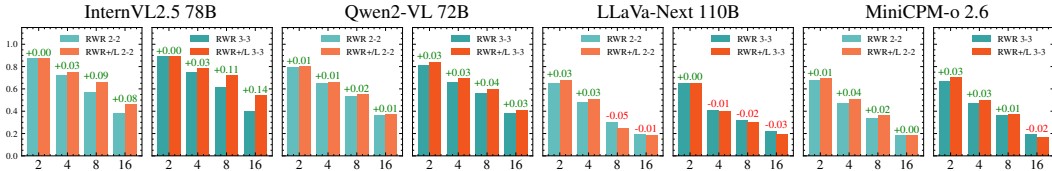

Figure 9: **Functional equivalence of generated images and real-world images.** Performance trends in the CS task ($K \in \{2, 4, 8, 16\}$) on Bongard-RWR and Bongard-RWR+/LP for $P = 2, 3$ panels.

mance on standard benchmarks, ranks third, revealing potential limitations in reasoning over image descriptions. Using captions generated by the best-performing model, InternVL2.5, we report D1S and D2S task performance in Table 2. Compared to I1S and I2S, most models perform above chance in these settings, confirming that the intermediate captioning step helps models better ground their predictions.

**Does AVR performance scale with model size?** Prior works have shown that VLM performance tends to scale with model size (Hoffmann et al., 2022; Kaplan et al., 2020). To examine whether this observation holds for AVR, we evaluated smaller model variants in the CS task on Bongard-RWR+. Specifically, we tested 8B, 26B, and 34B variants of InternVL2.5, 7B, 32B, and 72B versions of LLaVA-Next and 7B variant of Qwen2-VL. We excluded additional MiniCPM-o 2.6 variants, as the 8B model already showed weak performance and no larger variants are currently available. As shown in Fig. 8a, accuracy generally increases with model size. Among the smallest configurations (7B/8B), InternVL2.5 performs best, followed by Qwen2-VL, mirroring the relation observed for their largest versions (70+B). Notably, both InternVL2.5 8B and Qwen2-VL 7B outperformed MiniCPM-o 2.6 8B, underscoring the generally stronger AVR capacity of these two model families.

**Is image color necessary for concept recognition?** To assess the role of color in abstract reasoning, we evaluated all 4 VLMs on Bongard-RWR+/GS, using the CS task. As shown in Fig. 8b, model performance remains comparable, or even improves, on grayscale images. For instance, LLaVA-Next consistently achieves higher accuracy across all $K$ (e.g. +5 p.p. for $K = 2$ concepts). This outcome aligns with the dataset's design – the underlying concepts are derived from structural properties of black-and-white synthetic BPs, where color is not semantically relevant. In this context, color may act as a visual distractor that adds additional complexity, especially for lower-performing models.

**Are generated images as effective as real ones?** Since Bongard-RWR+ relies on generated images, a key question is whether such images are as effective as real ones for evaluating visual reasoning. To investigate this, we compared model performance on the introduced dataset with that on Bongard-RWR, which consists exclusively of real-world images. However, given that Bongard-RWR includes only 60 instances, we constructed matrices based on smaller subsets of images per side to expand the dataset. For each BP in Bongard-RWR, we considered all pairs of $P$-element subsets from both matrix sides (analogously as described in Section 3.1), yielding $\binom{6}{P}^2$ new matrices per BP, i.e., 36, 225, 400, 225, 36 matrices for $P = 1, \ldots, 5$, resp. We applied this method to 54 Bongard-RWR BPs that were used to generate Bongard-RWR+, and then selected $P = 2, 3$ to ensure sufficient dataset

Table 3: **I1S and D1S on Bongard-RWR+/LP.** Results in I1S and D1S tasks with varying numbers of images per side $P$. Image captions in the D1S task were produced with InternVL2.5 78B.

(a) Image-to-Side (I1S)

| $P$ | IVL2.5 | Q2VL | LLaVA | MCPM | SC |
|---|---|---|---|---|---|
| 2 | 0.51 | 0.51 | 0.49 | 0.50 | **0.54** |
| 3 | 0.54 | 0.54 | 0.51 | 0.50 | **0.59** |
| 4 | 0.54 | 0.54 | 0.51 | 0.51 | **0.59** |
| 5 | 0.54 | 0.55 | 0.51 | 0.52 | **0.58** |
| 6 | 0.55 | 0.57 | 0.51 | 0.51 | **0.59** |

(b) Description-to-Side (D1S)

| $P$ | IVL2.5 | Q2VL | LLaVA | MCPM | DS-R1 | SC |
|---|---|---|---|---|---|---|
| 2 | **0.56** | 0.55 | 0.53 | 0.50 | **0.56** | 0.47 |
| 3 | 0.60 | 0.58 | 0.56 | 0.51 | **0.62** | 0.51 |
| 4 | 0.61 | 0.63 | 0.58 | 0.54 | **0.64** | 0.51 |
| 5 | 0.62 | 0.60 | 0.57 | 0.52 | **0.64** | 0.52 |
| 6 | 0.67 | 0.67 | 0.61 | 0.53 | **0.70** | 0.54 |

size, and capped the number of sampled matrices per source BP at $100$, resulting in $5\,400$ matrices per $P$. We evaluated all $4$ VLMs in the CS task on these Bongard-RWR variants and compared the results to those on Bongard-RWR+/LP, which uses analogous matrix construction approach but with generated images. As shown in Fig. 9, model performance follows similar trends on both datasets – accuracy decreases consistently as $K$ increases, regardless of whether the images are real or generated. This supports the validity of our generation-based approach for capturing the challenges of AVR.

**Do models learn from demonstrations?** As $P$ increases, models are presented with more demonstrations of the underlying concept. While this could enhance concept identification, it also increases the amount of visual information to process. Interestingly, Fig. 9 shows that InternVL2.5 and Qwen2-VL benefit from larger $P$ in the CS task, e.g., InternVL2.5's accuracy improves by $2, 3, 6, 8$ p.p. for $K = 2, \ldots, 16$, resp. when comparing the results on Bongard-RWR+/L2 with Bongard-RWR+/L3. In contrast, the performance of LLaVA-Next and MiniCPM-o 2.6 shows no consistent improvement, suggesting these models may struggle to leverage additional examples.

To explore this further, we evaluated models on Bongard-RWR+/LP with $P = 2, \ldots, 6$ using I1S and D1S tasks (Table 3). In I1S, InternVL2.5 and Qwen2-VL show a mild upward trend with increasing $P$, with Qwen2-VL achieving the accuracy of $57\%$ at $P = 6$, and InternVL2.5 close behind at $55\%$. In D1S, accuracy generally improves with $P$, showing that the models can effectively utilize additional concept demonstrations in a text-based setting. Notably, DeepSeek-R1 outperforms others for every $P > 2$ (and is on-par with InternVL2.5 for $P = 2$), demonstrating its strength in this setting. Additionally, D1S scores are consistently higher than I1S scores, showing that the models benefit from the explicit image captioning step. Nevertheless, performance remains modest on both tasks, in I1S being clearly inferior to the SC baseline, pointing to a persistent AVR gap of the current models.

Finally, models' accuracy on Bongard-RWR+/LP exceeds that on Bongard-RWR+ (e.g., InternVL2.5 scores $50\%$ and $57\%$ in I1S and D1S, resp. on Bongard-RWR+ (cf. Table 2), but reaches $55\%$ and $67\%$ on Bongard-RWR+/L6). This gap stems from the differences in dataset construction, which we analyze in detail in Appendix C.

**Visual diversity facilitates concept recognition.** Concept diversity can be expressed through both a larger number of examples per side ($P$) and greater variation among the images themselves. To measure diversity in a complementary way, we computed the maximum cosine similarity between ViT-L/14 image embeddings within each side of a BP, then took the negative of the larger value across both sides as the

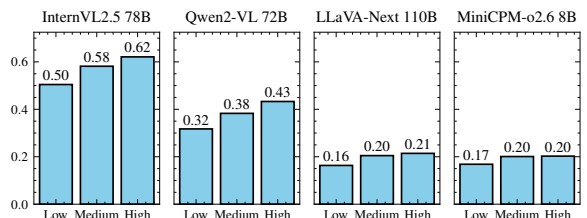

Figure 10: Accuracy vs. image diversity (CS, $K = 16$).

BP's diversity score. BPs were partitioned into three equally sized bins (Low, Medium, High diversity), and we analyzed model performance in the most challenging CS variant ($K = 16$).

As shown in Fig. 10, performance is lowest on Low-diversity matrices (e.g., InternVL2.5 78B scores $0.50$), where images are similar and depict the concept in similar ways, making it harder to identify. In contrast, performance is highest on High-diversity matrices (e.g., InternVL2.5 78B scores $0.62$,

+12 p.p. vs. Low-diversity), where the concept is expressed through more varied visual content, making it more apparent. This indicates that diversity helps disambiguate abstract concepts.

**Noisy images hinder concept recognition.** To quantify the impact of manual filtering on model performance, we conducted a targeted experiment introducing controlled noise into the dataset. Specifically, we constructed Bongard-RWR+/L2-Impure, a variant of Bongard-RWR+/L2 in which each BP side contains one image that passed human filtering and one image that was rejected. This yields BP instances where a subset of images violate the intended concept, producing ambiguity that would arise if human verification was removed.

We evaluated all four models on this impure dataset in the CS task (see details in Appendix C.5). Across $K \in \{2, 4, 8, 16\}$ levels, we observe mean absolute accuracy drops of $-3.85$, $-8.60$, $-5.73$, and $-4.60$ for MiniCPM-o 2.6 8B, InternVL2.5 78B, Qwen2-VL 72B, and LLaVA-Next 110B, resp. This consistent reduction in accuracy supports our hypothesis: including unfiltered images degrades dataset quality and harms model performance, demonstrating that the human review step meaningfully improves data quality.

**Concept generation.** We tested model ability to generate free-form concept descriptions in the CG task. Models were prompted to describe the concept behind each matrix, using either raw image inputs or InternVL2.5's captions. Responses were evaluated using standard NLP metrics: BLEU (Papineni et al., 2002), METEOR (Banerjee & Lavie, 2005), $ROUGE_L$ (Lin, 2004), CIDEr (Lin, 2004), and BERTScore (Zhang* et al., 2020). The results, reported in Appendix C, show that the models consistently achieved low scores on all metrics, with no clear best-performing model. Manual inspection of selected outputs further confirmed that current models generally struggle to articulate the BP concepts, highlighting this task as a particularly challenging direction for future research.

## 5 CONCLUSIONS

We introduced Bongard-RWR+, a large-scale dataset for evaluating AVR capabilities using generated real-world-like images of abstract concepts from classical BPs. The dataset supports diverse BP task formulations, including multiclass concept selection, binary side classification, and free-form text generation. Our results show that while current VLMs demonstrate certain ability to identify coarse-grained concepts, they consistently struggle with fine-grained concept recognition. Although explicit image captioning and increased visual demonstrations can improve performance, even the strongest models fall short in more demanding setups, notably unconstrained textual answer generation.

**Limitations.** Our dataset generation pipeline still requires human oversight to ensure adherence to intended concepts; advancing generative modeling is needed to improve scalability. Our evaluation includes selected VLMs; broader community involvement is essential to benchmark additional methods on Bongard-RWR+. We extend a discussion on limitations and their mitigation in Appendix D.

**Ethics statement.** We note that images generated by the T2I model can reflect social, cultural, or geographical biases. Although our data generation algorithm incorporates diversity safeguards, some biases may still be present in the dataset. To address this, we conducted a limited-scale bias audit, which we further discuss in Appendices D and E.1.

In addition, since our pipeline (Fig. 3) involves manual review, we recognize the potential for annotation-related bias. To mitigate this, each generated image was independently reviewed by two expert annotators, who verified whether the image faithfully expressed the intended concept. Inter-annotator agreement reached Cohen's $\kappa = 0.64$ (substantial agreement), with annotators agreeing on the same label in $77.3\%$ of cases. Disagreements were resolved through discussion, and images lacking consensus were discarded. Overall, $30.2\%$ of generated images were excluded through this filtering process. A more detailed description of the review process and an annotator agreement report, are provided in Appendix E.2.

All images in our dataset are generated with Flux.1-dev; none of them was taken from public image databases or libraries. The dataset was released under the CC BY 4.0 License, as this licensing is

compatible with the terms of the T2I model used for image generation[1]. We imposed an additional usage restriction by adding the following note: *"While Bongard-RWR+ is released under the CC BY 4.0 License, it is explicitly not intended for use in high-stakes human assessment contexts, including but not limited to standardized testing, educational admissions, employment screening, or cognitive evaluation"*.

**Reproducibility.** All datasets developed in this work are publicly available via the HuggingFace Datasets platform[2]. Code for dataset generation and experimental evaluation is available on GitHub[3] under the MIT License.

ACKNOWLEDGMENTS

This research was carried out with the support of the Laboratory of Bioinformatics and Computational Genomics and the High Performance Computing Center of the Faculty of Mathematics and Information Science Warsaw University of Technology.

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

## A    BROADER IMPACT

Our findings highlight that state-of-the-art VLMs continue to struggle with tasks that are relatively straightforward for humans. As public discourse increasingly focuses on the capabilities and risks of AI systems, benchmarks such as Bongard-RWR+ play a crucial role in dispelling misconceptions about current model capabilities. Our release of code and datasets under an open license promotes open research and lowers the barrier of entry for researchers to evaluate and improve their methods. However, we also acknowledge potential risks. An effective AVR solver could potentially be misused in online or remote IQ assessments, allowing individuals to inflate their scores. Such misuse may lead to unfair advantages in job recruitment or other competitive settings where cognitive ability plays a role. This underscores the importance of raising public awareness about both the capabilities and potential applications of such tools.

Prior work in cognitive psychology shows that cultural background can indeed shape reasoning strategies and outcomes (Helms-Lorenz et al., 2003), sometimes leading to qualitatively different solution approaches (Nisbett et al., 2008). In particular, studies on RPMs discussed the variation between ethnic groups within the United States (Raven, 2005). While little work has examined BPs in this context, related studies in concept learning and categorization suggest that cross-cultural differences can in fact influence how abstract visual concepts are formed (Rosch, 1973). Building on this line of work, we believe that our Bongard-RWR+ dataset, which uses real-world images rather than abstract patterns, can serve as a complementary resource for studying how multi-image reasoning generalizes across different cultures.

## B    DATASET VARIANTS

---

**Algorithm 1** The generation of Bongard-RWR+ images.

---

1: **Input:** A set of Bongard-RWR problems $\text{BP}_k \in \mathcal{RWR}$
2: **Output:** Generated images $G = (GL, GR)$

3: $N \leftarrow 15$
4:
5: **for** $(L_k, R_k, T_k, C_k) \in \mathcal{RWR}$ **do**
6: $\quad L_k \leftarrow L_k \cup T_{k,L}$
7: $\quad R_k \leftarrow R_k \cup T_{k,R}$
8: $\quad$ **for** $S_k \in \{L_k, R_k\}$ **do**
9: $\quad\quad$ **for** $S_{k,i} \in S_k$ **do**
10: $\quad\quad\quad (\mathcal{S}_{k,i}^+, \mathcal{S}_{k,i}^-) \leftarrow \text{Describe}(S_{k,i}, C_{S_k})$
11: $\quad\quad\quad \{\mathcal{S}_{k,i,j}^+\}_{j=1}^N \leftarrow \text{Augment}(\mathcal{S}_{k,i}^+, C_{S_k})$
12:
13: $\quad\quad\quad$ **for** $j = 1, ..., N$ **do**
14: $\quad\quad\quad\quad GS_{k,i,j} \leftarrow \text{Render}(\mathcal{S}_{k,i,j}^+, \mathcal{S}_{k,i}^-)$
15: $\quad\quad\quad$ **end for**
16: $\quad\quad$ **end for**
17: $\quad$ **end for**
18: **end for**

---

**Image generation.**    Algorithm 1 outlines the process of generating Bongard-RWR+ images. Each $\text{BP}_k$ $(L_k, R_k, T_k, C_k)$ from Bongard-RWR is processed by iterating over both matrix sides, $L_k$ and $R_k$. Test images $T_k$ are processed based on their corresponding side. For each image $S_{k,i}$ on a given side $S_k$ (either $L_k$ or $R_k$), a positive ($\mathcal{S}_{k,i}^+$) and a negative ($\mathcal{S}_{k,i}^-$) textual description are generated based on the side's concept $C_{S_k}$ (either $C_{L_k}$ or $C_{R_k}$). The positive description is then augmented $N = 15$ times to produce a diverse set of semantically consistent variations $\mathcal{S}_{k,i,j}^+, j = 1, \ldots, N$. Each augmented description is paired with the corresponding negative description and passed to a T2I model to generate a new image $GS_{k,i,j}$ (either $GL_{k,i,j}$ or $GR_{k,i,j}$). All generated images were reviewed manually by a human expert, and those that did not faithfully reflect the intended concept were discarded.

---

**Algorithm 2** Bongard-RWR+ matrix construction.

---

1: **Input:** Generated images $G = (GL, GR)$, Boolean image removal flag $F$, Number of images per side $P$
2: **Output:** Set of generated matrices $\mathcal{RWR}^+$

3: $M \leftarrow 10$
4: $\mathcal{RWR}^+ \leftarrow \emptyset$
5:
6: **for** $GL_k, GR_k \in G$ **do**
7:     **for** $GS_k \in \{GL_k, GR_k\}$ **do**
8:         $E_k \leftarrow \text{GetImageEmbedings}(GS_k)$
9:         $GS_k^* \leftarrow \emptyset$
10:         $\{\mathbf{GS_{k,1}}, \mathbf{E_{k,1}}\} \leftarrow \text{GetAllSubsets}(GS_k, E_k, P)$
11:         **for** $m = 1, \ldots, M$ **do**
12:           // Find the most diverse subset
13:           $\ell \leftarrow \arg\min_{l=1,\ldots,|\mathbf{E_k}|} \max\{\mathbf{E_{k,l,i}} \cdot \mathbf{E_{k,l,j}} \mid i, j = 1, \ldots, P \ \wedge \ i \neq j\}$
14:           $GS_k^* \leftarrow GS_k^* \cup \mathbf{GS_{k,\ell}}$
15:
16:           **if** $F$ **then**
17:             // Find the most redundant image and remove it from the overall image pool
18:             $p \leftarrow \arg\max_{i=1,\ldots,P}\{\mathbf{E_{k,\ell,i}} \cdot \mathbf{E_{k,\ell,j}} \mid j = 1, \ldots, P \ \wedge \ i \neq j\}$
19:             $GS_k \leftarrow GS_k - \{\mathbf{GS_{k,\ell,p}}\}$
20:             $E_k \leftarrow E_k - \{\mathbf{E_{k,\ell,p}}\}$
21:             $\{\mathbf{GS_k}, \mathbf{E_k}\} \leftarrow \text{RemoveSubsetsContainingImage}(\mathbf{GS_k}, \mathbf{E_k}, \mathbf{GS_{k,\ell,p}})$
22:             // Exit early if there is not enough images to construct the next subset
23:             **if** $|GS_k| < P$ **then**
24:               break
25:             **end if**
26:             // Exit early if the images become too similar
27:             $s \leftarrow \text{Mean}(\{\frac{E_{k,i} \cdot E_{k,j}}{\|E_{k,i}\|\|E_{k,j}\|} \mid i, j = 1, \ldots, |E_k| \ \wedge \ i \neq j\})$
28:             **if** $s \geq 0.85$ **then**
29:               break
30:             **end if**
31:           **end if**
32:         **end for**
33:     **end for**
34:
35:     $\mathcal{RWR}^+ \leftarrow \mathcal{RWR}^+ \cup (GL_k^* \times GR_k^*)$
36: **end for**

---

**Bongard-RWR+ construction.** Algorithm 2 outlines the construction of Bongard-RWR+ matrices from a pool of generated images $G = (GL, GR)$. For each $BP_k$, both sides $GL_k$ and $GR_k$, denoted as $GS_k$, are processed independently, to construct $M = 10$ visually diverse subsets per side $GS_k^*$. Each side's image pool $GS_k$ is first embedded using ViT-L/14 (Dosovitskiy et al., 2021), producing embeddings $E_k$. The algorithm arranges images into $P$-element subsets, denoted $\mathbf{GS_{k,1}}$, along with their embeddings $\mathbf{E_{k,1}}$. In each of the $M$ iterations, the algorithm selects the most visually diverse subset, defined as the subset whose maximum pairwise cosine similarity among its embeddings is minimal. This selected subset $\mathbf{GS_{k,\ell}}$ is then added to $GS_k^*$. If the image removal flag $F$ is enabled, the algorithm identifies the most redundant image in the selected subset—i.e., the one with the highest average similarity to the other images—and removes it from the overall image pool $GS_k$. The loop terminates early if either (1) the remaining pool contains fewer than $P$ images or (2) the overall average similarity among remaining images exceeds a threshold, indicating reduced diversity. Finally, the cartesian product $GL_k^* \times GR_k^*$ is computed to generate all possible pairings and added to the output set $\mathcal{RWR}^+$. Examples of resultant BPs are illustrated in Fig. 22.

**Bongard-RWR+/GS construction.** We constructed a grayscale variant of Bongard-RWR+, denoted Bongard-RWR+/GS, by converting all images in the original set to grayscale. The overall matrix construction procedure remained unchanged. Representative examples are shown in Fig. 23.

**Bongard-RWR+/LP construction.** We also constructed Bongard-RWR+/LP, a dataset variant in which the image removal step is disabled (i.e., the flag $F$ in Algorithm 2 is set to false). Without this removal step, the greedy subset selection procedure for building $GS_k^*$ often draws from a smaller set of unique images within $GS_k$. In effect, matrices in Bongard-RWR+/LP exhibit lower average intra-side embedding similarity but are composed of fewer distinct images. The increased visual diversity within each matrix can facilitate recognition by presenting concepts in a broader range of views. However, because the same images are reused more frequently across matrices, there is overall less image-level diversity compared to Bongard-RWR+. We constructed dataset variants with $P = 2, \ldots, 6$, denoted Bongard-RWR+/L2, Bongard-RWR+/L3, etc. Example BPs from Bongard-RWR+/LP are shown in Figs. 24 – 28.

**Construction of Bongard-RWR+/TVT and Bongard-RWR+/TVT-Large.** We also developed Bongard-RWR+/TVT, a dataset variant in which BPs are divided into train, validation, and test splits (TVT), enabling the evaluation of supervised approaches. To construct these splits, we first consider the six images on each side of a Bongard-RWR matrix, and the corresponding seventh test image. Images 1 – 4, along with the side's test image, are grouped into a shared pool used to construct BP contexts for all three splits and to define test targets for the training split. Images 5 and 6 are exclusively allocated as test images for the validation and test splits, resp. This partitioning strategy ensures relatively high image diversity in the training set while enabling evaluation on out-of-distribution images in the validation and test splits. Bongard-RWR+/TVT comprises $12\,150$ matrices ($7\,290 / 1\,215 / 3\,645$ in train / validation / test splits); we additionally provide an extended variant, Bongard-RWR+/TVT-Large, consisting of $86\,400$ BPs ($51\,840 / 8\,640 / 25\,920$ in train / validation / test splits).

## C  EXTENDED RESULTS

### C.1  IS IMAGE COLOR NECESSARY FOR CONCEPT RECOGNITION?

To extend the grayscale image analysis from the CS task, we evaluated model performance on Bongard-RWR+/GS with the I1S and I2S tasks. As shown in Fig. 11, grayscale images led to improved accuracy in certain cases, likely by encouraging the model to focus on structural cues. Nevertheless, performance remained close to the random guess level across both tasks, reiterating their inherent difficulty.

### C.2  ARE GENERATED IMAGES AS EFFECTIVE AS REAL ONES?

To quantify the relationship presented in the paper (Fig. 9) more rigorously, we conducted a statistical analysis across all four models for both counts of panels per side ($P = 2, 3$) and all difficulty levels ($K = 2, 4, 8, 16$). We computed Pearson correlations and fitted linear regression models to assess

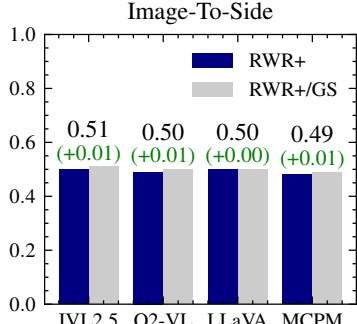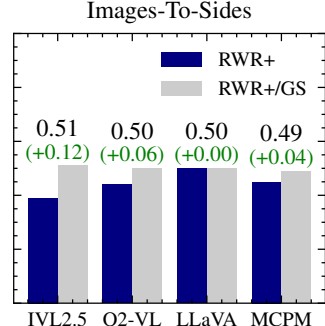

Figure 11: **Solving Bongard-RWR+/GS with I1S and I2S.** The impact of using grayscale images on accuracy in the I1S (left) and I2S (right) tasks.

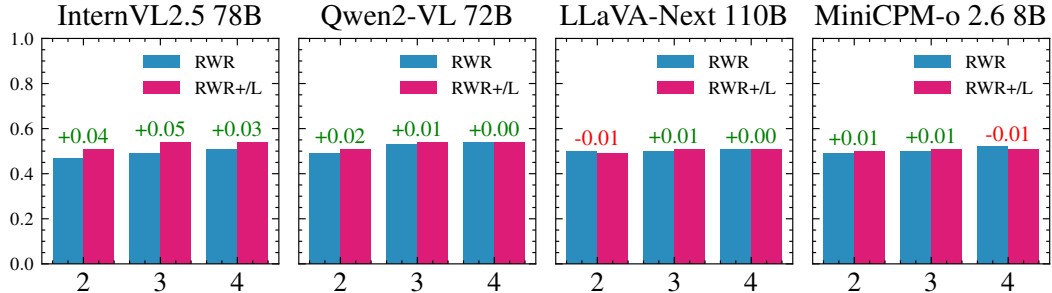

Figure 12: **Functional equivalence of generated images and real-world images.** Performance trends in the I1S task on Bongard-RWR and Bongard-RWR+/LP for a varying number of images per side $P = 2, 3, 4$.

how well performance on the generated-image dataset (Bongard-RWR+/LP) predicts performance on the real-image dataset (Bongard-RWR). The results are summarized in Table 4.

All models exhibit strong correlations between real-image and generated-image dataset performance ($r > 0.99$), with $R^2$ consistently above $0.987$. For example, InternVL2.5 78B for $P = 2$ shows $r = 0.994$ and a regression slope of $1.204$, meaning that, e.g., a 10 p.p. gain on Bongard-RWR+/L2 corresponds to roughly a 12 p.p. gain on Bongard-RWR. Error measures including mean absolute difference (MAD $\leq 7.0$) and RMSE ($\leq 2.09$) confirm that Bongard-RWR+/LP is a slightly easier set, on average, but the relationship is stable and predictable across difficulty levels.

These results demonstrate that the generated-image dataset preserves the underlying difficulty of the real-image dataset. Consequently, performance improvements on Bongard-RWR+ can be expected to translate reliably to improvements on real-image BPs, supporting the validity of using generated images for AVR evaluation.

Furthermore, we evaluated model performance on Bongard-RWR and Bongard-RWR+/LP in the I1S task for a varying number of images per side $P = 2, 3, 4$. As shown in Fig. 12, results are comparable across both datasets, regardless of whether the model processes real-world or generated images.

To further analyze the types of errors models make on real vs. generated images, we compared per-concept performance on Bongard-RWR (real) and Bongard-RWR+/LP for $P = 3$. Across the 54 concepts present in both datasets, performance was largely similar: in 40 concepts (74%) the difference was $\leq 10$ p.p., in 7 concepts (13%) differences were within $10 - 20$ p.p., and for 7 concepts (13%) differences exceeded 20 p.p. Notably, in all larger-difference cases, models performed better on average on Bongard-RWR+/LP than on Bongard-RWR. In the small-difference cases, models performed better on Bongard-RWR for 5 concepts and on Bongard-RWR+/LP for 2 concepts. Overall, the results indicate that despite certain variation, the two datasets present a comparable level of difficulty.

Table 4: **Performance correlation on generated and real-world images.** Pearson correlation ($r$), coefficient of determination ($R^2$), regression parameters, mean absolute difference (MAD) and RMSE across $K \in \{2, 4, 8, 16\}$ on Bongard-RWR and Bongard-RWR+/LP for $P \in \{2, 3\}$.

| Model | $P$ | $r$ | $R^2$ | Slope | Intercept | MAD | RMSE |
|---|---|---|---|---|---|---|---|
| MiniCPM-o 2.6 8B | 2 | 0.997 | 0.994 | 0.971 | −0.005 | 1.75 | 1.38 |
| MiniCPM-o 2.6 8B | 3 | 0.999 | 0.998 | 0.902 | 0.030 | 2.25 | 0.80 |
| InternVL2.5 78B | 2 | 0.994 | 0.987 | 1.204 | −0.190 | 5.00 | 2.04 |
| InternVL2.5 78B | 3 | 0.993 | 0.987 | 1.419 | −0.377 | 7.00 | 2.09 |
| Qwen2-VL 72B | 2 | 1.000 | 0.999 | 1.005 | −0.015 | 1.25 | 0.43 |
| Qwen2-VL 72B | 3 | 1.000 | 0.999 | 1.004 | −0.035 | 3.25 | 0.43 |
| LLaVA-Next 110B | 2 | 0.994 | 0.987 | 0.867 | 0.054 | 3.00 | 1.98 |
| LLaVA-Next 110B | 3 | 1.000 | 1.000 | 0.936 | 0.040 | 1.50 | 0.26 |

Table 6: **Concept Generation: Image-based on Bongard-RWR+.**

| | BLEU$_1$ | BLEU$_2$ | METEOR | ROUGE$_L$ | CIDEr | $P_{\text{BERT}}$ | $R_{\text{BERT}}$ | $F_{\text{BERT}}$ |
|---|---|---|---|---|---|---|---|---|
| InternVL2.5 78B | 0.063 | **0.011** | 0.056 | 0.159 | **0.017** | 0.114 | −0.049 | 0.027 |
| Qwen2-VL 72B | 0.048 | 0.004 | 0.043 | 0.121 | 0.007 | 0.066 | −0.102 | −0.025 |
| LLaVA-Next 110B | 0.071 | 0.008 | 0.061 | 0.174 | 0.006 | **0.128** | −0.037 | 0.040 |
| MiniCPM-o 2.6 8B | **0.077** | 0.008 | **0.063** | **0.181** | 0.004 | 0.127 | **-0.035** | **0.041** |

These findings are consistent with the experiment on the CS task presented in the main paper and reinforces our claim that generated images are equally effective as real-world images for probing visual reasoning capabilities of VLMs.

## C.3 DO MODELS LEARN FROM DEMONSTRATIONS?

We analyzed how the number of images per side ($P$), representing concept demonstrations, affects model performance. Extending the I1S and D1S analysis in the main paper, we evaluated model accuracy on Bongard-RWR+/LP using the CS task across varying values of $P$. As shown in Table 5, performance generally improves with more demonstrations, particularly for stronger models such as InternVL2.5, which achieves its best result at $P = 6$. However, not all models exhibit this trend – Qwen2-VL performs similarly for $P = 3$ and $P = 6$, while MiniCPM-o 2.6 achieves its peak

Table 5: Concept Selection on Bongard-RWR+/LP for $K = 16$ concepts.

| $P$ | IVL2.5 | Q2-VL | LLaVA | MCPM |
|---|---|---|---|---|
| 2 | 0.46 | 0.37 | 0.18 | 0.18 |
| 3 | 0.54 | 0.41 | 0.19 | 0.17 |
| 4 | 0.46 | 0.39 | 0.19 | 0.18 |
| 5 | 0.58 | 0.39 | 0.19 | 0.19 |
| 6 | 0.62 | 0.41 | 0.20 | 0.18 |

performance for $P = 5$. These results suggest that while additional demonstrations can be beneficial, some models may not be able to fully exploit them.

## C.4 ADDITIONAL BASELINES

We evaluated the performance of supervised learning methods on the I1S and D1S tasks. The models operated on embeddings produced by pre-trained models. For I1S, raw images were encoded using ViT-L/14 (Dosovitskiy et al., 2021), while for D1S, image captions produced by InternVL2.5 were embedded using fine-tuned MiniLM (HuggingFace, 2021; Wang et al., 2020a). We considered an MLP with a single hidden layer applied to concatenated embeddings, Wild Relation Network (WReN) (Barrett et al., 2018), which processes embedding pairs and aggregates them via summation, and SNAIL (Mishra et al., 2018), an attention-based meta-learner. As shown in Table 12, all models performed near the random guess level on both Bongard-RWR+/TVT and Bongard-RWR+/TVT-Large, demonstrating that the proposed BPs present a significant challenge not only for VLMs but also for standard supervised learning approaches.

Table 7: **Concept Generation: Image-based on Bongard-RWR+/L6.**

| | BLEU$_1$ | BLEU$_2$ | METEOR | ROUGE$_L$ | CIDEr | $P_{\text{BERT}}$ | $R_{\text{BERT}}$ | $F_{\text{BERT}}$ |
|---|---|---|---|---|---|---|---|---|
| InternVL2.5 78B | 0.060 | **0.011** | 0.053 | 0.149 | **0.021** | 0.111 | $-0.045$ | 0.028 |
| Qwen2-VL 72B | 0.042 | 0.005 | 0.039 | 0.110 | 0.008 | 0.046 | $-0.118$ | $-0.042$ |
| LLaVA-Next 110B | 0.072 | 0.007 | 0.061 | 0.172 | 0.008 | 0.133 | $-0.031$ | 0.045 |
| MiniCPM-o 2.6 8B | **0.076** | 0.009 | **0.064** | **0.182** | 0.005 | **0.144** | **-0.024** | **0.054** |

Table 8: **Concept Generation: Text-based on Bongard-RWR+.**

| | BLEU$_1$ | BLEU$_2$ | METEOR | ROUGE$_L$ | CIDEr | $P_{\text{BERT}}$ | $R_{\text{BERT}}$ | $F_{\text{BERT}}$ |
|---|---|---|---|---|---|---|---|---|
| InternVL2.5 78B | 0.122 | 0.021 | 0.063 | 0.158 | 0.017 | 0.142 | 0.073 | 0.107 |
| Qwen2-VL 72B | **0.132** | **0.021** | **0.066** | 0.161 | 0.018 | 0.145 | **0.090** | **0.117** |
| LLaVA-Next 110B | 0.111 | 0.018 | 0.064 | 0.165 | 0.013 | **0.147** | 0.062 | 0.103 |
| MiniCPM-o 2.6 8B | 0.105 | 0.019 | 0.064 | **0.170** | **0.019** | 0.141 | 0.033 | 0.085 |
| DeepSeek-R1 70B | 0.129 | 0.013 | 0.060 | 0.138 | 0.015 | 0.069 | 0.050 | 0.060 |

## C.5 NOISY IMAGES HINDER CONCEPT RECOGNITION

Fig. 13 compares model performance between Bongard-RWR+/L2 and Bongard-RWR+/L2-Impure in the CS task across $K$ levels. Substituting an accepted image with a rejected one consistently reduces performance across models, with only a single case of comparable results (LLaVA-Next, $K = 8$). The effect is strongest for InternVL2.5 (mean drop of $8.6$ p.p.) and Qwen2-VL ($5.7$ p.p.), moderate for LLaVA-Next ($4.6$ p. p.), and smallest for MiniCPM-o 2.6 ($3.9$ p.p.).

## C.6 PERFORMANCE ON CONCEPT GENERATION TASK

Tables 6 – 9 evaluate model performance on the CG task, where models are prompted to produce free-form descriptions of the underlying concept for each matrix. Inputs were either raw images or image captions produced by InternVL2.5. Experiments were conducted on both Bongard-RWR+ and Bongard-RWR+/L6 datasets. We assess generated descriptions using standard NLP metrics: BLEU (BLEU$_1$ and BLEU$_2$) (Papineni et al., 2002), METEOR (Banerjee & Lavie, 2005), ROUGE$_L$ (Lin, 2004), CIDEr (Lin, 2004), and BERTScore (precision $P_{\text{BERT}}$, recall $R_{\text{BERT}}$, F1 $F_{\text{BERT}}$) (Zhang* et al., 2020). For BERTScore, we apply baseline rescaling[4] and the DeBERTa[5] model (He et al., 2021), which has strong alignment with human evaluations. Higher scores indicate better performance across all metrics.

MiniCPM-o 2.6 consistently performs best in image-based setups (Tables 6 – 7), ranking first in 5 of 8 metrics on Bongard-RWR+ and 6 of 8 on Bongard-RWR+/L6, though differences with other models are small. In text-based setups (Tables 8 – 9), Qwen2-VL leads in 5 of 8 metrics on Bongard-RWR+ and 4 of 8 on Bongard-RWR+/L6, again with relatively small gaps to competing models. We further compare top-performing models on Bongard-RWR+ with the best results reported for Bongard-OpenWorld (Wu et al., 2024, Appendix E, Table 4), as shown in Tables 10 and 11. Regardless of input representation, models achieve significantly higher scores on Bongard-OpenWorld than on Bongard-RWR+, indicating that the concepts in our dataset are more challenging to recognize even for state-of-the-art models.

## C.7 ASSESSING DIFFICULTY OF BONGARD-RWR+ VARIANTS

We compared model performance on a range of tasks using both Bongard-RWR+ and Bongard-RWR+/L6, with the results shown in Table 13. Overall, the models consistently perform better on Bongard-RWR+/L6 across both image- and text-based strategies. In particular, DeepSeek-R1 presents notable gains in text-based setups, achieving 70% accuracy with D1S and 72% using

---

[4] `https://github.com/Tiiiger/bert_score/blob/master/journal/rescale_baseline.md`

[5] `https://huggingface.co/microsoft/deberta-xlarge-mnli`

Table 9: **Concept Generation: Text-based on Bongard-RWR+/L6.**

|  | BLEU$_1$ | BLEU$_2$ | METEOR | ROUGE$_L$ | CIDEr | $P_{\text{BERT}}$ | $R_{\text{BERT}}$ | $F_{\text{BERT}}$ |
|---|---|---|---|---|---|---|---|---|
| InternVL2.5 78B | 0.123 | 0.023 | 0.065 | 0.162 | 0.019 | **0.154** | 0.087 | 0.120 |
| Qwen2-VL 72B | **0.132** | 0.023 | **0.066** | 0.160 | 0.018 | 0.153 | **0.095** | **0.124** |
| LLaVA-Next 110B | 0.114 | 0.020 | 0.064 | 0.164 | 0.012 | 0.149 | 0.070 | 0.108 |
| MiniCPM-o 2.6 8B | 0.110 | **0.024** | 0.066 | **0.169** | **0.022** | 0.146 | 0.046 | 0.094 |
| DeepSeek-R1 70B | 0.128 | 0.015 | 0.060 | 0.134 | 0.018 | 0.072 | 0.052 | 0.062 |

Table 10: **Concept Generation: Image-based on Bongard-RWR+ vs. Bongard-OpenWorld.** Comparison of concept descriptions generated from images in Bongard-RWR+ (using MiniCPM-o 2.6 8B) and Bongard-OpenWorld (using GPT-4V (Wu et al., 2024, Appendix E, Table 4)).

|  | BLEU$_1$ | BLEU$_2$ | METEOR | ROUGE$_L$ | CIDEr |
|---|---|---|---|---|---|
| Bongard-RWR+ | 0.077 | 0.008 | 0.063 | 0.181 | 0.004 |
| Bongard-OpenWorld | 0.190 | 0.073 | 0.111 | 0.188 | 0.527 |

D2S. We attribute these improvements to the greedy matrix construction strategy used in Bongard-RWR+/L6, which prioritizes subsets that minimize intra-side embedding similarity. This encourages the selection of more visually diverse images to represent a concept, making the underlying shared aspects more distinguishable and thus easier to recognize for the models.

## C.8 Performance of the Similarity Classifier

The Similarity Classifier achieves high results in I1S experiments, as it exploits a fundamental property of BPs: a defining concept must hold for all images on a given side. This motivates the use of a max-aggregation heuristic, which assigns the test input to the side for which the maximum distance between individual image embeddings and the test image embedding (most outlying pair) is smaller. Fig. 15 illustrates the strength of the underlying design concept of SC.

## C.9 Qualitative analysis

To understand how concept difficulty varies across input modalities, we analyzed the average number of correctly solved instances per concept for all models. Fig. 16 presents the distribution of concept difficulty for both I1S and D1S tasks.

For I1S, each concept was solved in $79 - 109$ instances (out of 200) on average. This relatively narrow range indicates that most concepts remain uniformly challenging for models in the visual domain, which is consistent with their near-random performance levels. One noteworthy case is "Shading thicker on the right side" (concept #63), which appears among top-5 solved concepts in I1S (109) but is relatively difficult in D1S (92), highlighting the relevance of visual input in identifying fine-grained concepts.

In contrast, D1S exhibits a much broader difficulty range, from 70 to 182 solved instances, reflecting high performance on certain concepts that appear to be easier. Notably, the top-5 solved concepts include counting-based ones such as "three parts" or "one figure", which may be easier to capture in the text domain.

Tables 14 and 15 present the top 5 correctly recognized concepts per model for the I1S and D1S tasks, resp. In the image-based setup, we observe a diverse distribution of top concepts across the models, with little overlap. This variation suggests that each model relies on distinct reasoning strategies when processing visual input. For example, InternVL2.5 shows some capacity for identifying rules involving contour, similarity, and shape, whereas Qwen2-VL performs better on concepts related to count and angle. In contrast, the text-based setup yields more consistent results across models. Counting-related concepts dominate the top-performing examples, followed by shape and size.

The contrast between I1S and D1S results highlights modality-specific strengths, as the models tend to prefer different concept groups depending on whether they are reasoning over images or text.

Table 11: **Concept Generation: Text-based on Bongard-RWR+ vs. Bongard-OpenWorld.** Comparison of concept descriptions generated from image captions in Bongard-RWR+ (using Qwen2-VL 72B for prediction and InternVL2.5 78B for captioning) and Bongard-OpenWorld (using ChatGPT for prediction and BLIP-2 w/ Fine-tuning for captioning (Wu et al., 2024, Appendix E, Table 4)).

|  | $BLEU_1$ | $BLEU_2$ | METEOR | $ROUGE_L$ | CIDEr |
|---|---|---|---|---|---|
| Bongard-RWR+ | 0.132 | 0.021 | 0.066 | 0.161 | 0.018 |
| Bongard-OpenWorld | 0.441 | 0.292 | 0.222 | 0.417 | 1.714 |

Table 12: **Baseline performance.** Test accuracy of supervised learning models on Bongard-RWR+/TVT and Bongard-RWR+/TVT-Large, using image and text-based input representations. Each experiment was repeated 10 times with different seeds; the table reports mean $\pm$ std.

|  | Bongard-RWR+/TVT | | Bongard-RWR+/TVT-Large | |
|---|---|---|---|---|
|  | Image | Text | Image | Text |
| MLP | $0.49 \pm 0.00$ | $0.49 \pm 0.00$ | $0.47 \pm 0.01$ | $0.49 \pm 0.00$ |
| WReN | $0.49 \pm 0.01$ | $0.48 \pm 0.01$ | $0.48 \pm 0.01$ | $0.48 \pm 0.01$ |
| SNAIL | $0.48 \pm 0.01$ | $0.51 \pm 0.01$ | $0.48 \pm 0.01$ | $0.48 \pm 0.01$ |

These trends are further illustrated in Fig. 14, which shows average accuracy per concept group. Most models, InternVL2.5 and Qwen2-VL in particular, exhibit performance spikes in size, count, and shape groups with the D1S task compared to I1S. This discrepancy indicates a weak connection between visual and textual processing pathways in multimodal systems and suggests that improving the integration between these modalities could further boost model performance via knowledge transfer.

### C.10 SOURCES OF ERROR AND FAILURE MODES

In Section 4 (Concept Selection) we noted that concepts such as Contour, Rotation, and Angle pose particular difficulties because they rely on subtle geometric cues or precise spatial relations that current models struggle to capture reliably. To further explore these weaknesses, we deconstructed the concept-group aggregates (Fig. 6) and reviewed model performance on specific concepts in the CS task with $K = 4$. For example, InternVL2.5 achieved consistently strong results on all concepts within the Size, Shape, and Branching groups, but struggled with certain concepts from the three more difficult groups mentioned above (Contour, Rotation, and Angle):

1. For Angle (69% average accuracy), the model performed particularly poorly on concept #76 *"Long sides concave vs. Long sides convex"* (16%), which requires abstracting beyond surface-level features to recognize a geometric property across diverse real-world scenes.

2. For Rotation (73%), performance dropped on concept #17 *"An acute angle directed inward vs. No acute angle directed inward"* (43%), again pointing to difficulties in detecting precise spatial arrangements.

3. For Contour (56%), concept #3 *"Outline figures vs. Solid figures"* (22%) was especially challenging, suggesting difficulties in recognizing fine-grained details such as object boundaries.

Solving these tasks requires abstraction beyond immediate object features – precisely the type of reasoning our benchmark aims to measure.

We also investigated typical model error patterns in the CS task by manually reviewing prediction explanations. Several recurring failure modes emerged: (1) predicting a concept that applies only to a subset of images on a given side, (2) confusing the sides images belong to (3) predicting a concept for one side while ignoring that it also applies to the opposite side, (4) relying on shallow surface-level features rather than deep image understanding required to detect fine-grained concepts. Fig. 17 presents the examples.

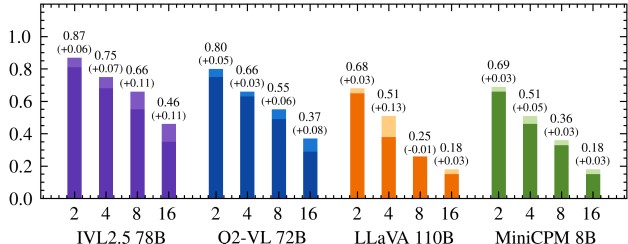

Figure 13: Model performance on Bongard-RWR+/L2 and the corresponding gain over Bongard-RWR+/L2-Impure in the CS task for $K \in \{2, 4, 8, 16\}$. Substituting a human-approved image with one rejected during filtering hinders concept recognition.

Table 13: **Relative difficulty of Bongard-RWR+/L6 vs. Bongard-RWR+.** Each entry reports accuracy on Bongard-RWR+/L6, with the absolute change vs. Bongard-RWR+ shown in parentheses.

| | Image | | | Text | |
|---|---|---|---|---|---|
| | CS, $K = 16$ | I1S | I2S | D1S | D2S |
| InternVL2.5 78B | **0.62** (+0.05) | 0.55 (+0.05) | 0.48 (+0.09) | 0.67 (+0.10) | 0.64 (+0.15) |
| Qwen2-VL 72B | 0.41 (+0.03) | 0.57 (+0.08) | 0.52 (+0.08) | 0.67 (+0.09) | 0.56 (+0.14) |
| LLaVA-Next 110B | 0.20 (+0.01) | 0.51 (+0.01) | 0.49 (−0.01) | 0.61 (+0.07) | 0.52 (+0.09) |
| MiniCPM-o 2.6 8B | 0.18 (−0.01) | 0.51 (+0.03) | 0.51 (+0.06) | 0.53 (+0.02) | 0.45 (+0.04) |
| DeepSeek-R1 70B | N/A | N/A | N/A | **0.70** (+0.13) | **0.72** (+0.16) |
| Similarity Classifier | N/A | **0.59** (+0.07) | **0.68** (+0.14) | 0.54 (+0.05) | 0.59 (+0.09) |

## C.11 FRONTIER PROPRIETARY MODELS

To contextualize the performance of open VLMs, we conducted a limited evaluation of three state-of-the-art proprietary models—Claude Sonnet 4.5, Gemini 2.5 Pro, and GPT 5.1—on Bongard-RWR+ across the I1S, D1S, and CS ($K = 16$) tasks. As shown in Fig. 18a, these models consistently outperform open ones (cf. Fig. 5 and Table 2), with particularly strong and stable performance on the Shape, Size, and Count concept groups (Fig. 18b). Their advantage is most pronounced in the vision-intensive I1S task, where all open models fall to near-chance accuracy despite performing reasonably well on the same groups in CS and, to a lesser extent, D1S. Nevertheless, even the best-performing proprietary model (Gemini 2.5 Pro) reaches only $65\%$ on I1S, $70\%$ on D1S, and $82\%$ on CS, indicating that Bongard-RWR+ presents a significant challenge also for frontier models.

## D LIMITATIONS AND FUTURE WORK

Our dataset generation pipeline requires manual verification to ensure that generated images accurately reflect the intended concepts. During initial experiments, we identified limitations in the ability of T2I models to reliably render certain abstract or fine-grained visual concepts. For instance, the concept *"there are (no) inside figures of the second order"* from Bongard-RWR could not be accurately represented by the T2I model, preventing us from including it in Bongard-RWR+. We believe future advances in accurate representation of fine-grained concepts in generative models will be critical for improving scalability of datasets like Bongard-RWR+.

Our main experiments evaluated the ability of VLMs to solve BPs, alongside selected supervised learning baselines described in Appendix C.4. The results highlight clear limitations in the current models' capacity for abstract visual reasoning, especially when dealing with fine-grained concepts, raising the need for more sophisticated approaches. In particular, multimodal reasoning models that integrate visual and textual information may offer a promising direction, analogous to how reasoning models like DeepSeek-R1 operate in the text domain. We envision the introduced datasets as valuable benchmarks for tracking progress in AVR capabilities and, more broadly, in multi-image reasoning.

Multiple recent studies confirm the limitations of VLMs highlighted in our work. MindSet: Vision (Biscione et al., 2024) tests models on 30 psychological findings inspired by human visual

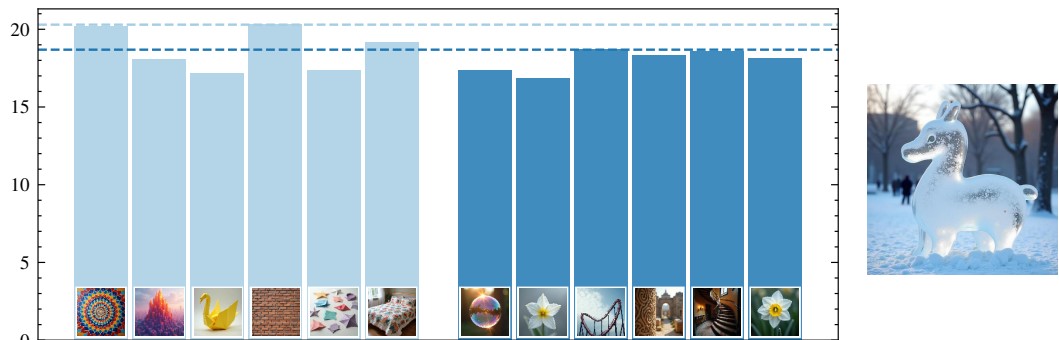

Figure 14: **Model performance on the CS, I1S and D1S tasks across concept groups on Bongard-RWR+.** The presented CS results are for $K = 4$.

Figure 15: **An I1S example illustrating the strength of the Similarity Classifier.** Left: Polygons. Right: Curvilinear figures. The right side shows a test image, and the left side presents a plot of distances from the test image to images on both sides in the ViT-L/14 embedding space. The ice sculpture is farther from the brick wall in the Left side than from any image in the Right side, leading to a higher probability of being classified as a curvilinear figure. In contrast, InternVL2.5 relies on local features and doesn't take into account all side images, resulting in an incorrect classification: "The test image shows a symmetric ice sculpture, matching the symmetric patterns seen in the LEFT class images. Conversely, the RIGHT class images feature more natural, organic, or architectural elements that lack symmetry."

perception, revealing fundamental differences between human and machine vision. VCog-Bench (Cao et al., 2025) evaluates models on abstract and commonsense visual reasoning as well as visual question answering, showing that VLMs consistently struggle with multi-image reasoning tasks. Evaluations on the WAIS-IV test indicate that VLMs underperform in perceptual reasoning tasks (Galatzer-Levy et al., 2024). A systematic review of AVR performance across such benchmarks could provide a more comprehensive understanding of current model capabilities and limitations.

Our evaluation of bias and fairness (Appendix E) reveals that, although Flux.1-dev offers certain demographic variation, it disproportionately generates White figures and tends to produce racially homogeneous crowds. These results are consistent with prior studies on demographic biases in Text-to-Image models (Shukla et al., 2025), highlighting the need for further research on diversity and fairness in Text-to-Image generation. Nevertheless, we do not expect these demographic imbalances to affect the benchmark's validity for its primary purpose: evaluating abstract visual reasoning. The concepts underlying Bongard-RWR+ (e.g., spatial relations, counting, rotations) are independent of demographic attributes such as gender, race, or age. Human figures serve only as one possible way for expressing an abstract rule, not as entities whose demographic characteristics are relevant to the task.

Although we consider the expansion of Bongard-RWR dataset (60 instances) to Bongard-RWR+ (5400 instances) a significant step forward, some Bongard-RWR matrices and many original Bongard

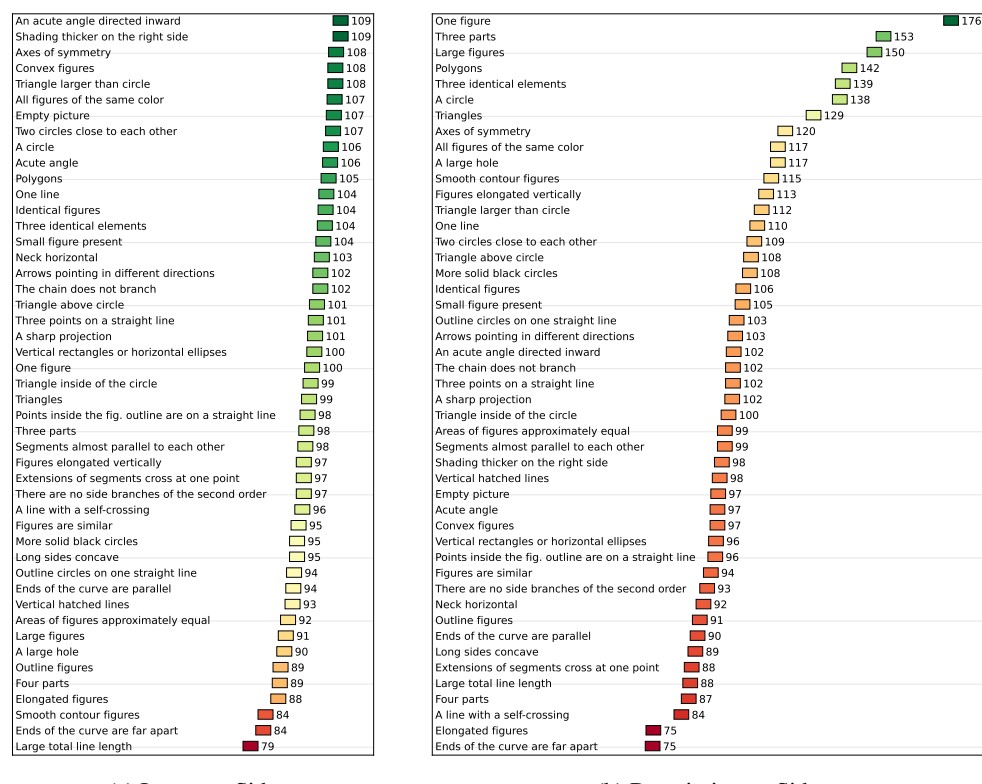

(a) Image-to-Side                    (b) Description-to-Side

Figure 16: **Per-concept difficulty distributions.** Average number of correctly solved instances (out of 200) per concept across all models in Bongard-RWR+ for (a) I1S and (b) D1S tasks.

problems remain unrepresented in the extended Bongard-RWR+ dataset. As we note in Appendix E.2, this is mainly due to the limited number of images passing review and the difficulty of fully capturing complex synthetic concepts in the real-world images (e.g., *"spiral curls clockwise/counterclockwise"*). We believe that expanding Bongard-RWR+ to cover the missing concepts is a challenging yet promising research direction, and we encourage the community to contribute to this endeavor. Below, we outline several directions that look promising.

**(1) Advancing T2I models.** In our current pipeline, human verification served primarily as a quality control step. In practice, we filtered approximately $30.2\%$ of generated images, which indicates that the majority of outputs were already acceptable. This suggests that the pipeline could operate with minimal human intervention or potentially in a fully automated manner as generative models become more reliable at following complex prompts and avoiding rendering unintended concepts. Our approach is therefore forward-compatible with future advances in generative modeling.

**(2) Reducing demographic bias.** The demographic skew in our dataset (Appendix E) stems from the underlying T2I model. While our pipeline does not amplify these biases, future refinements could mitigate them directly. One approach is to explicitly encode demographic diversity into prompts during both the Augment and Generate stages – for example, by specifying varied gender, racial, or age attributes when appropriate for expressing an abstract concept. Additionally, greater diversity could be achieved by leveraging a broader set of I2T, T2T, and T2I models within the pipeline, thereby reducing reliance on any single generative model.

**(3) Cycle-consistency verification.** In our current pipeline, a T2I model generates images from textual descriptions, and humans verify whether the outputs faithfully depict the intended concept. A natural extension is to introduce an automated cycle-consistency check: after generating an image from a source caption, an I2T model could re-caption the image to produce a reference caption. A T2T model would then compare the source and reference captions to assess semantic alignment.

Table 14: Top 5 correctly recognized concepts for each model using the I1S task on Bongard-RWR+.

| Model | Accuracy | Concept Group | Left-side Rule | Right-side Rule |
|---|---|---|---|---|
| IVL2.5 | 0.64 | Contour | Shading thicker on the right side | Shading thicker on the left side |
| | 0.61 | Similarity | Three identical elements | Four identical elements |
| | 0.61 | Shape | A circle | No circle |
| | 0.59 | Count | Three parts | Five parts |
| | 0.58 | Angle | Acute angle | No acute angle |
| Q2-VL | 0.65 | Count | Empty picture | Not empty picture |
| | 0.62 | Angle | A sharp projection | No sharp projection |
| | 0.61 | Angle | Convex figures | Nonconvex figures |
| | 0.59 | Position | Axes of symmetry | No axes of symmetry |
| | 0.59 | Branching | The chain does not branch | The chain branches |
| LLaVA | 0.55 | Angle | A sharp projection | No sharp projection |
| | 0.55 | Rotation | An acute angle directed inward | No angle directed inward |
| | 0.55 | Similarity | All figures of the same color | Figures of different colors |
| | 0.54 | Position | Points inside the figure outline are on a straight line | Points inside the figure outline are not on a straight line |
| | 0.54 | Similarity | Figures are similar | Figures are not similar |
| MCPM | 0.61 | Similarity | All figures of the same color | Figures of different colors |
| | 0.60 | Count | One figure | Two figures |
| | 0.58 | Similarity | Figures are similar | Figures are not similar |
| | 0.53 | Angle | Convex figures | Nonconvex figures |
| | 0.53 | Branching | There are no side branches of the second order | There are side branches of the second order |
| Gemini 2.5 Pro | 1.00 | Count | One figure | Two figures |
| | 0.93 | Size | Large figures | Small figures |
| | 0.93 | Shape | Triangles | Circles |
| | 0.92 | Angle | A sharp projection | No sharp projection |
| | 0.91 | Similarity | All figures of the same color | Figures of different colors |
| Claude Sonnet 4.5 | 0.97 | Count | One figure | Two figures |
| | 0.92 | Similarity | All figures of the same color | Figures of different colors |
| | 0.91 | Angle | A sharp projection | No sharp projection |
| | 0.90 | Shape | Triangles | Circles |
| | 0.88 | Count | Three parts | Four parts |
| GPT 5.1 | 1.00 | Count | One figure | Two figures |
| | 0.98 | Position | Triangle inside of the circle | Circle inside of the triangle |
| | 0.95 | Shape | Triangles | Circles |
| | 0.94 | Angle | A sharp projection | No sharp projection |
| | 0.93 | Similarity | All figures of the same color | Figures of different colors |

Images whose re-captioned descriptions fail to preserve key elements of the source prompt could be filtered automatically.

**(4) Property-based testing.** A more structured direction for automation is to move beyond free-form captions and operate on explicit image properties. During the Augment stage, the T2T model could be prompted not only to produce a natural language description of the scene, but also output a set of properties that the generated image should satisfy to faithfully express the intended concept. An initial closed vocabulary could include attributes such as size, shape, count, rotation, lighting, camera perspective, color, and distance. Alternatively, the model could generate open-vocabulary, instance-specific properties tailored to each concept. After image generation, an I2T model could verify whether these properties are present in the output. Images that fail to meet the specified property set (or fall below a chosen threshold) could be filtered automatically.

We note, however, that while such model-based approaches could improve scalability, some human oversight will likely remain important in the near term to ensure high data quality.

Table 15: Top 5 correctly recognized concepts for each model using the D1S task on Bongard-RWR+.

| Model | Accuracy | Concept Group | Left-side Rule | Right-side Rule |
|---|---|---|---|---|
| IVL2.5 | 0.98 | Count | One figure | Two figures |
| | 0.93 | Count | Three parts | Four parts |
| | 0.86 | Shape | Triangles | Circles |
| | 0.84 | Size | Large figures | Small figures |
| | 0.78 | Count | Three parts | Five parts |
| Q2-VL | 0.96 | Count | Three parts | Four parts |
| | 0.95 | Count | One figure | Two figures |
| | 0.85 | Shape | Triangles | Circles |
| | 0.83 | Size | Large figures | Small figures |
| | 0.81 | Shape | Polygons | Curvilinear figures |
| LLaVA | 0.86 | Count | Three parts | Four parts |
| | 0.85 | Count | One figure | Two figures |
| | 0.81 | Shape | Polygons | Curvilinear figures |
| | 0.78 | Size | Large figures | Small figures |
| | 0.71 | Count | Three parts | Five parts |
| MCPM | 0.80 | Count | Three parts | Four parts |
| | 0.73 | Count | One figure | Two figures |
| | 0.59 | Count | Three parts | Five parts |
| | 0.59 | Similarity | Three identical elements | Four identical elements |
| | 0.57 | Similarity | All figures of the same color | Figures of different colors |
| DS-R1 | 1.00 | Count | One figure | Two figures |
| | 0.98 | Count | Three parts | Four parts |
| | 0.85 | Shape | Polygons | Curvilinear figures |
| | 0.84 | Shape | Triangles | Circles |
| | 0.78 | Size | Large figures | Small figures |
| Gemini 2.5 Pro | 1.00 | Count | One figure | Two figures |
| | 1.00 | Count | Three parts | Four parts |
| | 0.98 | Position | Triangle inside of the circle | Circle inside of the triangle |
| | 0.97 | Similarity | All figures of the same color | Figures of different colors |
| | 0.96 | Size | Triangle larger than circle | Triangle smaller than circle |
| Claude Sonnet 4.5 | 0.99 | Count | One figure | Two figures |
| | 0.98 | Similarity | All figures of the same color | Figures of different colors |
| | 0.94 | Shape | Polygons | Curvilinear figures |
| | 0.90 | Count | Three parts | Four parts |
| | 0.88 | Contour | Large total line length | Small total line length |
| GPT-5.1 | 1.00 | Count | One figure | Two figures |
| | 0.98 | Position | Triangle inside of the circle | Circle inside of the triangle |
| | 0.96 | Similarity | All figures of the same color | Figures of different colors |
| | 0.95 | Shape | Polygons | Curvilinear figures |
| | 0.94 | Count | Three parts | Four parts |

# E  BIAS AND FAIRNESS

## E.1  DEMOGRAPHIC BIAS AND DIVERSITY

Our data generation pipeline includes several design choices aimed at promoting visual diversity and thereby mitigating potential biases. First, augmentations made in Algorithm 1 introduce variation in visual scenes, thus increasing conceptual and demographic diversity. For example, a description originally referencing *"A man standing with a surfboard"* was augmented to *"A woman standing with a surfboard"* (Fig. 19). Second, in Algorithm 2 we explicitly encourage visual diversity within BPs representing the same concept. This is accomplished through iteratively selecting image subsets that maximize intra-set visual diversity, as measured by the total pairwise cosine similarity of ViT-L/14 embeddings. Furthermore, after each matrix is constructed, we randomly exclude one image from the selected subset from future sampling, thus increasing the total number of unique images used.

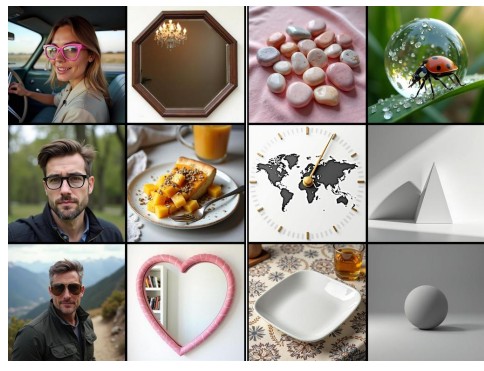

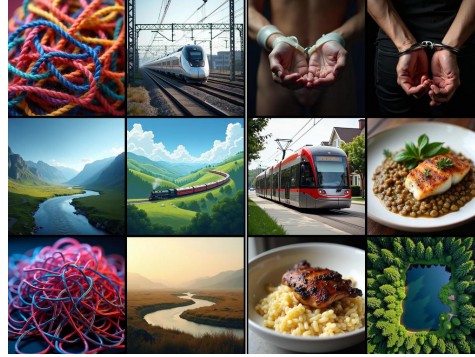

(a) Outline vs. solid figures          (b) Large vs. small total line length

Figure 17: **Failure mode examples.** (a) InternVL2.5's prediction: "Convex hull of figure elongated vs. Convex hull of figure compact"; explanation: "The images on the left side feature objects or shapes that are more elongated, such as the person's face, the mirror, the stones, and the clock. [...]". Assessment: The omission of the pie in the description of the left side illustrates error type (1). Furthermore, while faces and mirrors indeed appeared on the left side, the stones and clock were on the right, exemplifying error type (2). (b) Qwen2-VL's prediction: "Closed lines vs. Open lines"; explanation: "The left side of the image contains natural landscapes [...]". Assessment: Although some left-side images indeed contain landscapes, landscapes also appeared on the right side, exemplifying error type (3). The explanation also reflects type (4) error, as it relies on surface-level features while ignoring the underlying geometric property.

| Model | CS | D1S | I1S |
|---|---|---|---|
| Gemini 2.5 Pro | **0.82** | **0.70** | **0.65** |
| GPT 5.1 | 0.81 | 0.67 | 0.65 |
| Claude Sonnet 4.5 | 0.76 | 0.67 | 0.63 |

(a)

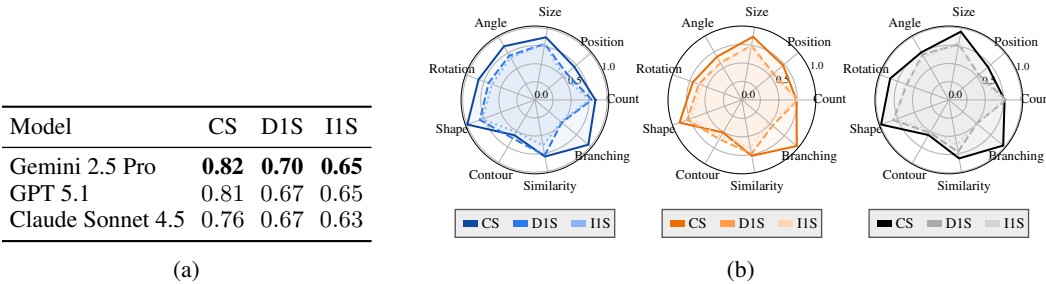

(b)

Figure 18: Proprietary model results on Bongard-RWR+ in the CS ($K = 16$), D1S, and I1S tasks.

Despite these methodological safeguards, Bongard-RWR+ may still inherit biases from the underlying models used in the generation pipeline, particularly Flux.1-dev for Text-to-Image generation. Since these models are trained on large-scale web data, they may reflect social, cultural, or geographic biases present in that data. To better understand these risks, we conducted a limited-scale bias audit. Two expert annotators independently reviewed a subset of 800 images from Bongard-RWR+ depicting humans, and to each of them assigned labels related to Gender (Female, Male, Hard to tell), Race (Asian, Black, White, Hard to tell), Age (Adult, Child, Hard to tell), and Image type (Single, Pair, Crowd, Hands, Legs, Other). We assessed the annotators agreement computing Cohen's $\kappa$ for each of the categories (Gender = 0.82, Race = 0.84, Age = 0.78, Image type = 0.89).

The performed analysis (Fig. 20) revealed that Flux.1-dev produces a relatively balanced Gender distribution (45.1% Female, 36.6% Male, 33.0% Hard to tell). However, the model shows racial imbalance: 79.9% of generated humans were annotated as White, rising to 98.7% in images limited to human hands. Adults were far more frequent (67.9%) than Children (12.8%). Crowd scenes were largely racially homogeneous, with 82.2% appearing to consist of a single race.

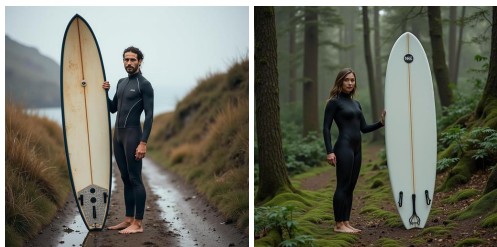

(a) *"A man standing with a surfboard..."* was augmented to *"A woman standing with a surfboard..."*.

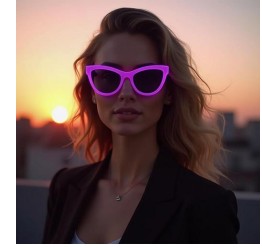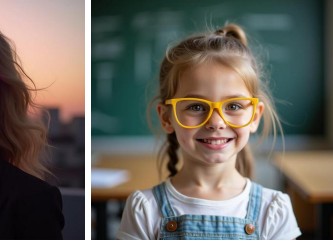

(b) *"A woman wearing retro glasses with a pink outline..."* was augmented to *"A child wearing square glasses with a yellow outline..."*.

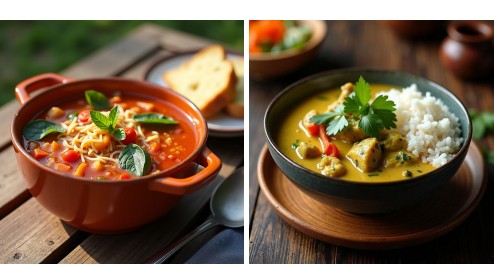

(c) *"A hearty bowl of Italian minestrone soup..."* was augmented to *"A bowl of Thai green curry with jasmine rice..."*.

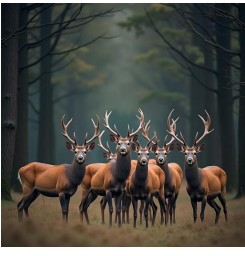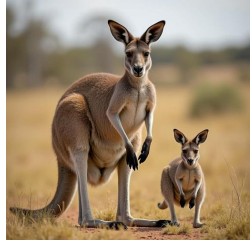

(d) *"A herd of deer standing together in a dense forest..."* was augmented to *"A group of kangaroos hopping closely in a grassy plain..."*.

Figure 19: **Examples of (a) Gender, (b) Age, (c) Culture and (d) Geographic diversity augmentations used in Algorithm 1**.

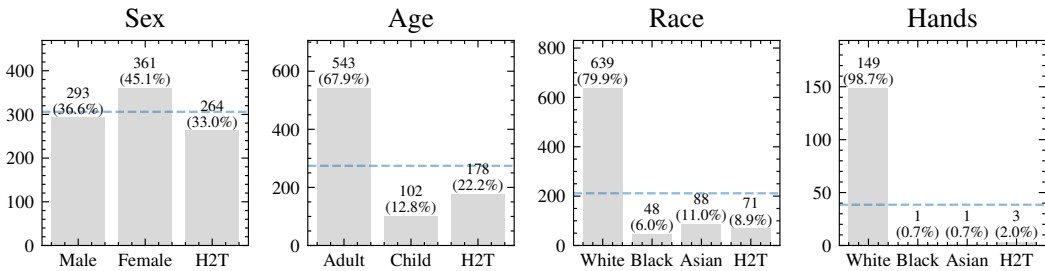

Figure 20: **Distribution of the four selected human categories in Bongard-RWR+ images (Gender, Race, Age, Image type).** The blue dashed line indicates the expected value under equal group representation. *H2T* is used as an abbreviation for Hard to tell.

## E.2 ANNOTATION RELIABILITY AND BIAS

In our pipeline (Fig. 3), the human role is purely discriminative rather than generative and limited to filtering generated images. This substantially reduces the manual burden compared to annotation-heavy approaches such as object detection or image segmentation. Nevertheless, this dicriminative step may potentially introduce some bias. To mitigate this risk, each image generated by Algorithm 1 was independently reviewed by two expert annotators, who assigned one of the 3 labels: Left, Right or None. This way, each annotator verified whether an image (1) adhered to the concept of its side of the matrix and (2) did not adhere to the concept of the opposing side. Images failing either criterion were flagged for exclusion.

Inter-annotator agreement reached a Cohen's $\kappa$ of $0.64$, indicating substantial consensus. Annotators disagreed on side assignment (Left vs. Right) in $5.0\%$ of the cases, disagreed between None vs. either Left or Right in $17.7\%$, and agreed on the same label in $77.3\%$. Disagreements were resolved through discussion. In case of the lack of an agreement, the image was discarded. Overall, $30.2\%$ generated images were discarded. On average, we removed $52\%$ of images for concepts that were ultimately rejected, compared to $23\%$ of them for concepts that were retained. Table 16 reports the

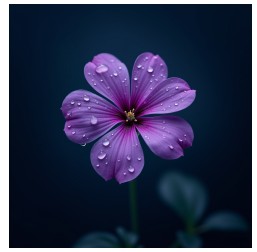 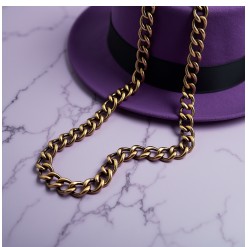 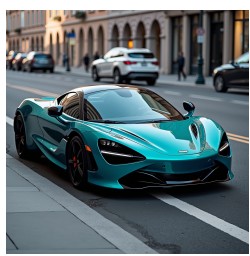 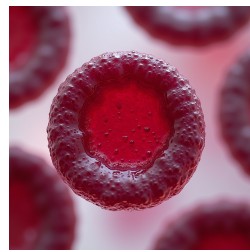

(a) Left: Four parts. Right: Five parts.

(b) Left: Ends of the curve are parallel. Right: Ends of the curve are perpendicular.

(c) Left: One line. Right: Two lines.

(d) Left: Convex figures. Right: Nonconvex figures.

Figure 21: **Examples of images dropped during review.** (a) Depicts six petals, and therefore does not fit either side of the problem. (b) The ends of the chain extend beyond the image, making the concept ambiguous. (c) In front of the car there is a single white line, but in the background two yellow lines are visible. (d) The blood vessel appears circular, which implies convexity; however, when viewed in three dimensions, it is concave.

percentage of generated images discarded during human filtering, broken down by concept group. Rotation concepts show the highest rejection rate ($39\%$), followed by Contour and Count (both $30\%$), indicating these are most challenging for the T2I model to render faithfully. In contrast, Branching and Similarity have the lowest rejection rates ($14\%$), suggesting they are easier for the pipeline to capture reliably. Overall, this strict filtering also explains why some concepts (e.g., *"spiral curls clockwise/counterclockwise"*) did not reach the target of $100$ matrices, due to the limited number of images passing review.

Additionally, we performed a more detailed per-concept analysis, focusing on the top five concepts with the lowest and highest image dropout. Concepts requiring strong object recognition skills had the lowest dropout (under $5\%$), such as: *"A sharp projection vs. No sharp projection"*, *"Triangles vs. Other figures"*, *"Three parts vs. Four parts"*, *"Vertical hatched lines vs. Horizontal hatched lines"*, and *"Triangle inside of the circle vs. Circle inside of the triangle"*. To understand why images were rejected, annotators explained their decisions for the highest-dropout concepts. The main reasons for rejection can be grouped into three categories: (1) the image did not satisfy the concept constraints, (2) the image was ambiguous (e.g., conflicting cues in the foreground and background), or (3) the image did not provide enough information to judge the concept. For example, in *"Ends of the curve are parallel vs. Ends of the curve are perpendicular"*, images were rejected mainly because at least one end of the curve lay outside the frame, preventing clear recognition of the concept. Selected examples are presented in Fig. 21.

Table 16: Percentage of discarded generated images per concept group.

| Concept Group | Rejection Rate |
| --- | --- |
| Rotation | 39% |
| Contour | 30% |
| Count | 30% |
| Shape | 27% |
| Position | 26% |
| Size | 19% |
| Angle | 19% |
| Branching | 14% |
| Similarity | 14% |

## F  MODEL PROMPTS

Prompts 1, 2, and 3 correspond to the Describe, Augment, and Render steps in Algorithm 1, resp. Prompts 4, 6, 7–8, 10, and 11–12 are used for the CS, I1S, D1S, I2S, and D2S tasks. Prompt 5 outlines the shared task introduction for both I1S and D1S, while Prompt 9 provides the common introduction for I2S and D2S. Prompts include 2 illustrative examples to help models understand the task format. Importantly, these examples use different class labels to avoid biasing the model toward always selecting the same answer as in the examples. When applicable, models are instructed to respond in a structured JSON format. The validity of these outputs is enforced using the Outlines decoding backend (Willard & Louf, 2023).

**Prompt 1**: Generate both positive and negative image descriptions.

```
You are given an image and a general concept. Your task is to refine
↪  the concept into two concise, visually descriptive prompts: one
↪  that aligns with the image (positive prompt) and one that contrasts
↪  with it (negative prompt). Focus on making each prompt specific,
↪  clearly grounded in the image, and reflective of the core idea. You
↪  don't need to match every detail--just convey the main visual
↪  concept.

### Example:
Image: Human legs wearing socks with vertical lines
Concept: Vertical lines.
Positive prompt: Socks with vertical lines
Negative prompt: Socks with horizontal lines

Now, it's your turn:
Concept: {concept}

### Instructions:
1. Generate a positive and negative prompt based on the provided image
↪   and concept.
2. Answer using the following format:
Positive prompt:
Negative prompt:
```

## G   THE USE OF LARGE LANGUAGE MODELS

We used LLMs solely to assist with polishing the manuscript, limited to grammar correction and stylistic refinement. No model contributed to research ideas, analysis, or conclusions. In our experimental pipeline, however, VLMs, LLMs, and T2I models served as the primary tools for data generation. Specifically, they were employed for (1) producing image descriptions, (2) augmenting captions, and (3) generating synthetic images. In addition, we evaluated the AVR abilities of several VLMs and LLMs.

**Prompt 2**: Augment a positive image description.

```
You are tasked with modifying a given prompt for a diffusion model.
Your primary objective is to preserve the specified **concept** while
↪  altering unrelated details or environments.
You are encouraged to create diverse, creative, and unique
↪  augmentations that stay true to the concept but introduce variety
↪  in interpretation.

### Example:
Prompt: "An empty white bowl with a thin black rim placed on a solid
↪  blue background."
Concept: "Empty picture"
Output Augmentations: [
    "An empty red plate on a wooden table.",
    "A clear glass cup sitting on a marble countertop.",
    "A white ceramic vase on a patterned fabric."
]

Now, it's your turn:
Prompt: {prompt}
Concept: {concept}

### Instructions:
1. Generate **{n_augmentations} unique augmentations** for the provided
↪  prompt.
2. Output the results in the following JSON string array format:
[
    "<augmented_prompt_1>",
    "<augmented_prompt_2>",
    ...
]

Ensure that each augmentation aligns with the concept and introduces
↪  creative variations in other details.
```

**Prompt 3**: Positive prompt for rendering new images. Negative prompt was provided unchanged.

```
Sharp, high quality image of <positive prompt>.
```

**Prompt 4**: Concept Selection (CS) task prompt.

```
You are a vision understanding module (an expert) designed to provide
↪   short, clear and accurate answers. The goal in solving a Bongard
↪   Problem is to identify a concept that differentiates the left and
↪   right matrix sides. You will receive a list of concepts. Only one
↪   of them correctly describes the image.

Select the concept that correctly describes the image. Respond in JSON
↪   using the following format.
EXAMPLE START
REQUEST
{
    "concepts": [
        {
            "left": "Parallel lines",
            "right": "Perpendicular lines",
            "label": 1
        },
        {
            "left": "Negative slope",
            "right": "Positive slope",
            "label": 2
        }
    ]
}
RESPONSE
{
    "explanation": "The image presents several lines that don't
    ↪   intersect.",
    "label": 1
}
EXAMPLE END
EXAMPLE START
REQUEST
{
    "concepts": [
        {
            "left": "Two shapes",
            "right": "Six shapes",
            "label": 1
        },
        {
            "left": "A cycle",
            "right": "No cycle",
            "label": 2
        },
        {
            "left": "Similar shapes",
            "right": "Different shapes",
            "label": 3
        },
        {
            "left": "Small variance",
            "right": "Large variance",
            "label": 4
        }
    ]
}
RESPONSE
{
    "explanation": "All images on the left side feature circles of
    ↪   similar size, while the images on the right side present
    ↪   triangles, squares, and lines.",
    "label": 3
}
EXAMPLE END
REQUEST
<request>
RESPONSE
```

**Prompt 5**: Shared task introduction used in both I1S and D1S settings.

```
You are a vision understanding module (an expert) designed to provide
↪  short, clear and accurate answers. The goal in solving a Bongard
↪  Problem is to classify a test image to the corresponding class.
↪  There are two classes: LEFT and RIGHT. All images belonging to the
↪  LEFT class represent a common, shared concept which is not present
↪  in any image from the RIGHT class, and vice versa - all images
↪  belonging to the RIGHT class represent a common, shared concept
↪  which is not present in any image from the LEFT class.
```

**Prompt 6**: Image-to-Side (I1S) task prompt.

```
Solve the provided Bongard Problem. Respond in json using the following
↪  format.
EXAMPLE START
RESPONSE
{
    "concept": "small vs big",
    "explanation": "The test image shows a small shape, similarly as
    ↪  all images on the left side. Conversely, the images on the
    ↪  right side feature big shapes.",
    "answer": "LEFT"
}
EXAMPLE END
EXAMPLE START
RESPONSE
{
    "concept": "two circles vs one circle",
    "explanation": "The donut sits alone as the primary round element
    ↪  in the scene, matching the RIGHT examples where a single
    ↪  rounded object takes visual focus amid other details, like the
    ↪  soup bowl or drum.",
    "answer": "RIGHT"
}
EXAMPLE END
RESPONSE
```

**Prompt 7**: Description-to-Side (D1S) task prompt. (1/2)

```
Use the descriptions below to solve the provided Bongard Problem.
↪  Respond in JSON using the following format.
EXAMPLE START
REQUEST
{
    "left_descriptions": [
        "A wired telephone with a coiled cord resting on a wooden
        ↪  table",
        "Power lines stretching across a clear blue sky with birds
        ↪  perched on them",
        "A clothes drying rack with colorful clothes hanging under the
        ↪  sunlight",
        "A swing hanging from a tree branch in a park with a child
        ↪  playing nearby",
        "A needle with thread passing through fabric, placed on a
        ↪  sewing table",
        "A pair of shoes with neatly tied laces placed on a tiled
        ↪  floor"
    ],
    "right_descriptions": [
        "A smartphone with a cracked screen lying on a desk with
        ↪  visible damage",
        "A bird flying freely in the sky with its wings spread wide",
        "A pile of folded clothes neatly stacked on a shelf in a
        ↪  wardrobe",
        "A park bench surrounded by fallen leaves in an autumn
        ↪  setting",
        "A piece of fabric with intricate embroidery placed on a sewing
        ↪  table",
        "A pair of slip-on sandals placed on a carpeted floor near a
        ↪  doorway"
    ],
    "test_description": "A sailboat with its sails fully open gliding
    ↪  across a calm lake under a clear sky"
}
RESPONSE
{
    "concept": "line is present vs no line is present",
    "explanation": "The first test image presents a sailboat with ropes
    ↪  hanging the sails. The ropes are lines, the same as items on
    ↪  the LEFT, like the telephone cord, power lines, and
    ↪  shoelaces.",
    "answer": "LEFT"

}
EXAMPLE END
```

**Prompt 8**: Description-to-Side (D1S) task prompt. (2/2)

```
EXAMPLE START
REQUEST
{
    "left_descriptions": [
        "Two bagels on a cutting board beside a knife",
        "A pair of headphones resting on a desk with the earcups facing
        ↪   up",
        "Two apples placed next to each other on a picnic blanket",
        "A pair of camera lenses lined up on a shelf",
        "A bicycle lying on the ground with both wheels visible",
        "A snowman with two stacked parts, one above the other"
    ],
    "right_descriptions": [
        "Two forks placed on a plate",
        "A dartboard mounted on a brick wall with scores written on two
        ↪   papers beside it",
        "A spotlight shining onto a dark floor surrounded by cables",
        "A pie baking in the oven next to a tray of gingers bread men",
        "A record spinning on a turntable with a hand reaching toward
        ↪   it",
        "A teacup seen from above with steam rising",
    ],
    "test_description": "An octagonal dessert plate holding a single
    ↪   donut with sprinkles"
}
RESPONSE
{
    "concept": "two circles vs one circle",
    "explanation": "The donut sits alone as the primary round element
    ↪   in the scene, matching the RIGHT examples where a single
    ↪   rounded object takes visual focus amid other details, like the
    ↪   soup bowl or drum.",
    "answer": "RIGHT"
}
EXAMPLE END
REQUEST
<request>
RESPONSE
```

**Prompt 9**: Shared task introduction used in both I2S and D2S settings.

```
You are a vision understanding module (an expert) designed to provide
↪   short, clear and accurate answers. The goal in solving a Bongard
↪   Problem is to classify each of the two test images to the
↪   corresponding class. There are two classes: LEFT and RIGHT. Each
↪   image belongs to exactly one class (either LEFT or RIGHT). All
↪   images belonging to the LEFT class represent a common, shared
↪   concept which is not present in any image from the RIGHT class, and
↪   vice versa – all images belonging to the RIGHT class represent a
↪   common, shared concept which is not present in any image from the
↪   LEFT class. The two test images belong to different classes.
```

**Prompt 10**: Images-to-Sides (I2S) task prompt.

```
Solve the provided Bongard Problem. Respond in json using the following
↪  format.
EXAMPLE START
RESPONSE
{
    "concept": "small vs big",
    "first": {
        "explanation": "The test image shows a small shape, similarly
        ↪  as all images on the left side. Conversely, the images on
        ↪  the right side feature big shapes.",
        "answer": "LEFT"
    },
    "second": {
        "explanation": "The test image shows a big shape, similarly as
        ↪  all images on the right. The images on the left, on the
        ↪  other hand, feature small shapes.",
        "answer": "RIGHT"
    }
}
EXAMPLE END
EXAMPLE START
RESPONSE
{
    "concept": "two circles vs one circle",
    "first": {
        "explanation": "The donut sits alone as the primary round
        ↪  element in the scene, matching the RIGHT examples where a
        ↪  single rounded object takes visual focus amid other details,
        ↪  like the soup bowl or drum.",
        "answer": "RIGHT"
    },
    "second": {
        "explanation": "The tennis net and two balls represent two
        ↪  circular elements in the scene, aligning with the LEFT
        ↪  examples where pairs of circular objects are present, such
        ↪  as the bagels, apples, or camera lenses.",
        "answer": "LEFT"
    }
}
EXAMPLE END
RESPONSE
```

---

**Prompt 11**: Descriptions-to-Sides (D2S) task prompt. (1/2)

```
Use the image and the descriptions below to solve the provided Bongard
↪  Problem. Respond in JSON using the following format.
EXAMPLE START
REQUEST
{
    "left_descriptions": [
        "A wired telephone with a coiled cord resting on a wooden
        ↪  table",
        "Power lines stretching across a clear blue sky with birds
        ↪  perched on them",
        "A clothes drying rack with colorful clothes hanging under the
        ↪  sunlight",
        "A swing hanging from a tree branch in a park with a child
        ↪  playing nearby",
        "A needle with thread passing through fabric, placed on a
        ↪  sewing table",
        "A pair of shoes with neatly tied laces placed on a tiled
        ↪  floor"
    ],
    "right_descriptions": [
        "A smartphone with a cracked screen lying on a desk with
        ↪  visible damage",
        "A bird flying freely in the sky with its wings spread wide",
        "A pile of folded clothes neatly stacked on a shelf in a
        ↪  wardrobe",
        "A park bench surrounded by fallen leaves in an autumn
        ↪  setting",
        "A piece of fabric with intricate embroidery placed on a sewing
        ↪  table",
        "A pair of slip-on sandals placed on a carpeted floor near a
        ↪  doorway"
    ],
    "first": "A sailboat with its sails fully open gliding across a
    ↪  calm lake under a clear sky",
    "second": "A launch boat with an engine cruising through the water
    ↪  leaving a trail of waves behind"
}
RESPONSE
{
    "concept": "line is present vs no line is present",
    "first": {
        "explanation": "The first test image presents a sailboat with
        ↪  ropes hanging the sails. The ropes are lines, the same as
        ↪  items on the LEFT, like the telephone cord, power lines,
        ↪  and shoelaces.",
        "answer": "LEFT"
    },
    "second": {
        "explanation": "The second image shows a motorized launch boat,
        ↪  which typically doesn't have sails or ropes visible. This
        ↪  aligns more with the RIGHT side examples where no lines are
        ↪  present, like the smartphone, slip-on sandals, or park
        ↪  bench.",
        "answer": "RIGHT"
    }
}
EXAMPLE END
```

**Prompt 12**: Descriptions-to-Sides (D2S) task prompt. (2/2)

```
EXAMPLE START
REQUEST
{
    "left_descriptions": [
        "Two bagels on a cutting board beside a knife",
        "A pair of headphones resting on a desk with the earcups facing
        ↪   up",
        "Two apples placed next to each other on a picnic blanket",
        "A pair of camera lenses lined up on a shelf",
        "A bicycle lying on the ground with both wheels visible",
        "A snowman with two stacked parts, one above the other"
    ],
    "right_descriptions": [
        "Two forks placed on a plate",
        "A dartboard mounted on a brick wall with scores written on two
        ↪   papers beside it",
        "A spotlight shining onto a dark floor surrounded by cables",
        "A pie baking in the oven next to a tray of gingers bread men",
        "A record spinning on a turntable with a hand reaching toward
        ↪   it",
        "A teacup seen from above with steam rising",
    ],
    "first": "An octagonal dessert plate holding a single donut with
    ↪   sprinkles",
    "second": "A tennis net with two balls lying nearby"
}
RESPONSE
{
    "concept": "two circles vs one circle",
    "first": {
        "explanation": "The donut sits alone as the primary round
        ↪   element in the scene, matching the RIGHT examples where a
        ↪   single rounded object takes visual focus amid other details,
        ↪   like the soup bowl or drum.",
        "answer": "RIGHT"
    },
    "second": {
        "explanation": "The tennis net and two balls represent two
        ↪   circular elements in the scene, aligning with the LEFT
        ↪   examples where pairs of circular objects are present, such
        ↪   as the bagels, apples, or camera lenses.",
        "answer": "LEFT"
    }
}
EXAMPLE END
REQUEST
<request>
RESPONSE
```

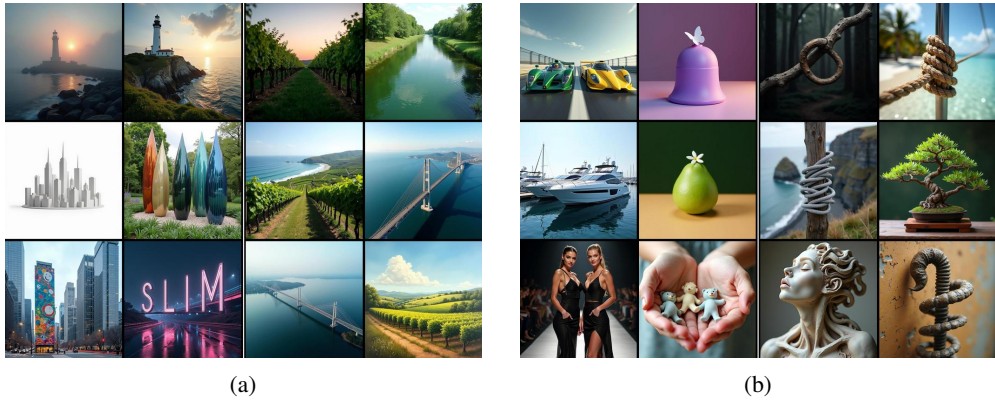

Figure 22: **Bongard-RWR+.** (a) Left: Figures elongated vertically. Right: Figures elongated horizontally. (b) Left: Smooth contour figures. Right: Twisting contour figures.

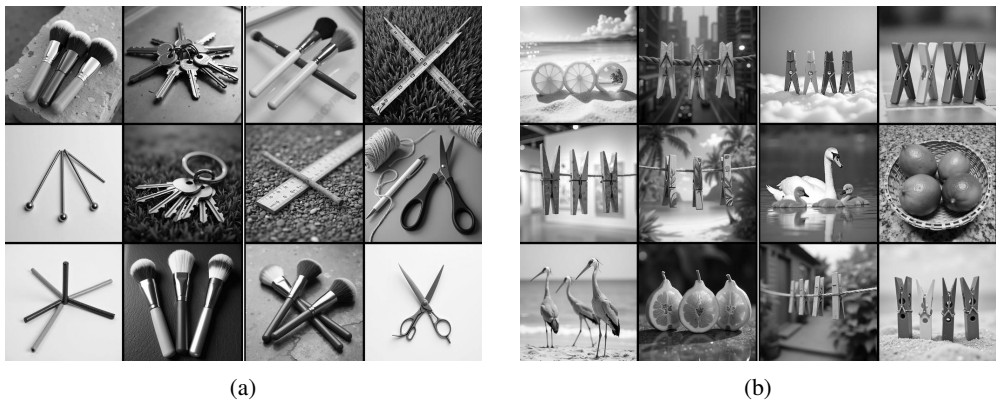

Figure 23: **Bongard-RWR+/GS.** (a) Left: Extensions of segments cross at one point. Right: Extensions of segments do not cross at one point. (b) Left: Three parts. Right: Four parts.

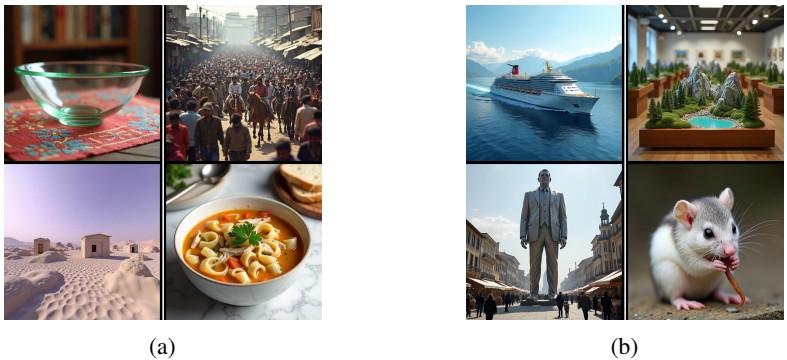

Figure 24: **Bongard-RWR+/L2.** (a) Left: Empty. Right: Not empty. (b) Left: Large figures. Right: Small figures.

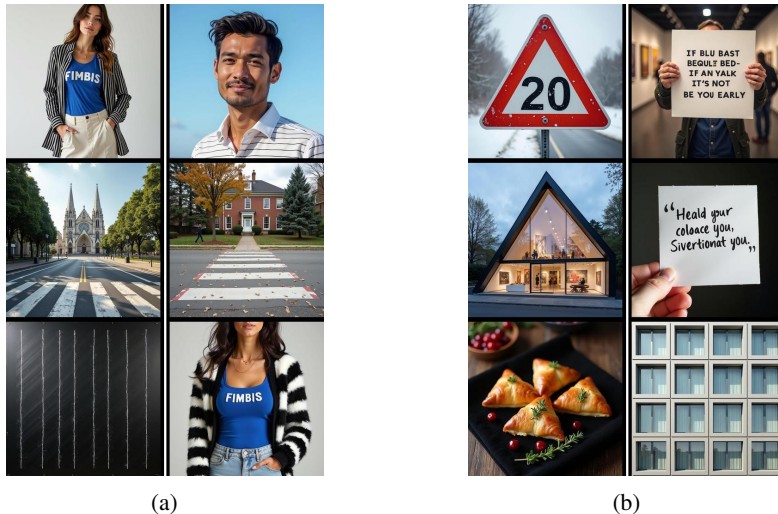

(a)                                                    (b)

Figure 25: **Bongard-RWR+/L3.** (a) Left: Vertical hatched lines. Right: Horizontal hatched lines. (b) Left: Triangles. Right: Quadrangles.

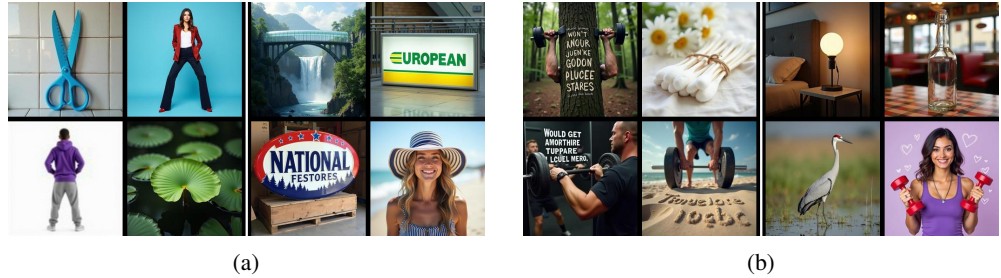

(a)                                                    (b)

Figure 26: **Bongard-RWR+/L4.** (a) Left: An acute angle directed inward. Right: No angle directed inward. (b) Left: Neck horizontal. Right: Neck vertical.

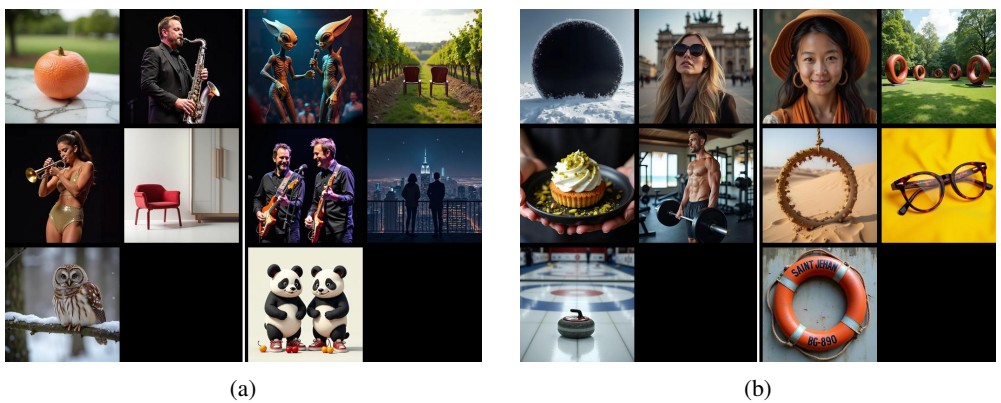

(a)                                                    (b)

Figure 27: **Bongard-RWR+/L5.** (a) Left: One figure. Right: Two figures. (b) Left: More solid black circles. Right: More outline circles.

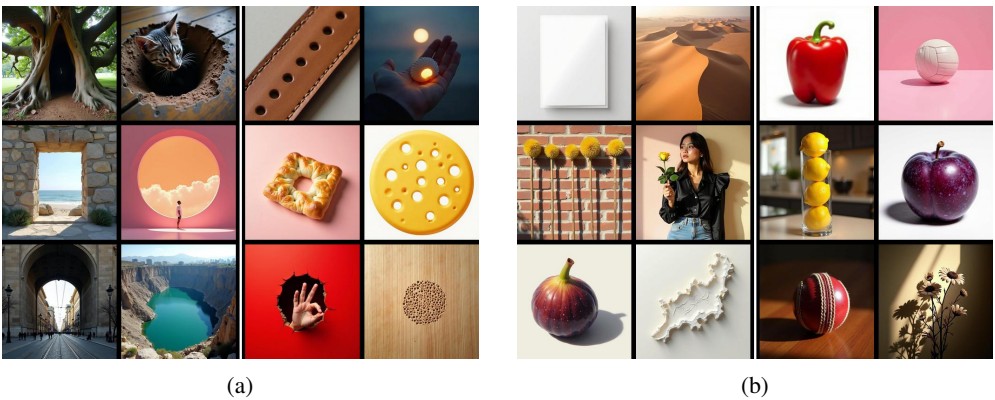

Figure 28: **Bongard-RWR+/L6.** (a) Left: A large hole. Right: A small hole. (b) Left: Shading thicker on the right side. Right: Shading thicker on the left side.

