# OpenReview forum: "Bongard-RWR+: Real-World Representations of Fine-Grained Concepts in Bongard Problems"
_ICLR.cc/2026/Conference — ICLR 2026 Poster_

### Official Review · Reviewer_c11F · 2025-10-15

**Soundness:** 3
**Presentation:** 3
**Contribution:** 3
**Rating:** 8
**Confidence:** 4

**Summary:**

This paper is about Bongard Problems (a classical test of abstract visual reasoning).

The first contribution is a pipeline to generate such problems at scale, with synthetic photo-realistic images.

The second contribution is an evaluation of existing models on the resulting set of new problems.

**Strengths:**

The paper addresses a gap in the literature: there was no large dataset of such photo-realistic BPs.
BPs are a classical test so there is interest in keeping to use this paradigm to test AI models.
The literature review does a good job of reviewing the extensive related work in abstract visual reasoning.

The evaluation uses several (4) current large models. It includes several formulations of the task, and multiple interesting variations and ablations (model size, color vs. grayscale,

**Weaknesses:**

I don't see any clear weakness to this paper.

**Questions:**

N/A

---
Tips for the presentation:

- Typo on the 2nd line of the abstract
- Figure 5: may be interesting to mark the "random choice" ("chance") baseline for each K, e.g. as a small gray line?
- Inconsistent capitalization in paragraph headers: e.g. L309 "Concept selection" (no need to capitalize "selection", it's still a simple noun even if it's the name of an experiment), L342, L357, L465
- Figure 8: I don't think these are "ablation" experiments; ablation means *taking away* (some part of the model, usually). A correct name for these experiments could be, for example, a "sensitivity analysis".
- In the experiments, it may be worth reminding a few more times what some key symbols mean (K, P), for the readers who will scan through the paper without reading everything top to bottom. Another option may be to replace these "unguessable" symbols by something more explicit (mabye something like N_candidates? I'm not totally sure it's a good idea, maybe it's too cumbersome). Or maybe include/define these symbols in one of the figures that describe the dataset/setup?

---

> ### Author Response · Authors · 2025-11-20
> **Response to Reviewer c11F**
>
> > I don't see any clear weakness to this paper.
>
> Thank you for the positive evaluation of our work.
>
>
> > Typo on the 2nd line of the abstract
>
> We corrected the typo, changing “fromjust” to “from just” in the revision.
>
>
> > Figure 5: may be interesting to mark the "random choice" ("chance") baseline for each K, e.g. as a small gray line?
>
> Thank you for this suggestion. We updated Figure 5 to include the random guess baseline for each value of K.
>
>
> > Inconsistent capitalization in paragraph headers: e.g. L309 "Concept selection" (no need to capitalize "selection", it's still a simple noun even if it's the name of an experiment), L342, L357, L465
>
> We applied the recommended corrections.
>
>
> > Figure 8: I don't think these are "ablation" experiments; ablation means taking away (some part of the model, usually). A correct name for these experiments could be, for example, a "sensitivity analysis".
>
> Thank you for this comment. We changed the caption wording from “Concept Selection ablations” to “Concept Selection sensitivity analysis”.
>
>
> > In the experiments, it may be worth reminding a few more times what some key symbols mean (K, P), for the readers who will scan through the paper without reading everything top to bottom. Another option may be to replace these "unguessable" symbols by something more explicit (mabye something like N_candidates? I'm not totally sure it's a good idea, maybe it's too cumbersome). Or maybe include/define these symbols in one of the figures that describe the dataset/setup?
>
> Due to the strict page limit, we relied on previously defined notation in the main paper. However, based on the Reviewer’s suggestion, we added clarifications in relevant places. Specifically:
> 1. Figure 9: “for P = 2, 3.” -> “for P = 2, 3 panels.”
> 2. Section 4 “Is image color necessary for concept recognition?”: “for K = 2.” -> “for K = 2 concepts.”
> 3. Appendix C.2: “for P = 2, 3, 4.” -> “for a varying number of images per side P = 2, 3, 4.”
> 4. Figure 12: “for P = 2, 3, 4.” -> “for a varying number of images per side P = 2, 3, 4.”
> 5. Table 5: “for K = 16.” -> “for K = 16 concepts.”

---

### Official Review · Reviewer_SHcE · 2025-10-27

**Soundness:** 3
**Presentation:** 3
**Contribution:** 2
**Rating:** 2
**Confidence:** 5

**Summary:**

This paper introduces Bongard-RWR+, a dataset of 5,400 Bongard Problems that uses AI-generated real-world-like images to represent abstract visual concepts from classical Bongard Problems. The authors develop a semi-automated pipeline using vision-language models to generate images, then evaluate state-of-the-art VLMs on multiple task formulations including binary classification, multiclass concept selection, and free-form text generation. Their experiments reveal that while current VLMs can recognize coarse-grained visual concepts, they consistently struggle with fine-grained concept recognition, highlighting limitations in their abstract visual reasoning capabilities.

**Strengths:**

The paper addresses a relevant problem by scaling up the Bongard-RWR dataset from 60 to 5,400 instances through a semi-automated generation pipeline, which represents a reasonable engineering contribution to the abstract visual reasoning benchmark landscape. The experimental evaluation is comprehensive, covering multiple task formulations (binary classification, multiclass selection, text generation) and including useful ablation studies on factors like model size, color versus grayscale, and number of demonstrations. The paper is generally well-structured and clearly written, with detailed appendices documenting the generation process, prompts, and bias analysis. The findings consistently demonstrate that current VLMs struggle with fine-grained visual reasoning, which confirms existing concerns about their capabilities.

**Weaknesses:**

The paper's core limitation is that it provides dataset scaling rather than methodological innovation. The reliance on generated images raises validity concerns, especially given significant demographic bias (79.9% White figures) and the number of exclusion of original concepts due to generation failures, suggesting the approach cannot capture full abstract reasoning complexity. The experimental analysis is shallow—it confirms known VLM limitations without investigating why models fail, lacks detailed error analysis or failure mode characterization, and dismisses notable performance differences  between real and generated images rather than examining their implications. The supervised baselines achieve random performance but receive minimal analysis, missing opportunities to understand whether difficulties stem from data, architecture, or fundamental reasoning gaps. Additionally, the finding that explicit image captioning improves performance suggests models may be exploiting caption artifacts rather than demonstrating genuine visual reasoning.

**Questions:**

no

---

> ### Author Response · Authors · 2025-11-20
> **Response to Reviewer SHcE 1/**
>
> > The paper's core limitation is that it provides dataset scaling rather than methodological innovation.
>
> The core contribution of our work is the introduction of Bongard-RWR+, a large-scale benchmark for abstract visual reasoning (AVR), together with the methodology required to construct it. This contribution is aligned with the Datasets & Benchmarks track, where the primary expectation is a high-quality dataset accompanied by a systematic empirical study. That said, our work also introduces non-trivial methodological components that go beyond simple dataset scaling.
>
> As discussed in the Introduction, generating Bongard Problems is inherently challenging:
> 1. Early BPs were handcrafted, yielding only a few hundred black-and-white instances.
> 2. Procedural generation in Bongard-LOGO increased scale but remained restricted to synthetic black-and-white line drawings.
> 3. Real-world benchmarks such as Bongard HOI and Bongard-OpenWorld focus on coarse-grained concepts, where solving BPs often reduces to object recognition rather than abstract reasoning.
> 4. Bongard-RWR, closest to ours, relied entirely on manual construction, thus preventing scaling and limiting evaluation robustness.
>
> In contrast, our pipeline introduces a semi-automated approach specifically designed to express fine-grained abstract concepts using real-world-like images. This requires orchestration of I2T, T2T, and T2I models to generate diverse visual representations of abstract concepts (Figure 3, Algorithm 1). We further developed a diversity-aware matrix construction method (Algorithm 2) that assembles BPs in a way that ensures visual diversity.
>
> Finally, this pipeline supports multiple dataset variants (Bongard-RWR+/GS, /LP, /TVT; Appendix 2) and six problem formulations (Figure 4), overall resulting in a comprehensive benchmark for evaluating visual reasoning with over 100 000 Bongard Problem instances.
>
>
> > The reliance on generated images raises validity concerns, especially given significant demographic bias (79.9% White figures) [...]
>
> We acknowledge the Reviewer’s concern regarding demographic bias in generated images. Importantly, this imbalance stems from the underlying T2I model (Flux.1-dev) and is consistent with prior studies on demographic biases in T2I models (Shukla et al., 2025). Our pipeline itself does not introduce or amplify such bias, the same effects would appear whenever Flux.1-dev (or similar models) is used to depict humans. Nevertheless, it is essential to make such limitations explicit. For this reason, we conducted a dedicated bias audit and reported the results in the Ethics statement of the main paper as well as in detail in Appendices D and E.
>
> Crucially, we do not expect these demographic imbalances to affect the benchmark’s validity for its primary purpose: evaluating abstract visual reasoning. The concepts underlying Bongard-RWR+ (e.g., spatial relations, counting, rotations) are independent of demographic attributes such as gender, race, or age. Human figures serve only as one possible way for expressing an abstract rule, not as entities whose demographic characteristics are relevant to the task. We have added this callout to Appendix D.
>
> That said, we agree that demographic biases in T2I models represent an important broader issue. By reporting these findings transparently, we aim to inform users of Bongard-RWR+ about these limitations and encourage future work at the intersection of fairness and image generation.
>
> > [...] and the number of exclusion of original concepts due to generation failures, suggesting the approach cannot capture full abstract reasoning complexity.
>
> Out of the 60 root Bongard-RWR matrices that we provided as input to the pipeline, only 6 (10%) were discarded because the T2I model was unable to reliably produce a sufficient number of images faithfully expressing the underlying abstract rule. These failures reflect known limitations of current T2I models rather than a limitation of our methodology, and we explicitly account for them through a manual verification step to ensure high quality of the final dataset (Appendix D).
>
> Importantly, generation failures did not affect all images associated with the discarded concepts, many were generated correctly. On average, we discarded 52% of images for rejected concepts, compared to 23% of them for accepted concepts. Although it would have been possible to construct a small number of valid BPs from these partial successes, doing so would have introduced undesirable imbalance across concepts. We therefore opted to exclude the affected concepts entirely, prioritizing dataset consistency and quality over marginal increase in coverage.
>
> [...]

---

> > ### Author Response · Authors · 2025-11-20
> > **Response to Reviewer SHcE 2/**
> >
> > [...]
> >
> > To further assess whether generated images provide a valid testbed for evaluating abstract visual reasoning, we directly compared performance on generated images with performance on real images. Specifically, we constructed expanded variants of the original Bongard-RWR dataset (using real-world images only) and evaluated the same models under the same matrix-construction approach as in Bongard-RWR+. Across all models and difficulty levels, we observed highly similar performance trends on both real and generated datasets (“Are generated images as effective as real ones? “ in Section 4 and Appendix C.2). This indicates that generated images capture the same underlying reasoning challenges as real ones and supports the validity of the generation-based approach used in Bongard-RWR+.
> >
> > Taken together, these results suggest that while current T2I models impose some limitations on concept coverage, the resulting dataset remains valid for its primary purpose: evaluating abstract visual reasoning. Advances in T2I models will naturally enable expanding concept coverage in future versions.
> >
> > > The experimental analysis is shallow—it confirms known VLM limitations without investigating why models fail, lacks detailed error analysis or failure mode characterization, and dismisses notable performance differences between real and generated images rather than examining their implications.
> >
> > Thank you for this insightful and important comment. We agree that deeper analysis of model failure modes is valuable. We already conducted several additional investigations but did not include them in the submitted version due to the paper’s already substantial scope, not to overwhelm the readers. We are happy, however, to revise the manuscript based on the Reviewer’s specific recommendations. Below we present the details of this analysis.
> >
> > **Error sources and failure modes**
> >
> > In Section 4 (Concept Selection) we noted that concepts such as Contour, Rotation, and Angle pose particular difficulties because they rely on subtle geometric cues or precise spatial relations that current models struggle to capture reliably. To better understand these weaknesses, we deconstructed the concept-group aggregates (Figure 6) and reviewed model performance on specific concepts in the CS task with K=4. For example, InternVL2.5 achieved consistently strong results on all concepts within the Size, Shape, and Branching groups, but struggled with certain concepts from the three more difficult groups mentioned above (Contour, Rotation, and Angle):
> > * For Angle (69% average accuracy), the model performed particularly poorly on concept #76 *“Long sides concave vs. Long sides convex”* (16%), which requires abstracting beyond surface-level features to recognize a geometric property across diverse real-world scenes.
> > * For Rotation (73%), performance dropped on concept #17 *“An acute angle directed inward vs. No acute angle directed inward”* (43%), again pointing to difficulties in detecting precise spatial arrangements.
> > * For Contour (56%), concept #3 *“Outline figures vs. Solid figures”* (22%) was especially challenging, suggesting difficulties in recognizing fine-grained details such as object boundaries.
> >
> > These cases require abstraction beyond surface-level object features – precisely the type of reasoning our benchmark aims to measure.
> >
> > We also investigated typical model error patterns in the CS task by manually reviewing prediction explanations. Several recurring failure modes emerged:
> > 1. predicting a concept that applies only to a subset of images on the given side,
> > 2. confusing the sides images belong to,
> > 3. predicting a concept for one side while ignoring that it also applies to the opposite side,
> > 4. relying on shallow surface-level features rather than deep image understanding required to detect fine-grained concepts.
> >
> > For example, in the BP presented in the newly added Figure 17a (*“Outline figures vs. Solid figures”*), InternVL2.5 misclassified the concept as *“Convex hull of figure elongated vs. Convex hull of figure compact"*, explaining: *“The images on the left side feature objects or shapes that are more elongated, such as the person's face, the mirror, the stones, and the clock. [...]”*. While faces and mirrors indeed appeared on the left side, the stones and clock were on the right, exemplifying error type (2). Furthermore, the omission of the pie in the description of the left side illustrates error type (1).
> >
> > In Figure 17b (*“Large total line length vs. Small total line length”*), Qwen2-VL incorrectly predicted *“Closed lines vs. Open lines"*, justifying with: *“The left side of the image contains natural landscapes [...]”*. Although some left-side images indeed contain landscapes, landscapes also appeared on the right side, exemplifying error type (3). The explanation also reflects type (4) error, as it relies on surface-level features while ignoring the underlying geometric property.
> >
> > [...]

---

> > > ### Author Response · Authors · 2025-11-20
> > > **Response to Reviewer SHcE 3/**
> > >
> > > [...]
> > >
> > > We have included this extended analysis in the newly added Appendix C.10, along with Figure 17 that illustrates these failure modes. In addition, we will release instance-level predictions of all evaluated models to facilitate deeper error analysis in the future.
> > >
> > > **Real vs. generated images**
> > >
> > > The main implication we draw from analyzing performance differences between real and generated images is that model behavior across these two domains is highly correlated, suggesting that improvements on Bongard-RWR+ should transfer to Bongard-RWR and other real-world BP datasets. This is supported by our analysis in Section 4 (“Are generated images as effective as real ones?”), where we observe consistent performance trends across four CS difficulty levels, as well as by Appendix C.2, which reinforces this finding using the I1S task and provides a per-concept comparison. Together, these analyses indicate that the two domains pose similar challenges for VLMs.
> > >
> > > During rebuttal, in order to quantify this relationship more rigorously, we conducted a statistical analysis across all four models for both counts of panels per side (P = 2, 3) and all difficulty levels (K = 2, 4, 8, 16). We computed Pearson correlations and fitted linear regression models to assess how well performance on the generated dataset (RWR+/LP) predicts performance on the real dataset (RWR). The results are summarized in the Table below.
> > >
> > > | Model            | P | r     | R²    | Slope | Inter. | MAD  | RMSE |
> > > |------------------|---|-------|-------|-------|--------|------|------|
> > > | MiniCPM-o 2.6 8B | 2 | 0.997 | 0.994 | 0.971 | -0.005 | 1.75 | 1.38 |
> > > | MiniCPM-o 2.6 8B | 3 | 0.999 | 0.998 | 0.902 |  0.030 | 2.25 | 0.80 |
> > > | InternVL2.5 78B  | 2 | 0.994 | 0.987 | 1.204 | -0.190 | 5.00 | 2.04 |
> > > | InternVL2.5 78B  | 3 | 0.993 | 0.987 | 1.419 | -0.377 | 7.00 | 2.09 |
> > > | Qwen2-VL 72B     | 2 | 1.000 | 0.999 | 1.005 | -0.015 | 1.25 | 0.43 |
> > > | Qwen2-VL 72B     | 3 | 1.000 | 0.999 | 1.004 | -0.035 | 3.25 | 0.43 |
> > > | LLaVA-Next 110B  | 2 | 0.994 | 0.987 | 0.867 |  0.054 | 3.00 | 1.98 |
> > > | LLaVA-Next 110B  | 3 | 1.000 | 1.000 | 0.936 |  0.040 | 1.50 | 0.26 |
> > >
> > > All models exhibit strong correlations between real and generated dataset performance (r > 0.99), with R² consistently above 0.987. For example, InternVL2.5 78B for P=2 shows r = 0.994 and a regression slope of 1.204, meaning that, for instance, a 10 p.p. gain on RWR+/L2 corresponds to roughly a 12 p.p. gain on RWR. Error measures including mean absolute difference (MAD, <= 7.0) and RMSE (<=2.09) confirm that RWR+/LP is slightly easier on average, but the relationship is stable and predictable across difficulty levels.
> > >
> > > These results demonstrate that the generated dataset preserves the underlying difficulty of the real-image dataset. Consequently, performance improvements on Bongard-RWR+ can be expected to translate reliably to improvements on real-image BPs, supporting the validity of using generated images for AVR evaluation. We have included this analysis in Appendix C.2, with results presented in the newly added Table 4.
> > >
> > > **How input domain shapes concept difficulty**
> > >
> > > During rebuttal, in order to supplement the input modality analysis in Appendix C.9, we plotted absolute per-concept difficulty for both I1S and D1S by computing the avg. number of correctly solved instances per concept, across all models.
> > >
> > > For I1S, each concept was solved in 79-109 instances (out of 200) on average. The relatively narrow range indicates that most concepts remain uniformly challenging for models in the visual domain, which is consistent with their near-random performance. One noteworthy case is “Shading thicker on the right side” (concept #63), which appears among top-5 solved concepts in I1S (109) but is relatively difficult in D1S (92), highlighting the relevance of visual input in identifying fine-grained concepts.
> > >
> > > In contrast, D1S exhibits a much broader difficulty range, from 70 to 182 solved instances, reflecting high performance on certain concepts that appear to be easier. Notably, top-5 solved concepts include counting-based ones such as “three parts” or “one figure”, which may be easier to capture in the text domain.
> > >
> > > We have included this extended analysis in Appendix C.9, with visualization presented in Figure 16.

---

> > > > ### Author Response · Authors · 2025-11-20
> > > > **Response to Reviewer SHcE 4/4**
> > > >
> > > > > The supervised baselines achieve random performance but receive minimal analysis, missing opportunities to understand whether difficulties stem from data, architecture, or fundamental reasoning gaps.
> > > >
> > > > The supervised baselines (Appendix C.4) were introduced as a sanity check to test whether classical models could solve the benchmark when given substantial training data. Even with 7290 training matrices in Bongard-RWR+/TVT and 51840 in Bongard-RWR+/TVT-Large, all approaches failed to achieve meaningful performance. As described in Appendix B, these variants are explicitly designed so that images used in validation and test splits never appear during training, requiring out-of-distribution generalization capabilities – the model must assign a novel test image to the correct side of the problem despite never encountering that image in any training instance. All tested approaches proved insufficient for this challenging generalization task.
> > > >
> > > > More broadly, we believe the difficulties observed in Bongard-RWR+ stem from the intrinsic complexity of the dataset. In the CS task, models perform strongly for K = 2 and clearly surpass random baselines for all K, indicating that they have sufficient high-level understanding of the images to match a matrix to the correct concept when the set of choices is limited. However, they struggle on tasks such as image-to-side assignment (I1S, I2S), which require fine-grained understanding of the underlying concept rather than high-level image recognition. These tasks demand precise analysis of low-level properties across the two sides of a matrix – something that current models consistently fail at.
> > > >
> > > >
> > > > > Additionally, the finding that explicit image captioning improves performance suggests models may be exploiting caption artifacts rather than demonstrating genuine visual reasoning.
> > > >
> > > > For clarification, the captions used in our experiments are not ground-truth annotations that could be exploited; they are model-generated descriptions produced independently for each image. In “Description(s)-to-Side(s)” settings (Section 4), each of the four VLMs first produced captions for all images, and we then used those captions as input to the same models to solve the side prediction task. Similarly, in the D1S experiment in Table 3b the captions were generated by InternVL2.5 78B.
> > > >
> > > > We agree that the performance improvements in text-based settings highlight limitations of current VLMs. Many contemporary VLMs are trained primarily on image-text pairs, making them far more proficient at captioning images than at performing multi-image reasoning directly in the visual domain. When the reasoning task is reformulated into textual space, via model-generated descriptions, the models can leverage their strong language modeling capabilities. Similar effects have been documented in other multimodal reasoning benchmarks (Zhang et al., 2024; Hao et al., 2025), where models often perform better when visual information is first translated into text.
> > > >
> > > > Importantly, the gap between image-based and text-based performance in Bongard-RWR+ reinforces the value of the benchmark: it shows that current VLMs struggle with fine-grained visual reasoning when required to operate directly on images. This gap also highlights that image-based setups of Bongard-RWR+ pose a challenging testbed for multi-image relational reasoning, distinguishing it from benchmarks where models can rely on textual shortcuts.
> > > >
> > > > Zhang, Renrui, et al. “MathVerse: Does Your Multi-modal LLM Truly See the Diagrams in Visual Math Problems?" European Conference on Computer Vision. 2024.
> > > >
> > > > Hao, Yunzhuo, et al. "Can MLLMs Reason in Multimodality? EMMA: An Enhanced MultiModal ReAsoning Benchmark." International Conference on Machine Learning. 2025.

---

### Official Review · Reviewer_sMpD · 2025-10-29

**Soundness:** 2
**Presentation:** 3
**Contribution:** 3
**Rating:** 4
**Confidence:** 3

**Summary:**

The paper introduces Bongard-RWR+, a new large-scale benchmark for abstract visual reasoning, expanding on the Bongard Problems by generating 5,400 instances using a vision-language model pipeline. Despite promising results with coarse-grained concepts, state-of-the-art VLMs struggle with fine-grained reasoning tasks, highlighting significant gaps in their capabilities for multi-image abstract visual reasoning.

**Strengths:**

1. This paper introduces Bongard-RWR+, an innovative extension of the Bongard Problems, using a semi-automated pipeline with vision-language models (VLMs) to generate large-scale, fine-grained real-world images, addressing the limitations of manual datasets like Bongard-RWR.
2. The methodology is solid, ensuring image diversity and offering valuable insights through ablations on model size, color, and image diversity.
3. The paper is clear and well-structured, with effective visuals. Its significance lies in advancing AVR, providing a scalable benchmark that exposes gaps in current VLM capabilities and guides future research in multimodal reasoning.

**Weaknesses:**

1.The paper lacks a detailed exploration of how the semi-automated pipeline can be further refined for more reliable image generation. Manual filtering still plays a key role, and automated verification could improve scalability and reduce bias.
2.Although the experiments cover multiple tasks, the evaluation of fine-grained reasoning is limited. It would benefit from including more diverse models or comparing performance with human-level reasoning.
3.Lastly, a deeper analysis of errors, especially in fine-grained tasks, is needed. Understanding specific challenges faced by the models would help guide future improvements.

**Questions:**

1.	The paper claims that the main advantage of Bongard-RWR+ lies in its scalability. However, the image selection still relies on manual review, and according to Appendix E.2, as much as 30.2% of images were discarded. This significantly weakens the argument of "automation" in dataset construction. Why not further quantify the impact of manual intervention on the results? Is there any evidence of systematic bias in image selection due to reviewer subjectivity?
2.	The paper only compares InternVL2.5, Qwen2-VL, LLaVA-Next, and MiniCPM-o. However, stronger closed-source models, like GPT-4V and Gemini, have been used in several AVR benchmark tests. Why were these stronger models not included in the comparison?
3.	The authors intentionally maximize intra-side visual diversity (e.g., in Bongard-RWR+/LP) to aid concept identification. However, this strategy risks introducing distracting, spurious concepts. For instance, representing "Vertical" with a tree, a building, and a person may also activate unrelated concepts like "Nature" or "Architecture," potentially confusing the model. The claim that greater diversity makes concepts "easier to identify" is not systematically verified. The authors should analyze if an optimal level of diversity exists by examining whether accuracy on the Bongard-RWR+/LP variants plateaus or even decreases as P (and thus diversity) increases.
4.	Although the paper claims that “models perform poorly with fine-grained concepts,” there is no comparison with human-level benchmarks. The absence of such a comparison makes it hard to interpret whether the models’ performance is truly “bad.”
5.	As shown in Figure 16 and Appendix E, many discarded images were due to unclear structure, background confusion, or perspective errors. This suggests that the main issue might lie in the ambiguity or noise within the images themselves, rather than the models' poor reasoning capabilities. The authors should provide a quantitative analysis of “image quality vs. accuracy” to demonstrate that the problem lies in reasoning rather than perception.

---

> ### Author Response · Authors · 2025-11-20
> **Response to Reviewer sMpD 1/**
>
> > W1: The paper lacks a detailed exploration of how the semi-automated pipeline can be further refined for more reliable image generation. Manual filtering still plays a key role, and automated verification could improve scalability and reduce bias.
>
> Thank you for raising this important point. Automated verification in the free-form visual domain is challenging because, unlike synthetic or symbolic settings, real-world images lack an underlying structure that would allow deterministic assessment of whether an image faithfully expresses an abstract concept. This challenge is precisely what makes BPs valuable: abstract visual rules are instantiated through analogy rather than explicit properties, making full automation inherently difficult. That said, human oversight in dataset construction is standard practice across vision benchmarks – most datasets require human annotation or verification at some stage. In our pipeline, the human role is purely discriminative (limited to filtering generated images) rather than generative, which significantly reduces the manual burden compared to approaches requiring detailed human annotation (e.g. for dense object detection or image segmentation).
>
> We agree that discussing future refinements is valuable, and we outline several promising directions for reducing manual effort and increasing scalability.
>
> **(1) Advancing T2I models.** In our current pipeline, human verification served primarily as a quality control step. In practice, we filtered approximately 30.2% of generated images, which indicates that the majority of outputs were already acceptable. This suggests that the pipeline could operate with minimal human intervention or potentially in a fully automated manner as generative models become more reliable at following complex prompts and avoiding rendering unintended concepts. Our approach is therefore forward-compatible with future advances in generative modeling.
>
> **(2) Reducing demographic bias.** As discussed in Appendices D and E, the demographic skew in our dataset stems from the underlying T2I model. While our pipeline does not amplify these biases, future refinements could mitigate them directly. One approach is to explicitly encode demographic diversity into prompts during both the Augment and Generate stages – for example, by specifying varied gender, racial, or age attributes when appropriate for expressing an abstract concept. Additionally, greater diversity could be achieved by leveraging a broader set of I2T, T2T, and T2I models within the pipeline, thereby reducing reliance on any single generative model.
>
> **(3) Cycle-consistency verification.** In our current pipeline, a T2I model generates images from textual descriptions, and humans verify whether the outputs faithfully depict the intended concept. A natural extension is to introduce an automated cycle-consistency check: after generating an image from a source caption, an I2T model could re-caption the image to produce a reference caption. A T2T model would then compare the source and reference captions to assess semantic alignment. Images whose re-captioned descriptions fail to preserve key elements of the source prompt could be filtered automatically.
>
> **(4) Property-based testing.** A more structured direction for automation is to move beyond free-form captions and operate on explicit image properties. During the Augment stage, the T2T model could be prompted not only to produce a natural language description of the scene, but also output a set of properties that the generated image should satisfy to faithfully express the intended concept. An initial closed vocabulary could include attributes such as size, shape, count, rotation, lighting, camera perspective, color, and distance. Alternatively, the model could generate open-vocabulary, instance-specific properties tailored to each concept. After image generation, an I2T model could verify whether these properties are present in the output. Images that fail to meet the specified property set (or fall below a chosen threshold) could be filtered automatically.
>
> We note, however, that while such model-based approaches could improve scalability, some human oversight will likely remain important in the near term to ensure high data quality. We have highlighted the above points in Appendix D as promising directions for future research.

---

> > ### Author Response · Authors · 2025-11-20
> > **Response to Reviewer sMpD 2/**
> >
> > > W2: Although the experiments cover multiple tasks, the evaluation of fine-grained reasoning is limited. It would benefit from including more diverse models [...] / Q2: The paper only compares InternVL2.5, Qwen2-VL, LLaVA-Next, and MiniCPM-o. However, stronger closed-source models, like GPT-4V and Gemini, have been used in several AVR benchmark tests. Why were these stronger models not included in the comparison?
> >
> > When selecting models for our experiments, we focused on open-source VLMs across four representative model families, which we found sufficient for demonstrating the scope and utility of our benchmark. Rather than expanding the model pool, we prioritized a broad evaluation across task formulations, spanning classification-based concept selection (CS) across difficulty levels K, side prediction from images (I1S, I2S) and text (D1S, D2S), and open-ended text generation (CG). These included the impact of model size (Figure 8a), color (Figure 8b), the relationship between performance on generated vs. real-world data (Figure 9), the effect of few-shot demonstration counts (Table 3), and additional experiments in Appendix C such as evaluating supervised baselines on Bongard-RWR+/TVT and Bongard-RWR+/TVT-Large (Appendix C.4). We believe this breadth of settings offered more insight than adding more model variants.
> >
> > A practical consideration was cost – evaluating large proprietary multimodal models across all settings is substantially more expensive and exceeded the funding available for this project. For this reason, we prioritized models that could be evaluated reproducibly on our in-house computing cluster.
> >
> > That said, we agree with the Reviewer that including state-of-the-art closed-source VLMs would provide a valuable reference point. Based on this feedback, we have secured limited additional resources and initiated a targeted evaluation of three leading proprietary models: Claude Sonnet 4.5, Gemini 2.5 Pro, and GPT 5.1. Given their cost, we restrict this study to a subset of tasks.
> >
> > As a first step, we evaluated these models on Bongard-RWR+ in the I1S task. Their accuracies were 63.4% (Claude Sonnet 4.5), 64.9% (Gemini 2.5 Pro), and 64.8% (GPT 5.1). While these proprietary models perform noticeably better than open VLMs, which all remain near the random-guessing threshold in this task, they still leave substantial room for improvement on the benchmark. We have included these results in Appendix C.11 and will report additional results for these models in other task settings later during the rebuttal phase.
> >
> >
> > > W2: [...] or comparing performance with human-level reasoning. / Q4: Although the paper claims that “models perform poorly with fine-grained concepts,” there is no comparison with human-level benchmarks. The absence of such a comparison makes it hard to interpret whether the models’ performance is truly “bad.”
> >
> > Thank you for this comment. We agree that assessing human performance on the introduced dataset is an important research question. Conducting such a study requires careful experimental design (e.g., task framing, instructions, and evaluation protocols), which is challenging to execute within the limited rebuttal period. We will include such a study in future work.
> >
> > During rebuttal, we analyzed performance across real (RWR 3‑3) and generated (RWR+/L 3‑3) datasets. Across the 54 overlapping concepts, model performance was largely consistent: in 40 concepts (74%) the difference was ≤10 p.p., in 7 concepts (13%) differences were within 10–20 p.p., and in 7 concepts (13%) differences exceeded 20 p.p. Overall, these results indicate that both datasets present a comparable level of difficulty, with models solving similar types of concepts in each case. At the same time, the results on Bongard-RWR (Małkiński et al., 2025; Figure 8) show clear divergences between human and model capabilities: some problems are easy for humans but challenging for models, and vice versa. Since models exhibit closely aligned performance patterns on RWR 3-3 and RWR+/L 3-3, we hypothesize that analogous differences between humans and models may also emerge on Bongard-RWR+. We will present the results of verifying this hypothesis in future work.
> >
> > That said, we believe that despite the lack of human performance assessment, the proposed dataset demonstrates its utility by highlighting model challenges even in relatively simple setups such as I1S. In contrast, humans are capable of tackling much more intricate challenges, such as naming the underlying concepts in natural language – a task where humans scored 65% accuracy on Bongard-RWR, while models struggled to reach comparable accuracy even in the much simpler binary classification setting (Table 2). We therefore believe that the benchmark remains effective for studying visual reasoning in contemporary models, even without a direct human baseline.

---

> > > ### Author Response · Authors · 2025-11-20
> > > **Response to Reviewer sMpD 3/**
> > >
> > > > W3: Lastly, a deeper analysis of errors, especially in fine-grained tasks, is needed. Understanding specific challenges faced by the models would help guide future improvements.
> > >
> > > Thank you for this insightful and important comment. We agree that deeper analysis of model errors is valuable. Already before paper submission we conducted several additional investigations but did not include them in the submitted version due to the paper’s already substantial scope, not to overwhelm its readers. We are happy, however, to revise the manuscript based on the Reviewer’s specific recommendations.
> > >
> > > **Error sources and failure modes**
> > >
> > > In Section 4 (Concept Selection) we noted that concepts such as Contour, Rotation, and Angle pose particular difficulties because they rely on subtle geometric cues or precise spatial relations that current models struggle to capture reliably. To further explore these weaknesses, we deconstructed the concept-group aggregates (Figure 6) and reviewed model performance on specific concepts in the CS task with K=4. For example, InternVL2.5 achieved consistently strong results on all concepts within the Size, Shape, and Branching groups, but struggled with certain concepts from the three more difficult groups mentioned above (Contour, Rotation, and Angle):
> > > * For Angle (69% average accuracy), the model performed particularly poorly on concept #76 *“Long sides concave vs. Long sides convex”* (16%), which requires abstracting beyond surface-level features to recognize a geometric property across diverse real-world scenes.
> > > * For Rotation (73%), performance dropped on concept #17 *“An acute angle directed inward vs. No acute angle directed inward”* (43%), again pointing to difficulties in detecting precise spatial arrangements.
> > > * For Contour (56%), concept #3 *“Outline figures vs. Solid figures”* (22%) was especially challenging, suggesting difficulties in recognizing fine-grained details such as object boundaries.
> > >
> > > These cases require abstraction beyond surface-level object features – precisely the type of reasoning our benchmark aims to measure.
> > >
> > > We also investigated typical model error patterns in the CS task by manually reviewing prediction explanations. Several recurring failure modes emerged:
> > > 1. predicting a concept that applies only to a subset of images on the given side,
> > > 2. confusing the sides images belong to,
> > > 3. predicting a concept for one side while ignoring that it also applies to the opposite side,
> > > 4. relying on shallow surface-level features rather than deep image understanding required to detect fine-grained concepts.
> > >
> > > For example, in the BP presented in the newly added Figure 17a (*“Outline figures vs. Solid figures”*), InternVL2.5 misclassified the concept as *"Convex hull of figure elongated vs. Convex hull of figure compact"*, explaining: *“The images on the left side feature objects or shapes that are more elongated, such as the person's face, the mirror, the stones, and the clock. [...]”*. While faces and mirrors indeed appeared on the left side, the stones and clock were on the right, exemplifying error type (2). Furthermore, the omission of the pie in the description of the left side illustrates error type (1).
> > >
> > > In Figure 17b (*“Large total line length vs. Small total line length”*), Qwen2-VL incorrectly predicted *“Closed lines vs. Open lines"*, justifying with: *“The left side of the image contains natural landscapes [...]”*. Although some left-side images indeed contain landscapes, landscapes also appeared on the right side, exemplifying error type (3). The explanation also reflects type (4) error, as it relies on surface-level features while ignoring the underlying geometric property.
> > >
> > > We included this extended analysis in the newly added Appendix C.10, along with Figure 17 that illustrates these failure modes. In addition, we will release instance-level predictions of all evaluated models to facilitate deeper error analysis in the future.
> > >
> > > **How input domain shapes concept difficulty**
> > >
> > > During rebuttal, in order to supplement the input modality analysis in Appendix C.9, we plotted absolute per-concept difficulty for both I1S and D1S by computing the avg. number of correctly solved instances per concept, across all models.
> > >
> > > [...]

---

> > > > ### Author Response · Authors · 2025-11-20
> > > > **Response to Reviewer sMpD 4/**
> > > >
> > > > [...]
> > > >
> > > > For I1S, each concept was solved in 79-109 instances (out of 200) on average. The relatively narrow range indicates that most concepts remain uniformly challenging for models in the visual domain, which is consistent with their near-random performance. One noteworthy case is “Shading thicker on the right side” (concept #63), which appears among top-5 solved concepts in I1S (109) but is relatively difficult in D1S (92), highlighting the relevance of visual input in identifying fine-grained concepts.
> > > >
> > > > In contrast, D1S exhibits a much broader difficulty range, from 70 to 182 solved instances, reflecting high performance on certain concepts that appear to be easier. Notably, top-5 solved concepts include counting-based ones such as “three parts” or “one figure”, which may be easier to capture in the text domain.
> > > >
> > > > We have included this extended analysis in Appendix C.9, with visualization presented in Figure 16.
> > > >
> > > > > Q1: The paper claims that the main advantage of Bongard-RWR+ lies in its scalability. However, the image selection still relies on manual review, and according to Appendix E.2, as much as 30.2% of images were discarded. This significantly weakens the argument of "automation" in dataset construction. Why not further quantify the impact of manual intervention on the results? Is there any evidence of systematic bias in image selection due to reviewer subjectivity?
> > > >
> > > > Our filtering strategy was intentionally conservative, prioritizing precision over recall to ensure that each generated image faithfully expressed the intended concept. As detailed in Appendix E.2, the human verification step focused on two objective criteria: (1) whether the image adhered to the concept of its own side of the matrix, and (2) whether it did not adhere to the concept of the opposing side. Importantly, human reviewers did not evaluate the aesthetic quality or subjective appeal of images. The examples in Figure 20 illustrate this objectivity:
> > > > * (a) the flower has six petals and therefore matches neither side;
> > > > * (b) the chain extends beyond the image boundary, making the concept ambiguous;
> > > > * (c) a single white line appears in front of the car but two yellow lines appear in the background, violating consistency;
> > > > * (d) the vessel appears circular, implying convexity, but the 3D structure is actually concave.
> > > >
> > > > These cases represent clear conceptual discrepancies, not subjective judgments.
> > > >
> > > > That said, we appreciate the Reviewer’s insightful suggestion to quantify the impact of manual intervention, and we agree this is an important dimension for understanding the role of human verification. In addition to the research directions outlined earlier for reducing or automating manual filtering, the Reviewer’s comment motivated us to conduct a targeted experiment to measure its effect on model performance.
> > > >
> > > > Specifically, during rebuttal, we constructed Bongard-RWR+/L2-Impure, a variant of Bongard-RWR+/L2 in which each BP side contains one image that passed human filtering and one image that was rejected. This yields BP instances where a subset of images violate the intended concept, producing ambiguity that would arise if human verification was removed.
> > > >
> > > > We evaluated all four models used in the main paper on this impure dataset in the CS task for K = 2, 4, 8, 16. Across K levels, we observe mean absolute accuracy drops of -3.85, -8.60, -5.73, and -4.60 for MiniCPM-o 2.6 8B, InternVL2.5 78B, Qwen2-VL 72B, and LLaVA-Next 110B, resp. This consistent reduction in accuracy supports our hypothesis that including unfiltered images degrades dataset quality and harms model performance, thus demonstrating the human review step meaningfully improves data quality.
> > > >
> > > > We have included these results in Section 4 (“Noisy images hinder concept recognition”) and Appendix C.5.

---

> > > > > ### Author Response · Authors · 2025-11-20
> > > > > **Response to Reviewer sMpD 5/**
> > > > >
> > > > > > Q3: The authors intentionally maximize intra-side visual diversity (e.g., in Bongard-RWR+/LP) to aid concept identification. However, this strategy risks introducing distracting, spurious concepts. For instance, representing "Vertical" with a tree, a building, and a person may also activate unrelated concepts like "Nature" or "Architecture," potentially confusing the model. The claim that greater diversity makes concepts "easier to identify" is not systematically verified. The authors should analyze if an optimal level of diversity exists by examining whether accuracy on the Bongard-RWR+/LP variants plateaus or even decreases as P (and thus diversity) increases.
> > > > >
> > > > > Thank you for raising this important point. Our experiments already include two complementary analyses that address this concern.
> > > > >
> > > > > The experiment “Do models learn from demonstrations?“ (Section 4) was specifically designed to test whether increasing P, and thus increasing intra-side diversity, facilitates concept recognition. In I1S (Table 3a), both InternVL2.5 and Qwen2-VL show a mild upward trend with increasing P, with their highest accuracy at P = 6 (+4 p.p. for InternVL2.5 and +6 p.p. for Qwen2-VL compared to P=2). In D1S (Table 3b), accuracy similarly improves with P for all models except MiniCPM-o2.6. In particular, DeepSeek-R1 improves by 14 p.p. from P= 2 to P=6. Appendix C.3 further confirms this trend for the CS task: model performance generally increases with more demonstrations, especially for stronger models such as InternVL2.5, which achieves its best result at P = 6 (+16 p.p. vs. P=2). Overall, these results demonstrate that models benefit from additional examples across tasks and modalities, supporting our claim that visual diversity aids concept recognition.
> > > > >
> > > > > During rebuttal, in order to address the Reviewer’s request for a more systematic analysis, we additionally examined diversity from a complementary angle – not by varying P, but by measuring the intrinsic visual diversity of  Bongard-RWR+ (with fixed P=6). We computed the maximum cosine similarity between ViT-L/14 image embeddings within each side of a BP, then took the negative of the larger value across both sides as the BP’s diversity score. BPs were partitioned into three equally sized bins (Low, Medium, High diversity), and we analyzed model performance in the most challenging variant of the Concept Selection task (CS, K=16).
> > > > >
> > > > > Across models, performance is lowest on Low-diversity matrices (e.g., InternVL2.5 78B scores 0.50), where images are similar and depict the concept in similar ways, making it harder to identify. In contrast, performance is highest on High-diversity matrices (e.g., InternVL2.5 78B scores 0.62, +12 p.p. vs. Low-diversity), where the concept is expressed through more varied visual content, making it more apparent. This further strengthens our claim that diversity helps disambiguate abstract concepts.
> > > > >
> > > > > We have included these additional results in Section 4 (“Visual diversity facilitates concept recognition”), along with the new Figure 10. We thank the Reviewer again for suggesting this insightful direction, which meaningfully strengthens our analysis.
> > > > >
> > > > > | Model            | Low  | Medium | High |
> > > > > | ---------------- | ---- | ------ | ---- |
> > > > > | InternVL2.5 78B  | 0.50 |   0.58 | 0.62 |
> > > > > | Qwen2-VL 72B     | 0.32 |   0.38 | 0.43 |
> > > > > | LLaVA-Next 110B  | 0.16 |   0.20 | 0.21 |
> > > > > | MiniCPM-o 2.6 8B | 0.17 |   0.20 | 0.20 |

---

> > > > > > ### Author Response · Authors · 2025-11-20
> > > > > > **Response to Reviewer sMpD 6/6**
> > > > > >
> > > > > > > Q5: As shown in Figure 16 and Appendix E, many discarded images were due to unclear structure, background confusion, or perspective errors. This suggests that the main issue might lie in the ambiguity or noise within the images themselves, rather than the models' poor reasoning capabilities. The authors should provide a quantitative analysis of “image quality vs. accuracy” to demonstrate that the problem lies in reasoning rather than perception.
> > > > > >
> > > > > > We agree with the Reviewer that analyzing the relationship between image quality and model accuracy could provide valuable insights. However, performing such an analysis is unfortunately not feasible with the current data. In our pipeline, the human filtering step was strictly binary: an image was accepted only if it (1) correctly represented the intended concept on its side of the matrix and (2) did not express the opposite-side concept. As detailed in Appendix E.2, two expert annotators independently reviewed all images, and disagreements were resolved through discussion, prioritizing quality over scale. Crucially, no detailed quality annotations were collected during this process, which makes the suggested analysis not feasible.
> > > > > >
> > > > > > That said, several existing results in the paper already provide indirect evidence that the difficulty of Bongard-RWR+ cannot be attributed solely to image quality issues. First, the already discussed “Do models learn from demonstrations?” experiment (Section 4 and Appendix C.3) shows that providing additional images per side consistently improves accuracy across models, tasks and modalities. If the images introduced with increasing P were primarily noisy or ambiguous, adding more images would not systematically increase performance. Second, the D1S experiments demonstrate that inserting an explicit image captioning step, requiring the model to start with a perception-oriented subtask, improves performance. For example, InternVL2.5 improves from 0.55 on Bongard-RWR+/L6 in I1S (Table 3a) to 0.67 in D1S (Table 3b), an increase of 12 p.p. This indicates that models are capable of extracting meaningful information from the generated images.
> > > > > >
> > > > > > On a more general note, we’d like to challenge the notion that perception and reasoning can be cleanly separated in VLMs. As Hofstadter argues in his work on analogy-making (Hofstadter, 1995), these processes are deeply intertwined and their boundary is unclear.
> > > > > >
> > > > > > Crucially, perception already involves pattern matching and hypothesis testing. When a human examines BP images, he/she doesn’t neutrally perceive them first and then reason separately. Extracting meaningful visual features requires forming hypotheses about what distinguishes the two sides, and these hypotheses reshape what the person notices as important.
> > > > > >
> > > > > > While VLMs do create explicit image representations through their vision encoders, this doesn’t necessarily mean perception and reasoning are separated. The representations themselves may encode different features that facilitate various downstream tasks. In transformer-based models, attention mechanisms allow later reasoning steps to selectively reweigh and reinterpret visual tokens, essentially “re-perceiving” the images in light of tentative hypotheses. The model may initially attend to color or texture, then shift attention to shape or spatial relationships as it refines its understanding of the underlying concept.
> > > > > >
> > > > > > If perception and reasoning work together this way, where understanding shapes which features get noticed, which in turn shapes understanding, then it becomes difficult to say whether errors come from perception or reasoning. This is precisely why we believe Bongard-RWR+ is valuable: it may help evaluate how well future models handle tasks where perception and reasoning are intertwined rather than separate.
> > > > > >
> > > > > > Douglas R Hofstadter. “Fluid concepts and creative analogies: Computer models of the fundamental mechanisms of thought.” Basic books, 1995.

---

> > > > > > > ### Author Response · Authors · 2025-12-02
> > > > > > > **Follow-up response to Reviewer sMpD #W2**
> > > > > > >
> > > > > > > Due to limited time and resources, in our first response we reported the evaluation of selected proprietary models on I1S task and announced subsequent presentation of further results. In this response, we extend our analysis to two additional tasks: D1S (Description-to-Side) and CS (Concept Selection) with $K=16$. All three proprietary models outperform open ones, particularly in the more challenging CS setting, where Gemini 2.5 Pro reaches 81.8% compared to 57% for the best open model (InternVL2.5 78B). Performance is also strong in D1S, with accuracies up to 69.9% (Gemini 2.5 Pro) versus 58% for the best open model (Qwen2-VL 72B-Instruct). Despite these gains, none of the models come close to saturation on any task, reinforcing that Bongard-RWR+ remains challenging even for state-of-the-art proprietary models.
> > > > > > >
> > > > > > > | Model              |       I1S  |        D1S |    CS K=16 |
> > > > > > > | ------------------| ----------| ----------| ----------|
> > > > > > > | Claude Sonnet 4.5  |      63.37 |      66.61 |      75.87 |
> > > > > > > | Gemini 2.5 Pro     |      64.70 |      69.90 |      81.83 |
> > > > > > > | GPT 5.1            |      64.70 |      67.37 |      80.67 |
> > > > > > >
> > > > > > > Proprietary models present particularly strong and stable performance on the Shape, Size, and Count concept groups in CS, D1S, and crucially the vision-intensive I1S setting. Additionally, fine-grained spatial concepts such as “triangle inside the circle” vs. “circle inside the triangle” appear among the top-performing ones for proprietary models but not for the open models, further underscoring their advantage in spatial reasoning.
> > > > > > >
> > > > > > > |          | Gemini 2.5  Pro: CS | Gemini 2.5  Pro: D1S | Gemini 2.5  Pro: I1S | Claude Sonnet 4.5: CS | Claude Sonnet 4.5: D1S | Claude Sonnet 4.5: I1S | GPT-5.1: CS | GPT-5.1: D1S | GPT-5.1: I1S |
> > > > > > > |------------|-------|-------|-------|-------|-------|-------|-------|-------|-------|
> > > > > > > | Angle      | 0.858 | 0.709 | 0.675 | 0.675 | 0.580 | 0.599 | 0.755 | 0.721 | 0.688 |
> > > > > > > | Branching  | 0.965 | 0.485 | 0.470 | **0.970** | 0.542 | 0.465 | 0.975 | 0.502 | 0.512 |
> > > > > > > | Contour    | 0.565 | 0.614 | 0.512 | 0.510 | 0.611 | 0.574 | 0.557 | 0.520 | 0.484 |
> > > > > > > | Count      | 0.838 | 0.787 | 0.758 | 0.739 | 0.753 | **0.757** | 0.766 | 0.787 | 0.759 |
> > > > > > > | Position   | 0.712 | 0.571 | 0.513 | 0.728 | 0.556 | 0.523 | 0.703 | 0.539 | 0.506 |
> > > > > > > | Rotation   | 0.830 | 0.683 | 0.631 | 0.706 | 0.622 | 0.580 | 0.858 | 0.616 | 0.643 |
> > > > > > > | Shape      | **0.997** | **0.811** | 0.778 | 0.898 | **0.785** | 0.732 | **0.990** | **0.811** | 0.783 |
> > > > > > > | Similarity | 0.796 | 0.780 | 0.642 | 0.768 | 0.741 | 0.661 | 0.816 | 0.725 | 0.623 |
> > > > > > > | Size       | 0.880 | 0.787 | **0.786** | 0.868 | 0.746 | 0.736 | 0.953 | 0.776 | **0.801** |
> > > > > > >
> > > > > > > Experiments concerning proprietary models are discussed in the revised Appendix C.11, with results presented in the newly introduced Figure 18, and extended Tables 14 and 15.

---

### Official Review · Reviewer_kdgb · 2025-10-29

**Soundness:** 3
**Presentation:** 3
**Contribution:** 3
**Rating:** 8
**Confidence:** 5

**Summary:**

The paper introduces Bongard-RWR+, a large-scale benchmark (5,400 matrices) for abstract visual reasoning (AVR) that preserves the *fine-grained concepts* of classic Bongard Problems while using real-world-like images synthesized by a VLM-assisted pipeline. Starting from the concepts in Bongard-RWR, the authors (i) describe exemplar images via an I2T model, (ii) augment descriptions with a T2T model, (iii) render images with a T2I model, and (iv) perform human verification. The benchmark supports six task formulations—binary and paired image(s)/description(s)-to-side(s), multiclass concept selection, and free-form concept generation—and includes grayscale and varying-shots variants to analyze the role of color and number of demonstrations.

Across 4 strong open VLMs, results reveal a consistent pattern: performance is decent on coarse concepts (e.g., shape/size) but drops on fine-grained cues (e.g., contour, angle, rotation). Caption-then-reason pipelines help, and concept-selection accuracy scales with model size, but free-form concept generation remains weak. The authors further test functional equivalence between generated and real images and show broadly similar scaling trends, supporting the dataset’s validity.

**Strengths:**

- Clever use of I2T/T2T/T2I with human vetting yields large, diverse matrices that preserve fine-grained Bongard concepts rather than only coarse, object-level cues.
- Binary/paired side assignments, concept selection, and free-form concept generation give a multifaceted picture of AVR.
- Grayscale shows color is a distractor; more demonstrations help certain models; generated vs. real yields similar trends, supporting external validity.
- Per-concept-group breakdowns (size/shape vs. contour/angle/rotation) isolate where VLMs fail, informing future model design.
- A simple similarity classifier is competitive in image-based tasks, underscoring that current VLM prompting often fails to internalize the concept—a useful, humbling baseline.

**Weaknesses:**

- Caption/augmentation quality inherits biases and blind spots of the I2T/T2T models; although human verification mitigates this, a quantitative *masking* or *counterfactual* stress test of pipeline robustness would help.
- BLEU/ROUGE/CIDEr/BERTScore only weakly capture conceptual correctness and fine-grained relations; a concept-aware rubric (or human evaluation on a subset) would better reflect success/failure in CG.
- Most core experiments use a small pool of open VLMs; including stronger closed models (or stronger open ones as they appear) would calibrate the difficulty spectrum more completely.
- While the release mitigates this, adding inter-annotator agreement for the acceptance filter directly in the main paper (not just appendix) would strengthen transparency.

**Questions:**

1. Have you considered automatic checkers that evaluate whether generated concepts *logically partition* the two sides (e.g., via structured parsers / learned validators), beyond surface n-gram overlap?
2. What fraction of generated candidates fail human vetting per concept family (e.g., angle vs. shape)? Any systematic failure patterns in the T2I step?
3. How sensitive are results to chain-of-thought or *explicit analogy* prompting in I1S/I2S (e.g., “explain the difference, then decide”) and to few-shot textual exemplars?
4. Could the benchmark be extended to interactive *concept discovery* (iteratively request descriptions or crops) and do models improve with interaction?
5. Beyond the bias audit, do you plan to ship per-matrix *concept tags* and *hardness annotations* so researchers can target specific failure modes?
6. The manuscript shows “ICLR 2025” in the PDF header but the submission targets ICLR 2026.

---

> ### Author Response · Authors · 2025-11-20
> **Response to Reviewer kdgb 1/**
>
> > W1: Caption/augmentation quality inherits biases and blind spots of the I2T/T2T models; although human verification mitigates this, a quantitative masking or counterfactual stress test of pipeline robustness would help.
>
> Thank you for this insightful suggestion. Automated methods for assessing the robustness of the I2T and T2T models connect to the broader challenge of detecting hallucinations, a topic that has received significant recent attention and remains fundamentally difficult (Li et al., 2023; Huang et al., 2025). Below, we restate the Reviewer’s suggestions in a way we understand them, and discuss their feasibility in our setting.
>
> A quantitative masking robustness test for the I2T model could involve perturbing the input image (e.g., masking different regions), generating captions for each perturbed variant, and measuring their consistency. Divergences across captions might indicate instability or hallucination tendencies of the I2T model, thereby providing a signal that a particular caption may not faithfully reflect the visual content. However, in our setup this approach poses a core difficulty, as concepts in Bongard-RWR+ are fine-grained and frequently depend on subtle visual details (e.g., exact flower petal counts, convexity vs. concavity). Masking even small regions can alter the underlying concept itself, which makes it difficult to distinguish robustness issues from legitimate semantic changes caused by masking. That said, we remain open to alternative masking approaches that might avoid this pitfall.
>
> Regarding a counterfactual stress test, we see it as a promising direction. One could employ a T2T model in an LLM-as-a-judge framework to check whether an augmented caption (1) aligns with the original caption and (2) does not align with the opposite-side concept. Such automatic checks could flag problematic captions before image generation, reducing the likelihood of producing images that later fail human verification. This form of automated safety filter aligns closely with two research directions now included in Appendix D. Namely:
>
> **(3) Cycle-consistency verification.** In our current pipeline, a T2I model generates images from textual descriptions, and humans verify whether the outputs faithfully depict the intended concept. A natural extension is to introduce an automated cycle-consistency check: after generating an image from a source caption, an I2T model could re-caption the image to produce a reference caption. A T2T model would then compare the source and reference captions to assess semantic alignment. Images whose re-captioned descriptions fail to preserve key elements of the source prompt could be filtered automatically.
>
> **(4) Property-based testing.** A more structured direction for automation is to move beyond free-form captions and operate on explicit image properties. During the Augment stage, the T2T model could be prompted not only to produce a natural language description of the scene, but also output a set of properties that the generated image should satisfy to faithfully express the intended concept. An initial closed vocabulary could include attributes such as size, shape, count, rotation, lighting, camera perspective, color, and distance. Alternatively, the model could generate open-vocabulary, instance-specific properties tailored to each concept. After image generation, an I2T model could verify whether these properties are present in the output. Images that fail to meet the specified property set (or fall below a chosen threshold) could be filtered automatically.
>
> We thank the Reviewer again for this thought-provoking idea, prompting us to articulate how such methods could increase pipeline robustness and reduce dependence on post-hoc manual filtering.
>
> Li, Yifan, et al. "Evaluating Object Hallucination in Large Vision-Language Models." Conference on Empirical Methods in Natural Language Processing. 2023.
>
> Huang, Lei, et al. "A Survey on Hallucination in Large Language Models: Principles, Taxonomy, Challenges, and Open Questions." ACM Transactions on Information Systems (2025).
>
> > W2: BLEU/ROUGE/CIDEr/BERTScore only weakly capture conceptual correctness and fine-grained relations; a concept-aware rubric (or human evaluation on a subset) would better reflect success/failure in CG.
>
> We agree with the Reviewer that standard NLP metrics primarily rely on lexical similarity and therefore only weakly capture conceptual correctness, particularly in a setting like Bongard-RWR+ involving fine-grained concept understanding. As noted in Section 4 and Appendix C.6, we use these metrics only as supplementary indicators for the CG task.
>
> [...]

---

> > ### Author Response · Authors · 2025-11-20
> > **Response to Reviewer kdgb 2/**
> >
> > [...]
> >
> > A more principled approach could involve an LLM-as-a-judge evaluation pipeline calibrated with human annotations, however, it may also bring its own challenges, e.g. in faithfully capturing intricate details of the concepts and mitigating potential biases introduced by the evaluation model. To our knowledge, there is currently no established metric that accurately captures fine-grained concepts in free-form textual responses, which underscores a broader gap in the field.
> >
> > For this reason, we focus primarily on classification-based tasks, where performance can be measured objectively with simple, interpretable metrics such as accuracy. Notably, even in these simpler discriminative settings, SOTA models still exhibit clear reasoning limitations. This further supports our decision to treat CG as a secondary approach to study model capabilities rather than a core quantitative benchmark.
> >
> > We also believe that Bongard-RWR+ naturally motivates new research on concept-aware evaluation metrics, and we view it as a promising research direction enabled by our dataset.
> >
> > Nevertheless, motivated by the Reviewer’s comment, during rebuttal we explored additional automatic evaluation strategies. Specifically, we evaluated two alternative approaches: (1) sentence-level cosine similarity (SIM), where we computed normalized sentence embeddings using MiniLM (HuggingFace, 2021; Wang et al., 2020a), calculated pairwise cosine similarities between embeddings of the ground-truth and predicted concepts, and averaged the results; and (2) bag-of-words reward (BoW), where we computed the overlap of word frequencies in the ground-truth and predicted concepts, normalized by the total vocabulary size. Both metrics were normalized to the [0, 1] range.
> >
> > To evaluate these metrics without selecting arbitrary thresholds, we computed their discriminative power using area under the curve (AUC). For each metric, we varied the threshold from 0.0 to 1.0 with a step of 0.05, classified predictions as correct or incorrect based on whether they exceeded the threshold, and computed the area under the resulting curve (SIM-AUC and BoW-AUC). Higher AUC indicates better separation between correct and incorrect predictions. We applied both metrics to model-generated concepts from I1S and D1S tasks (cf. Prompts 6 – 8).
> >
> > Accuracy and SIM-AUC in D1S show strong correlation: Qwen2-VL ranks first in both metrics (0.581 and 0.227), InternVL2.5 ranks second (0.569 and 0.218), and LLaVA-Next outperforms MiniCPM-o 2.6 in both (0.536 vs. 0.514, and 0.172 vs. 0.161). For I1S, correlation is difficult to assess as all models achieve near-random accuracy. However, SIM-AUC in I1S shows strong correlation with CS performance (K=16): InternVL2.5 leads in both (0.569 and 0.177), Qwen2-VL ranks second (0.378 and 0.153), LLaVA-Next third (0.194 and 0.081), and MiniCPM-o 2.6 fourth (0.191 and 0.060). Notably, SIM-AUC maintains consistent ranking across all models, indicating that this embedding-based similarity metric reliably captures discriminative performance. Per-concept analysis across 54 concepts further confirms this relationship: SIM-AUC in D1S correlates with CS accuracy (K=16) with Pearson ρ = 0.637 (p < 0.001) for InternVL2.5, the strongest correlation observed.
> >
> > Accuracy and BoW-AUC in D1S show similar correlation patterns: Qwen2-VL ranks first in both metrics (0.581 and 0.081) and InternVL2.5 ranks second (0.569 and 0.071). For I1S, BoW-AUC also correlates with CS performance (K=16): InternVL2.5 leads in both (0.569 and 0.051), Qwen2-VL ranks second (0.378 and 0.043), and LLaVA-Next / MiniCPM-o 2.6 show comparable scores. Per-concept analysis shows that BoW-AUC in D1S correlates with CS accuracy (K=16) with Pearson ρ = 0.532 (p < 0.001) for InternVL2.5. This suggests that both SIM-AUC and BoW-AUC can provide useful signals for evaluating free-form concept generation in a way that reflects models’ discriminative reasoning capabilities.
> >
> > Nevertheless, both metrics have important limitations, as they disregard word order and context, making them insufficient for capturing the fine-grained differences required in our benchmark. Developing more nuanced concept-aware evaluation methods remains an important direction for future work.
> >
> > | Model            | CS, K=16 (accuracy) | D1S (accuracy) | I1S (accuracy) | D1S (SIM AUC) | I1S (SIM AUC) | D1S (BoW AUC) | I1S (BoW AUC) |
> > |:-----------------|---------:|------:|------:|------:|------:|------:|------:|
> > | InternVL2.5 78B  |    0.569 | 0.569 | 0.499 | 0.218 | 0.177 | 0.071 | 0.051 |
> > | Qwen2-VL 72B     |    0.378 | 0.581 | 0.492 | 0.227 | 0.153 | 0.081 | 0.043 |
> > | LLaVA-Next 110B  |    0.194 | 0.536 | 0.501 | 0.172 | 0.081 | 0.052 | 0.011 |
> > | MiniCPM-o 2.6 8B |    0.191 | 0.514 | 0.482 | 0.161 | 0.060 | 0.062 | 0.013 |

---

> > > ### Author Response · Authors · 2025-11-20
> > > **Response to Reviewer kdgb 3/**
> > >
> > > > W3: Most core experiments use a small pool of open VLMs; including stronger closed models (or stronger open ones as they appear) would calibrate the difficulty spectrum more completely.
> > >
> > > When selecting models for our experiments, we focused on open-source VLMs across four representative model families, which we found sufficient for demonstrating the scope and utility of our benchmark. Rather than expanding the model pool, we prioritized a broad evaluation across task formulations, spanning classification-based concept selection (CS) across difficulty levels K, side prediction from images (I1S, I2S) and text (D1S, D2S), and open-ended text generation (CG). These included the impact of model size (Figure 8a), color (Figure 8b), the relationship between performance on generated vs. real-world data (Figure 9), the effect of few-shot demonstration counts (Table 3), and additional experiments in Appendix C such as evaluating supervised baselines on Bongard-RWR+/TVT and Bongard-RWR+/TVT-Large (Appendix C.4). We believe that this breadth of settings offered more insight than adding more model variants.
> > >
> > > A practical consideration was cost – evaluating large proprietary multimodal models across all settings is substantially more expensive and exceeded the funding available for this project. For this reason, we prioritized models that could be evaluated reproducibly on our in-house computing cluster.
> > >
> > > That said, we agree with the Reviewer that including state-of-the-art closed-source VLMs would provide a valuable reference point. Based on this feedback, we have secured limited additional resources and initiated a targeted evaluation of three leading proprietary models: Claude Sonnet 4.5, Gemini 2.5 Pro, and GPT 5.1. Given their cost, we restrict this study to a subset of tasks.
> > >
> > > As a first step, we evaluated these models on Bongard-RWR+ in the I1S task. Their accuracies were 63.4% (Claude Sonnet 4.5), 64.9% (Gemini 2.5 Pro), and 64.8% (GPT 5.1). While these proprietary models perform noticeably better than open VLMs, which all remain near the random-guessing threshold in this task, they still leave substantial room for improvement on the benchmark. We have included these results in Appendix C.11 and will report additional results in other task settings later during the rebuttal phase.
> > >
> > >
> > > > W4: While the release mitigates this, adding inter-annotator agreement for the acceptance filter directly in the main paper (not just appendix) would strengthen transparency.
> > >
> > > We agree with the Reviewer that reporting inter-annotator agreement is important. Following this suggestion, we expanded the Ethics statement in Section 5 by adding the following information.
> > >
> > > *“To mitigate this, each generated image was independently reviewed by two expert annotators, who verified whether the image faithfully expressed the intended concept. Inter-annotator agreement reached Cohen's $\kappa = 0.64$ (substantial agreement), with annotators agreeing on the same label in 77.3% of cases. Disagreements were resolved through discussion, and images lacking consensus were discarded. Overall, 30.2% of generated images were excluded through this filtering process.”*

---

> > > > ### Author Response · Authors · 2025-11-20
> > > > **Response to Reviewer kdgb 4/**
> > > >
> > > > > Q1: Have you considered automatic checkers that evaluate whether generated concepts logically partition the two sides (e.g., via structured parsers / learned validators), beyond surface n-gram overlap?
> > > >
> > > > Thank you for this comment. We agree that such automatic checkers would be valuable. As mentioned in our response to #W2, current metrics are insufficient, and developing metrics that go beyond surface n-gram overlap represents a fundamental research challenge.
> > > >
> > > > Some prior work has explored relevant approaches. Sonwane et al. (2021) applied program synthesis to BPs using DreamCoder, a system that learns hierarchical abstractions from simple drawing primitives (e.g., “move forward”, “pen up/down”) to construct programs for more complex figures (e.g., “draw a triangle”). Their pipeline synthesized drawing programs for images, converted them to symbolic representations, and learned logical rules separating the two sides – effectively implementing an automatic checker.
> > > >
> > > > However, their system solved only 8 of 14 problems, all from the original synthetic black-and-white Bongard dataset (Bongard, 1970). Scaling this approach to real-world or generated images as in Bongard-RWR+ would require addressing several fundamental challenges: (1) constructing a set of drawing primitives suitable for real-world domains, (2) defining relevant visual attributes in a tractable search space, and (3) handling visual ambiguity where objects can be rendered in multiple valid ways. Implementing such an approach would require developing a novel domain-specific language (DSL) capable of handling the complexities of real-world images. These are all challenging and open research directions.
> > > >
> > > > The difficulty of creating such automatic checkers connects to our earlier discussion in response to #W1. The model-based approaches we outlined in“(3) Cycle-consistency verification” and “(4) Property-based testing” (now in Appendix D) represent the most promising near-term directions we identified for making progress on this challenge.
> > > >
> > > >
> > > > > Q2: What fraction of generated candidates fail human vetting per concept family (e.g., angle vs. shape)? Any systematic failure patterns in the T2I step?
> > > >
> > > > Thank you for this comment, which motivated us to look for patterns in generation failures. The table below reports the percentage of generated images discarded during human filtering, broken down by concept group. Rotation concepts show the highest rejection rate (39%), followed by Contour and Count (both 30%), indicating these are most challenging for the T2I model to render faithfully. In contrast, Branching and Similarity have the lowest rejection rates (14%), suggesting they are easier for the pipeline to capture reliably. We have added these observations to Appendix E.2, with the results presented in the newly added Table 16.
> > > >
> > > > | Concept Group | Rejection Rate |
> > > > | --------------|--------------- |
> > > > | Rotation      |            39% |
> > > > | Contour       |            30% |
> > > > | Count         |            30% |
> > > > | Shape         |            27% |
> > > > | Position      |            26% |
> > > > | Size          |            19% |
> > > > | Angle         |            19% |
> > > > | Branching     |            14% |
> > > > | Similarity    |            14% |
> > > >
> > > >
> > > >
> > > > > Q3: How sensitive are results to chain-of-thought or explicit analogy prompting in I1S/I2S (e.g., “explain the difference, then decide”) and to few-shot textual exemplars?
> > > >
> > > > We thank the Reviewer for the suggestion to examine model sensitivity to prompting strategies. All of our tasks already include two few-shot textual demonstrations (see “EXAMPLE START / END” in Prompts 4, 6, 7, 8, 10, 11, 12). Moreover, model responses are structured to first predict the underlying concept, then provide an explanation, and finally output the classification label, which aligns with an “explain the difference, then decide” prompting style.
> > > >
> > > > Motivated by this question, we conducted two additional prompt ablations in the I1S task:
> > > > Step-by-step reasoning: We added the instruction “Let’s think step by step.” and updated the few-shot examples accordingly. The response format was also modified so that the model first produces a reasoning trace, then predicts the concept, and finally solves the task.
> > > > Explicit analogy prompting: We replaced the instruction with “First explain the difference between the images, then decide on the final answer.”, again updating the few-shot examples to reflect this reasoning pattern.
> > > > All four models are currently being evaluated under these variants. Experiments are in progress, and we will report the complete results later during the rebuttal phase.

---

> > > > > ### Author Response · Authors · 2025-11-20
> > > > > **Response to Reviewer kdgb 5/5**
> > > > >
> > > > > > Q4: Could the benchmark be extended to interactive concept discovery (iteratively request descriptions or crops) and do models improve with interaction?
> > > > >
> > > > > We appreciate this thought-provoking question. As part of our data release, we provide both complete BP matrix images and individual image panels, facilitating diverse solution strategies beyond processing the entire matrix at once. Models could iteratively describe images, contrast images from opposing sides, analyze subsets within each side, formulate and test hypotheses about one side against the other, and dynamically select the next operation to perform.
> > > > >
> > > > > This interactive approach, where a model formulates hypotheses and gathers evidence to accept or reject them, relates to prior work on multi-agent debate (Du et al., 2023; Chen et al., 2023; Liang et al., 2024). These methods have shown promise in improving factuality and reasoning in language models but have not yet been evaluated in the AVR domain. We view this as a particularly promising direction for future work.
> > > > >
> > > > > Notably, the recently announced ARC-AGI-3 benchmark (Chollet et al., 2025) was specifically designed to measure AI systems’ capabilities in interactive environments, underscoring growing interest in this paradigm. We hope our data release encourages the research community to explore interactive concept discovery approaches for visual reasoning tasks. We thank the Reviewer for suggesting this valuable research direction and will highlight it in Appendix XX.
> > > > >
> > > > > Du, Yilun, et al. "Improving Factuality and Reasoning in Language Models through Multiagent Debate." International Conference on Machine Learning. 2023.
> > > > >
> > > > > Chen, Weize, et al. "AgentVerse: Facilitating Multi-Agent Collaboration and Exploring Emergent Behaviors." International Conference on Learning Representations. 2023.
> > > > >
> > > > > Liang, Tian, et al. "Encouraging Divergent Thinking in Large Language Models through Multi-Agent Debate." Conference on Empirical Methods in Natural Language Processing. 2024.
> > > > >
> > > > > Chollet, François, et al. "ARC-AGI-3: Interactive Reasoning Benchmark." https://arcprize.org/arc-agi/3/ (Accessed: 2025-11-17).
> > > > >
> > > > >
> > > > > > Q5: Beyond the bias audit, do you plan to ship per-matrix concept tags and hardness annotations so researchers can target specific failure modes?
> > > > >
> > > > > In the public release on the HuggingFace Datasets platform, we have extended the dataset metadata with instance-level annotations including the problem concept, its concept group, and the image diversity metric used in our response to Reviewer sMpD’s (question #Q3). In addition, we will release instance-level model responses for all considered tasks to facilitate analysis of specific failure modes. Preparing these annotations requires additional time and careful validation to ensure consistency, but they will be included along with the final version of the paper.
> > > > >
> > > > >
> > > > > > Q6: The manuscript shows “ICLR 2025” in the PDF header but the submission targets ICLR 2026.
> > > > >
> > > > > Thank you for spotting this. We have corrected the header in the revision.

---

> > > > > > ### Author Response · Authors · 2025-12-02
> > > > > > **Follow-up response to Reviewer kdgb #W3**
> > > > > >
> > > > > > Due to limited time and resources, in our first response we reported the evaluation of selected proprietary models on I1S task and announced subsequent presentation of further results. In this response, we extend our analysis to two additional tasks: D1S (Description-to-Side) and CS (Concept Selection) with $K=16$. All three proprietary models outperform open ones, particularly in the more challenging CS setting, where Gemini 2.5 Pro reaches 81.8% compared to 57% for the best open model (InternVL2.5 78B). Performance is also strong in D1S, with accuracies up to 69.9% (Gemini 2.5 Pro) versus 58% for the best open model (Qwen2-VL 72B-Instruct). Despite these gains, none of the models come close to saturation on any task, reinforcing that Bongard-RWR+ remains challenging even for state-of-the-art proprietary models.
> > > > > >
> > > > > > | Model              |       I1S  |        D1S |    CS K=16 |
> > > > > > | ------------------| ----------| ----------| ----------|
> > > > > > | Claude Sonnet 4.5  |      63.37 |      66.61 |      75.87 |
> > > > > > | Gemini 2.5 Pro     |      64.70 |      69.90 |      81.83 |
> > > > > > | GPT 5.1            |      64.70 |      67.37 |      80.67 |
> > > > > >
> > > > > > Proprietary models present particularly strong and stable performance on the Shape, Size, and Count concept groups in CS, D1S, and crucially the vision-intensive I1S setting. Additionally, fine-grained spatial concepts such as “triangle inside the circle” vs. “circle inside the triangle” appear among the top-performing ones for proprietary models but not for the open models, further underscoring their advantage in spatial reasoning.
> > > > > >
> > > > > > |          | Gemini 2.5  Pro: CS | Gemini 2.5  Pro: D1S | Gemini 2.5  Pro: I1S | Claude Sonnet 4.5: CS | Claude Sonnet 4.5: D1S | Claude Sonnet 4.5: I1S | GPT-5.1: CS | GPT-5.1: D1S | GPT-5.1: I1S |
> > > > > > |------------|-------|-------|-------|-------|-------|-------|-------|-------|-------|
> > > > > > | Angle      | 0.858 | 0.709 | 0.675 | 0.675 | 0.580 | 0.599 | 0.755 | 0.721 | 0.688 |
> > > > > > | Branching  | 0.965 | 0.485 | 0.470 | **0.970** | 0.542 | 0.465 | 0.975 | 0.502 | 0.512 |
> > > > > > | Contour    | 0.565 | 0.614 | 0.512 | 0.510 | 0.611 | 0.574 | 0.557 | 0.520 | 0.484 |
> > > > > > | Count      | 0.838 | 0.787 | 0.758 | 0.739 | 0.753 | **0.757** | 0.766 | 0.787 | 0.759 |
> > > > > > | Position   | 0.712 | 0.571 | 0.513 | 0.728 | 0.556 | 0.523 | 0.703 | 0.539 | 0.506 |
> > > > > > | Rotation   | 0.830 | 0.683 | 0.631 | 0.706 | 0.622 | 0.580 | 0.858 | 0.616 | 0.643 |
> > > > > > | Shape      | **0.997** | **0.811** | 0.778 | 0.898 | **0.785** | 0.732 | **0.990** | **0.811** | 0.783 |
> > > > > > | Similarity | 0.796 | 0.780 | 0.642 | 0.768 | 0.741 | 0.661 | 0.816 | 0.725 | 0.623 |
> > > > > > | Size       | 0.880 | 0.787 | **0.786** | 0.868 | 0.746 | 0.736 | 0.953 | 0.776 | **0.801** |
> > > > > >
> > > > > > Experiments concerning proprietary models are discussed in the revised Appendix C.11, with results presented in the newly introduced Figure 18, and extended Tables 14 and 15.

---

> > > > > > > ### Author Response · Authors · 2025-12-02
> > > > > > > **Follow-up response to Reviewer kdgb #Q3**
> > > > > > >
> > > > > > > In our first response, we summarized our prompting strategy and, motivated by the Reviewer’s question, described two prompt ablations, reflecting the chain-of-thought (CoT) and explicit analogy prompting (EAP) strategies. In this response we present the results of all four open models with these modified prompting strategies on the I1S and CS (K=16 and K=8) tasks. Tables below report the results.
> > > > > > >
> > > > > > > In I1S, both CoT and EAP yield small gains for the two stronger models (InternVL2.5 and Qwen2-VL), while the two weaker models remain near the random-guessing threshold across all prompting styles. In contrast, for the CS task, our original few-shot prompting performs best overall: both CoT and EAP reduce accuracy, in some cases significantly (e.g., for K=16, InternVL2.5 decreases from 56.9% to 46.7% with CoT and to 47.4% with EAP).
> > > > > > >
> > > > > > > These findings suggest that the few-shot prompting approach used in the paper is generally robust, though alternative approaches can provide improvements in certain settings. We thank the Reviewer for raising this point, which highlights an interesting direction for further investigation.
> > > > > > >
> > > > > > > **Table Q3.1:** I1S prompt ablations.
> > > > > > >
> > > > > > > | Model    | Few-shot |   CoT |  EAP  |
> > > > > > > |:---------|---------:|------:|------:|
> > > > > > > | IVL2.5   |     49.9 |  53.8 |  53.3 |
> > > > > > > | Q2VL     |     49.2 |  52.6 |  50.9 |
> > > > > > > | LLaVA    |     50.0 |  50.0 |  49.3 |
> > > > > > > | MCPM     |     49.2 |  50.2 |  50.0 |
> > > > > > >
> > > > > > >
> > > > > > > **Table Q3.2:** CS $K=16$ prompt ablations.
> > > > > > >
> > > > > > > | Model    | Few-shot |   CoT |  EAP  |
> > > > > > > |:---------|---------:|------:|------:|
> > > > > > > | IVL2.5   |     56.9 |  46.7 |  47.4 |
> > > > > > > | Q2VL     |     37.8 |  35.0 |  38.7 |
> > > > > > > | LLaVA    |     19.4 |   7.8 |  11.1 |
> > > > > > > | MCPM     |     19.1 |  11.3 |  10.0 |
> > > > > > >
> > > > > > >
> > > > > > > **Table Q3.3:** CS $K=8$ prompt ablations.
> > > > > > >
> > > > > > > | Model    | Few-shot |   CoT |  EAP  |
> > > > > > > |:---------|---------:|------:|------:|
> > > > > > > | IVL2.5   |     75.0 |  70.0 |  64.6 |
> > > > > > > | Q2VL     |     63.4 |  51.1 |  52.3 |
> > > > > > > | LLaVA    |     29.4 |  13.0 |  18.7 |
> > > > > > > | MCPM     |     35.1 |  23.5 |  20.9 |

---

### Author Response · Authors · 2025-12-03
**Final Authors' Remarks**

We sincerely thank the Reviewers for their thoughtful and constructive feedback. Our rebuttal provides detailed responses to all Reviewer comments. To assist the newly assigned Area Chairs in evaluating our work, we prepared a summary of the rebuttal.

Across Reviewers, several key strengths were highlighted:
1. **Dataset construction pipeline:** Bongard-RWR+ provides a scalable expansion of Bongard-RWR from 60 to 5,400 instances using a semi-automated VLM + human expert pipeline while preserving fine-grained concepts **(all four Reviewers)**.
1. **Comprehensive evaluation:** The benchmark spans diverse task formulations, including four binary classification setups (I1S, I2S, D1S, D2S), multiclass classification (CS), and free-form text generation **(Reviewers kdgb, SHcE, c11F)**, with extensive ablations on model size, color, image diversity, and number of demonstrations **(all four Reviewers)**.
1. **Diagnostic evaluation:** Fine-grained error analyses isolate specific VLM failures across concepts and guide future model design **(Reviewers kdgb and sMpD)**.
1. **Clarity and quality of presentation:** Reviewers found the paper clear, well-structured, and well-documented **(Reviewers sMpD, SHcE, c11F)**.

**Reviewer c11F** explicitly noted that they see no clear weaknesses in the paper.

The Reviewers highlighted the following key points for discussion:
1. **Proprietary model evaluation (Reviewers sMpD and kdgb)**: Reviewers noted that the paper lacked comparison against leading proprietary VLMs. In response, we added evaluations of Gemini 2.5 Pro, Claude 4.5 Sonnet, and GPT-5.1 on the I1S, D1S, and CS ($K=16$) tasks (Fig. 18a), along with concept group analysis (Fig. 18b). Results are included in Appendix C.11.
1. **Deeper analysis of error sources and failure modes (Reviewers SHcE and sMpD):** Reviewers asked for clearer identification of where and why VLMs fail. We added detailed error analysis in Appendix C.10 and highlighted four main failure modes in Fig. 17, showing that fine-grained concepts, the key differentiator between our work and literature benchmarks, are especially challenging for contemporary models. We also expanded our study of how input modality shapes concept difficulty in Appendix C.9 with visualization in Fig. 16.
1. **Reliance on generated images with potential demographic bias (Reviewer SHcE):** Concern was raised about validity of using generated images. We conducted a statistical analysis showing that generated images preserve the difficulty of real images for fine-grained concept recognition. This analysis is included in Appendix C.2, with results in the newly added Table 4. We also clarified in Appendix D that demographic imbalances do not affect the benchmark’s validity for evaluating abstract visual reasoning.
1. **Quality of generated images (Reviewer sMpD):** The reviewer noted that generated images may introduce ambiguity or noise. We clarified that our human filtering step strictly discarded any image that could obscure the underlying concept (Appendix E.2). Reviewers explicitly highlighted the capability of the pipeline to generate photo-realistic images preserving fine-grained concepts (Reviewers kdgb and c11F). Moreover, existing results support the quality of generated images: accuracy increases with more demonstrations (Section 4; Appendix C.3), adding a captioning step improves performance (Table 3), indicating that models can reliably perceive the images.
1. **Dependence on human review (Reviewers sMpD and kdgb):** Reviewers asked how the pipeline could be refined for more reliable image generation. We added four specific directions for pipeline improvement in Appendix D: advancing T2I models, reducing demographic bias, cycle-consistency verification, and property-based testing. We also added new results showing that noisy images hinder concept recognition (Section 4, Appendix C.5), reinforcing the benefits of human review.
1. **Image diversity (Reviewer sMpD):** The reviewer acknowledged solid methodology ensuring image diversity and requested a more systematic verification of the claim that image diversity facilitates concept recognition. To validate the claim, we added a new paragraph in Section 4 and a new visualization (Fig. 10) showing that accuracy increases monotonically with image diversity.

We thank the Reviewers again for their insightful comments, which helped us substantially strengthen the paper.

---

### Meta-Review · Area_Chair_NASJ · 2025-12-23

**Summary:**

The submission introduces a Bongard Problem (BP) dataset composed of instances that represent original BP abstract concepts using real-world-like images.  Reviewers liked the idea but had concerns regarding the dataset scalability (as manual intervention is involved), potential dataset bias, limited comparisons, and insufficient analyses.

**Reviewer Concerns:**

Post rebuttal, most concerns have been properly addressed.  The outstanding concern appears to be the potential dataset bias.

**Reviewer Scores:**

The reviewers did not have the chance to change their rating before the policy change.  Most likely, the reviewers would have converged to borderline to positive ratings (e.g., 8, 4, 4, 8).

---

### Decision · Program_Chairs · 2026-01-26

Accept (Poster)